# On the Robustness of Removal-Based Feature Attributions

**Chris Lin**[*]
University of Washington
clin25@cs.washington.edu

**Ian Covert**[*†]
Stanford University
icovert@stanford.edu

**Su-In Lee**
University of Washington
suinlee@cs.washington.edu

## Abstract

To explain predictions made by complex machine learning models, many feature attribution methods have been developed that assign importance scores to input features. Some recent work challenges the robustness of these methods by showing that they are sensitive to input and model perturbations, while other work addresses this issue by proposing robust attribution methods. However, previous work on attribution robustness has focused primarily on gradient-based feature attributions, whereas the robustness of removal-based attribution methods is not currently well understood. To bridge this gap, we theoretically characterize the robustness properties of removal-based feature attributions. Specifically, we provide a unified analysis of such methods and derive upper bounds for the difference between intact and perturbed attributions, under settings of both input and model perturbations. Our empirical results on synthetic and real-world data validate our theoretical results and demonstrate their practical implications, including the ability to increase attribution robustness by improving the model's Lipschitz regularity.

## 1 Introduction

In recent years, machine learning has shown great promise in a variety of real-world applications. An obstacle to its widespread deployment is the lack of transparency, an issue that has prompted a wave of research on interpretable or explainable machine learning [45, 51, 53, 56, 60, 67]. One popular way of explaining a machine learning model is feature attribution, which assigns importance scores to input features of the model [45, 51, 53, 56, 60]. Many feature attribution methods can be categorized as either *gradient-based* methods that compute gradients of model predictions with respect to input features [6], or *removal-based* methods that remove features to quantify each feature's influence [17]. While substantial progress has been made and current tools are widely used for model debugging and scientific discovery [10, 18, 37, 50], some important concerns remain. Among them is *unclear robustness properties*: feature attributions are vulnerable to adversarial attacks, and even without an adversary appear unstable under small changes to the input or model.

As a motivating example, consider the work of Ghorbani et al. [28], which shows that minor perturbations to an image can lead to substantially different feature attributions while preserving the original prediction. Similar to adversarial examples designed to alter model predictions [29, 43, 61], this phenomenon violates the intuition that explanations should be invariant to imperceptible changes. A natural question, and the focus of this work, is therefore: *When can we guarantee that feature*

---

[*]Equal contribution. [†]Work done while at the University of Washington.

*attributions are robust to small changes in the input or small changes in the model?* In answering this question, we find that previous theoretical work on this topic has primarily focused on gradient-based methods. Hence, in this work we provide a unified analysis to characterize the robustness of removal-based feature attributions, under both input and model perturbations, and considering a range of existing methods in combination with different approaches to implement feature removal.

**Related work.** Recent work has demonstrated that gradient-based attributions for neural networks are susceptible to small input perturbations that can lead to markedly different attributions [21, 28, 41]. Theoretical analyses have established that this issue relates to model smoothness, and approaches like stochastic smoothing of gradients, weight decay regularization, smoothing activation functions, and Hessian minimization have been shown to generate more robust gradient-based attributions [4, 21, 22, 65]. As for the robustness of removal-based attributions under input perturbations, Alvarez-Melis and Jaakkola [5] empirically assess the robustness of LIME, SHAP, and Occlusion with the notion of Lipschitz continuity in local neighborhoods. Khan et al. [40] provide theoretical guarantees on the robustness of SHAP, RISE, and leave-one-out attributions using Lipschitz continuity, in the specific setting where held-out features are replaced with zeros. Agarwal et al. [3] study the robustness of discrete attributions for graph neural networks and establish Lipschitz upper bounds for the robustness of GraphLIME and GraphMASK. Our analysis of input perturbations is most similar to Khan et al. [40], but we generalize this work by considering alternative techniques for both feature removal and importance summarization (e.g., LIME's weighted least squares approach [53]).

Other works have considered model perturbations and manipulated neural networks to produce targeted, arbitrary gradient-based attributions [7, 34]. Anders et al. [7] explain this phenomenon through the insight that neural network gradients are underdetermined with many degrees of freedom to exploit, and they demonstrate better robustness when gradient-based attributions are projected onto the tangent space of the data manifold. As for removal-based attributions, it has been demonstrated that models can be modified to hide their reliance on sensitive features in LIME and SHAP [20, 58]; however, Frye et al. [27] show that such hidden features can be discovered if features are removed using their conditional distribution, suggesting a key role for the feature removal approach in determining sensitivity to such attacks. Our work formalizes these results through the lens of robustness under model perturbations, and provides new theoretical guarantees.

**Contribution.** The main contribution of this work is to develop a comprehensive characterization of the robustness properties of removal-based feature attributions, focusing on two notions of robustness that have received attention in the literature: stability under input changes and stability under model changes. Our specific contributions are the following: (1) We provide theoretical guarantees for the robustness of model predictions with the removal of arbitrary feature subsets, under both input and model perturbations. (2) We analyze the robustness of techniques for summarizing each feature's influence (e.g., Shapley values) and derive best- and worst-case robustness within several classes of such techniques. (3) We combine the analyses above to prove unified attribution robustness properties to both input and model perturbations, where changes in removal-based attributions are controlled by the scale of perturbations in either input space or function space. (4) We validate our theoretical results and demonstrate practical implications using synthetic and real-world datasets.

## 2 Background

Here, we introduce the notation used in the paper and review removal-based feature attributions.

### 2.1 Notation

Let $f : \mathbb{R}^d \mapsto \mathbb{R}$ be a model whose predictions we seek to explain. We consider that the input variable $\boldsymbol{x}$ consists of $d$ separate features, or $\boldsymbol{x} = (\boldsymbol{x}_1, \ldots, \boldsymbol{x}_d)$. Given an index set $S \subseteq [d] \equiv \{1, \ldots, d\}$, we define the corresponding feature subset as $\boldsymbol{x}_S \equiv \{\boldsymbol{x}_i : i \in S\}$. The power set of $[d]$ is denoted by $\mathcal{P}(d)$, and we let $\bar{S} \equiv [d] \setminus S$ denote the set complement. We consider the model's output space to be one-dimensional, which is not restrictive because attributions are typically calculated for scalar predictions (e.g., the probability for a given class). We use the bold symbols $\boldsymbol{x}, \boldsymbol{x}_S$ to denote random variables, the symbols $x, x_S$ to denote specific values, and $p(\boldsymbol{x})$ to represent the data distribution with support on $\mathcal{X} \subseteq \mathbb{R}^d$. We assume that all explanations are generated for inputs such that $x \in \mathcal{X}$.

## 2.2 Removal-based feature attributions

Our work focuses on a class of methods known as *removal-based explanations* [17]. Intuitively, these are algorithms that remove subsets of inputs and summarize how each feature affects the model. This framework describes a large number of methods that are distinguished by two main implementation choices: (1) how feature information is removed from the model, and (2) how the algorithm summarizes each feature's influence.[2] This perspective shows a close connection between methods like leave-one-out [67], LIME [53] and Shapley values [45], and enables a unified analysis of their robustness properties. Below, we describe the two main implementation choices in detail.

**Feature removal.** Most machine learning models require all feature values to generate predictions, so we must specify a convention for depriving the model of feature information. We denote the prediction given partial inputs by $f(x_S)$. Many implementations have been discussed in the literature [17], but we focus here on three common choices. Given a set of observed values $x_S$, these techniques can all be viewed as averaging the prediction over a distribution $q(\boldsymbol{x}_{\bar{S}})$ for the held-out feature values:

$$f(x_S) := \mathbb{E}_{q(\boldsymbol{x}_{\bar{S}})}\left[f(x_S, \boldsymbol{x}_{\bar{S}})\right] = \int f(x_S, x_{\bar{S}})q(x_{\bar{S}})dx_{\bar{S}}. \tag{1}$$

Specifically, the three choices we consider are:

- (Baseline values) Given a baseline input $b \in \mathbb{R}^d$, we can set the held-out features to their corresponding values $b_{\bar{S}}$ [59]. This is equivalent to letting $q(\boldsymbol{x}_{\bar{S}})$ be a Dirac delta centered at $b_{\bar{S}}$.
- (Marginal distribution) Rather than using a single replacement value, we can average across values sampled from the input's marginal distribution, or let $q(\boldsymbol{x}_{\bar{S}}) = p(\boldsymbol{x}_{\bar{S}})$ [45].
- (Conditional distribution) Finally, we can average across replacement values while conditioning on the available features, which is equivalent to setting $q(\boldsymbol{x}_{\bar{S}}) = p(\boldsymbol{x}_{\bar{S}} \mid x_S)$ [27].

The first two options are commonly implemented in practice because they are simple to estimate, but the third choice is viewed by some work as more informative to users [1, 17, 27]. The conditional distribution approach requires more complex and error-prone estimation procedures (see [12] for a discussion), but we assume here that all three versions of $f(x_S)$ can be calculated exactly.

**Summary technique.** Given a feature removal technique that allows us to query the model with arbitrary feature sets, we must define a convention for summarizing each feature's influence. This is challenging due to the exponential number of feature sets, or because $|\mathcal{P}(d)| = 2^d$. Again, many techniques have been discussed in the literature [17], with the simplest option comparing the prediction with all features included and with a single feature missing [67]. We take a broad perspective here, considering a range of approaches that define attributions as a linear combination of predictions with different feature sets. These methods include leave-one-out [67], RISE [51], LIME [53], Shapley values [45], and Banzhaf values [13], among other possible options.

All of these methods yield per-feature attribution scores $\phi_i(f, x) \in \mathbb{R}$ for $i \in [d]$. For example, the Shapley value calculates feature attribution scores as follows [45]:

$$\phi_i(f, x) = \frac{1}{d} \sum_{S \subseteq [d] \setminus \{i\}} \binom{d-1}{|S|}^{-1} \left(f(x_{S \cup \{i\}}) - f(x_S)\right). \tag{2}$$

The specific linear combinations for the other approaches we consider are shown in Table 3. The remainder of the paper uses the notation $f(x_S)$ to refer to predictions with partial information, and $\phi(f, x) = [\phi_1(f, x), \ldots, \phi_d(f, x)] \in \mathbb{R}^d$ to refer to feature attributions. We do not introduce separate notation to distinguish between implementation choices, opting instead to make the relevant choices clear in each result.

## 2.3 Problem formulation

The problem formulation in this work is straightforward: our goal is to understand the stability of feature attributions under input perturbations and model perturbations. Formally, we aim to study

---

[2]The framework also considers the choice to explain different *model behaviors* (e.g., the loss calculated over one or more examples) [17], but we focus on methods that explain individual predictions.

1. whether $\|\phi(f, x) - \phi(f, x')\|$ is controlled by $\|x - x'\|$ (**input perturbation**), and
2. whether $\|\phi(f, x) - \phi(f', x)\|$ is controlled by $\|f - f'\|$ (**model perturbation**).

The following sections show how this can be guaranteed using certain distance metrics, and under certain assumptions about the model and/or the data distribution.

## 3 Preliminary results

As an intermediate step towards understanding explanation robustness, we first provide results for sub-components of the algorithms. We begin by addressing the robustness of predictions with partial information to input and model perturbations, and we then discuss robustness properties of the summary technique. Proofs for all results are in the Appendix.

Before proceeding, we introduce two assumptions that are necessary for our analysis.

**Assumption 1.** *We assume that the model $f$ is globally $L$-Lipschitz continuous, or that we have $|f(x) - f(x')| \leq L \cdot \|x - x'\|_2$ for all $x, x' \in \mathbb{R}^d$.*

**Assumption 2.** *We assume that the model $f$ has bounded predictions, or that there exists a constant $B$ such that $|f(x)| \leq B$ for all $x \in \mathbb{R}^d$.*

The first assumption holds for most deep learning architectures, because they typically compose a series of Lipschitz continuous layers [63]. The second assumption holds with $B = 1$ for classification models. These are therefore mild assumptions, but they are discussed further in Section 6.

We also define the notion of a functional norm, which is useful for several of our results.

**Definition 1.** *The $L^p$ norm for a function $g : \mathbb{R}^d \mapsto \mathbb{R}$ is defined as $\|g\|_p \equiv (\int |g(x)|^p dx)^{1/p}$, where the integral is taken over $\mathbb{R}^d$. $\|g\|_{p, \mathcal{X}'}$ denotes the same integral taken over the domain $\mathcal{X}' \subseteq \mathbb{R}^d$.*

Our results adopt this to define a notion of distance between functions $\|g - g'\|_p$, focusing on the cases where $p = 1$ for the $L^1$ distance and $p = \infty$ for the Chebyshev distance.

### 3.1 Prediction robustness to input perturbations

Our goal here is to understand how the prediction function $f(x_S)$ for a feature set $x_S$ behaves under small input perturbations. We first consider the case where features are removed using the baseline or marginal approach, and we find that Lipschitz continuity is inherited from the original model.

**Lemma 1.** *When removing features using either the **baseline** or **marginal** approaches, the prediction function $f(x_S)$ for any feature set $x_S$ is $L$-Lipschitz continuous:*

$$|f(x_S) - f(x'_S)| \leq L \cdot \|x_S - x'_S\|_2 \quad \forall \, x_S, x'_S \in \mathbb{R}^{|S|}.$$

Next, we consider the case where features are removed using the conditional distribution approach. We show that the continuity of $f(x_S)$ depends not only on the original model, but also on the similarity of the conditional distribution for the held-out features.

**Lemma 2.** *When removing features using the **conditional** approach, the prediction function $f(x_S)$ for a feature set $x_S$ satisfies*

$$|f(x_S) - f(x'_S)| \leq L \cdot \|x_S - x'_S\|_2 + 2B \cdot d_{TV}\Big(p(\boldsymbol{x}_{\bar{S}} \mid x_S), p(\boldsymbol{x}_{\bar{S}} \mid x'_S)\Big),$$

*where the total variation distance is defined via the $L^1$ functional distance as*

$$d_{TV}\Big(p(\boldsymbol{x}_{\bar{S}} \mid x_S), p(\boldsymbol{x}_{\bar{S}} \mid x'_S)\Big) \equiv \frac{1}{2}\Big\|p(\boldsymbol{x}_{\bar{S}} \mid x_S) - p(\boldsymbol{x}_{\bar{S}} \mid x'_S)\Big\|_1.$$

Lemma 2 does not immediately imply Lipschitz continuity for $f(x_S)$, because we have not bounded the total variation distance in terms of $\|x_S - x'_S\|_2$. To address this, we require the following Lipschitz-like continuity property in the total variation distance.

**Assumption 3.** *We assume that there exists a constant $M$ such that for all $S \subseteq [d]$, we have*

$$d_{TV}\Big(p(\boldsymbol{x}_{\bar{S}} \mid x_S), p(\boldsymbol{x}_{\bar{S}} \mid x'_S)\Big) \leq M \cdot \|x_S - x'_S\|_2 \quad \forall x_S, x'_S \in \mathbb{R}^{|S|}.$$

Intuitively, this says that the conditional distribution $p(\boldsymbol{x}_{\bar{S}} \mid x_S)$ cannot change too quickly as a function of the observed features. This property does not hold for all data distributions $p(\boldsymbol{x})$, and it may hold in some cases only for large $M$; in such scenarios, the function $f(x_S)$ is not guaranteed to change slowly. However, there are cases where it holds. For example, we have $M = 0$ if the features are independent, and the following example shows a case where it holds with dependent features.

**Example 1.** *For a Gaussian random variable $\boldsymbol{x} \sim \mathcal{N}(\mu, \Sigma)$ with mean $\mu \in \mathbb{R}^d$ and covariance $\Sigma \in \mathbb{R}^{d \times d}$, Assumption 3 holds with $M = \frac{1}{2}\sqrt{\lambda_{\max}(\Sigma^{-1}) - \lambda_{\min}(\Sigma^{-1})}$. If $\boldsymbol{x}$ is assumed to be standardized, this captures the case where independent features yield $M = 0$. Intuitively, it also means that if one dimension is roughly a linear combination of the others, or if $\lambda_{\max}(\Sigma^{-1})$ is large, the conditional distribution can change quickly as a function of the conditioning variables.*

Now, assuming that this property holds for $p(\boldsymbol{x})$, we show that we can improve upon Lemma 2.

**Lemma 3.** *Under Assumption 3, the prediction function $f(x_S)$ defined using the **conditional** approach is Lipschitz continuous with constant $L + 2BM$ for any feature set $x_S$.*

Between Lemmas 1 and 3, we have established that the function $f(x_S)$ remains Lipschitz continuous for any feature set $x_S$, although in some cases with a larger constant that depends on the data distribution. These results are summarized in Table 1. In Appendix B we show that our analysis can be extended to account for sampling in eq. (1) when the expectation is not calculated exactly.

Table 1: Lipschitz constants induced by each feature removal technique.

| Baseline | Marginal | Conditional |
|----------|----------|-------------|
| $L$ | $L$ | $L + 2BM$ |

Table 2: Relevant functional distances for each feature removal technique.

| Baseline | Marginal | Conditional |
|----------|----------|-------------|
| $\|f - f'\|_\infty$ | $\|f - f'\|_\infty$ | $\|f - f'\|_{\infty,\mathcal{X}}$ |

## 3.2 Prediction robustness to model perturbations

Our next goal is to understand how the function $f(x_S)$ for a feature set $x_S$ behaves under small changes to the model. The intuition is that if two models make very similar predictions with all features, they should continue to make similar predictions with a subset of features. We first derive a general result involving the proximity between two models $f$ and $f'$ within a subdomain.

**Lemma 4.** *For two models $f, f' : \mathbb{R}^d \mapsto \mathbb{R}$ and a subdomain $\mathcal{X}' \subseteq \mathbb{R}^d$, the prediction functions $f(x_S), f'(x_S)$ for any feature set $x_S$ satisfy*

$$|f(x_S) - f'(x_S)| \leq \|f - f'\|_{\infty,\mathcal{X}'} \cdot Q_{x_S}(\mathcal{X}') + 2B \cdot \left(1 - Q_{x_S}(\mathcal{X}')\right),$$

*where $Q_{x_S}(\mathcal{X}')$ is the probability of imputed samples lying in $\mathcal{X}'$ based on the distribution $q(\boldsymbol{x}_{\bar{S}})$:*

$$Q_{x_S}(\mathcal{X}') \equiv \mathbb{E}_{q(\boldsymbol{x}_{\bar{S}})} \left[\mathbb{I}\left\{(x_S, \boldsymbol{x}_{\bar{S}}) \in \mathcal{X}'\right\}\right].$$

The difference in predictions therefore depends on the distance between the models only within the subdomain $\mathcal{X}'$, as well as the likelihood of imputed sampled lying in this subdomain. This result illustrates a point shown by Slack et al. [58]: that two models which are equivalent on a small subdomain, or even on the entire data manifold $\mathcal{X}$, can lead to different attributions if we use a removal technique that yields a low value for $Q_{x_S}(\mathcal{X}')$. In fact, when $Q_{x_S}(\mathcal{X}') \to 0$, Lemma 4 reduces to a trivial bound $|f(x_S) - f'(x_S)| \leq 2B$ that follows directly from Assumption 2.

The general result in Lemma 4 allows us to show two simpler ones. In both cases, we choose the subdomain $\mathcal{X}'$ and removal technique $q(\boldsymbol{x}_{\bar{S}})$ so that $Q_{x_S}(\mathcal{X}') = 1$.

**Lemma 5.** *When removing features using the **conditional** approach, the prediction functions $f(x_S), f'(x_S)$ for two models $f, f'$ and any feature set $x_S$ satisfy*

$$|f(x_S) - f'(x_S)| \leq \|f - f'\|_{\infty,\mathcal{X}}.$$

The next result is similar but involves a potentially larger upper bound $\|f - f'\|_\infty \geq \|f - f'\|_{\infty,\mathcal{X}}$.

**Lemma 6.** *When removing features using the **baseline** or **marginal** approach, the prediction functions $f(x_S), f'(x_S)$ for two models $f, f'$ and any feature set $x_S$ satisfy*

$$|f(x_S) - f'(x_S)| \leq \|f - f'\|_\infty.$$

Table 3: Summary technique linear operators used by various removal-based explanations.

| Summary | Method | $A_{iS}\ (i \in S)$ | $A_{iS}\ (i \notin S)$ | $\|A\|_{1,\infty}$ | $\|A\|_2$ |
|---|---|---|---|---|---|
| Leave-one-out | Occlusion [67] | $\mathbb{I}\{|S|=d\}$ | $-\mathbb{I}\{|S|=d-1\}$ | $2\sqrt{d}$ | $\sqrt{d+1}$ |
| Shapley value | SHAP [45] | $\frac{(|S|-1)!(d-|S|)!}{d!}$ | $-\frac{|S|!(d-|S|-1)!}{d!}$ | $2\sqrt{d}$ | $\sqrt{2/d}$ |
| Banzhaf value | Banzhaf [13] | $1/2^{d-1}$ | $-1/2^{d-1}$ | $2\sqrt{d}$ | $1/2^{d/2-1}$ |
| Mean when included | RISE [51] | $1/2^{d-1}$ | $0$ | $\sqrt{d}$ | $\sqrt{(d+1)/2^d}$ |
| Weighted least squares | LIME [53] | Depends on implementation choices (see Appendix D) | | | |

The relevant functional distances for Lemma 5 and Lemma 6 are summarized in Table 2.

**Remark 1.** Lemma 6 implies that if two models $f, f'$ are functionally equivalent, or $f(x) = f(x')$ for all $x \in \mathbb{R}^d$, they remain equivalent with any feature subset $x_S$. In Section 4, we will see that this implies equal attributions for the two models—a natural property that, perhaps surprisingly, is not satisfied by all feature attribution methods [47, 55], and that has been described in the literature as *implementation invariance* [60]. This property holds automatically for all removal-based methods.

**Remark 2.** In contrast, Lemma 5 implies the same for models that are equivalent *only on the data manifold* $\mathcal{X} \subseteq \mathbb{R}^d$. This is a less stringent requirement to guarantee equal attributions, and it is perhaps reasonable that models with equal predictions for all realistic inputs receive equal attributions. To emphasize this, we propose distinguishing between a notion of *weak implementation invariance* and *strong implementation invariance*, depending on whether equal attributions are guaranteed when we have $f(x) = f'(x)$ everywhere ($x \in \mathbb{R}^d$) or almost everywhere ($x \in \mathcal{X}$).

### 3.3 Summary technique robustness

We previously focused on the effects of the feature removal choice, so our goal is now to understand the role of the summary technique. Given a removal approach that lets us query the model with any feature set, the summary generates attribution scores $\phi(f, x) \in \mathbb{R}^d$, often using a linear combination of the outputs $f(x_S)$ for each $S \subseteq [d]$. We formalize this in the following proposition.

**Proposition 1.** *The attributions for each method can be calculated by applying a linear operator $A \in \mathbb{R}^{d \times 2^d}$ to a vector $v \in \mathbb{R}^{2^d}$ representing the predictions with each feature set, or*

$$\phi(f, x) = Av,$$

*where the linear operator $A$ for each method is listed in Table 3, and $v$ is defined as $v_S = f(x_S)$ for each $S \subseteq [d]$ based on the chosen feature removal technique.*

In this notation, the entries of $v$ are indexed as $v_S$ for $S \subseteq [d]$, and the entries of $A$ are indexed as $A_{iS}$ for $i \in [d]$ and $S \subseteq [d]$. The operation $Av$ can be understood as summing across all subsets, or $(Av)_i = \sum_{S \subseteq [d]} A_{iS} \cdot v_S$. Representing the attributions this way is convenient for our next results.

The entries for the linear operator in Table 3 are straightforward for the first four methods, but LIME depends on several implementation choices: these include the choice of weighting kernel, regularization, and intercept term. The first three methods are in fact special cases of LIME [17]. The class of weighting kernel used in practice is difficult to characterize analytically, but its default parameters for tabular, image and text data approach limiting cases where LIME reduces to other methods (Appendix H). For simplicity, the remainder of this section focuses on the other methods.

Perturbing either the input or model induces a change in $v \in \mathbb{R}^{2^d}$, and we must consider how the attributions differ between the original vector $v$ and a perturbed version $v'$. Proposition 1 suggests that we can bound the attribtuion difference via the change in model outputs $\|v - v'\|$ and properties of the matrix $A$. We can even do this under different distance metrics, as we show in the next result.

**Lemma 7.** *The difference in attributions given the same summary technique $A$ and different model outputs $v, v'$ can be bounded as*

$$\|Av - Av'\|_2 \le \|A\|_2 \cdot \|v - v'\|_2 \qquad or \qquad \|Av - Av'\|_2 \le \|A\|_{1,\infty} \cdot \|v - v'\|_\infty,$$

*where $\|A\|_2$ is the spectral norm, and the operator norm $\|A\|_{1,\infty}$ is the square root of the sum of squared row 1-norms, with values for each $A$ given in Table 3.*

For both inequalities, smaller norms $\|A\|$ represent stronger robustness to perturbations. Neither bound is necessarily tighter, and the choice of which to use depends on how $v - v'$ is bounded. The first bound using the Euclidean distance $\|v - v'\|_2$ is perhaps more intuitive, but the second bound is more useful here because the results in Sections 3.1 and 3.2 effectively relate to $\|v - v'\|_\infty$.

This view of the summary technique's robustness raises a natural question, which is whether the approaches used in practice have relatively good or bad robustness (Table 3). The answer is not immediately clear, because within the class of linear operators we can control the robustness arbitrarily: we can achieve $\|A\|_2 = \|A\|_{1,\infty} = 0$ by setting $A = 0$ (strong robustness), and we can likewise get $\|A\|_2, \|A\|_{1,\infty} \to \infty$ by setting entries of $A$ to a large value (weak robustness). These results are not useful, however, because neither limiting case results in meaningful attributions.

Rather than considering robustness within the class of *all* linear operators $A \in \mathbb{R}^{d \times 2^d}$, we therefore restrict our attention to attributions that are in some sense meaningful. There are multiple ways to define this, but we consider solutions that satisfy one or more of the following properties, which are motivated by axioms from the literature on game-theoretic credit allocation [46, 54].

**Definition 2.** *We define the following properties for linear operators $A \in \mathbb{R}^{d \times 2^d}$ that are used for removal-based explanations:*

- *(Boundedness) For all $v \in \mathbb{R}^{2^d}$, the attributions are bounded by each feature's smallest and largest contributions, or $\min_{S \not\ni i}(v_{S \cup \{i\}} - v_S) \leq (Av)_i \leq \max_{S \not\ni i}(v_{S \cup \{i\}} - v_S)$ for all $i \in [d]$.*

- *(Symmetry) For all $v \in \mathbb{R}^{2^d}$, two features that make equal marginal contributions to all feature sets must have equal attributions, or $(Av)_i = (Av)_j$ if $v_{S \cup \{i\}} = v_{S \cup \{j\}}$ for all $S \subseteq [d]$.*

- *(Efficiency) For all $v \in \mathbb{R}^{2^d}$, the attributions sum to the difference in predictions for the complete and empty sets, or $\mathbf{1}^\top A v = v_{[d]} - v_{\{\}}$.*

Among these properties, we prioritize boundedness because this leads to a class of solutions that are well-known in game theory, and which are called *probabilistic values* [46]. We now show that constraining the summary technique to satisfy different combinations of these properties yields clear bounds for the robustness. We first address the operator norm $\|A\|_{1,\infty}$, which is the simpler case.

**Lemma 8.** *When the linear operator $A$ satisfies the **boundedness** property, we have $\|A\|_{1,\infty} = 2\sqrt{d}$.*

The above result applies to several summary techniques shown in Table 3, including leave-one-out [67], Shapley values [45] and Banzhaf values [13]. We now address the case of the spectral norm $\|A\|_2$, beginning with the case where boundedness and symmetry are satisfied.

**Lemma 9.** *When the linear operator $A$ satisfies the **boundedness** and **symmetry** properties, the spectral norm is bounded as follows,*

$$\frac{1}{2^{d/2 - 1}} \leq \|A\|_2 \leq \sqrt{d + 1},$$

*with $1/2^{d/2 - 1}$ achieved by the Banzhaf value and $\sqrt{d + 1}$ achieved by leave-one-out.*

This shows that the Banzhaf value is the most robust summary within a class of linear operators, which are known as *semivalues* [46]. Its robustness is even better than the Shapley value, a result that was recently discussed by Wang and Jia [64]. However, the Banzhaf value is criticized for failing to satisfy the efficiency property [25], so we now address the case where this property holds.

**Lemma 10.** *When the linear operator $A$ satisfies the **boundedness** and **efficiency** properties, the spectral norm is bounded as follows,*

$$\sqrt{\frac{2}{d}} \leq \|A\|_2 \leq \sqrt{2 + 2 \cdot \cos\left(\frac{\pi}{d + 1}\right)},$$

*with $\sqrt{2/d}$ achieved by the Shapley value.*

We therefore observe that the Shapley value is the most robust choice within a different class of linear operators, which are known as *random-order values* [46]. In comparison, the worst-case robustness

is achieved by a method that has not been discussed in the literature, but which is a special case of a previously proposed method [26]. The upper bound in Lemma 10 is approximately equal to 2, whereas the upper bound in Lemma 9 grows with the input dimensionality, suggesting that the efficiency property imposes a minimum level of robustness but limits robustness in the best case.

Finally, we can trivially say that $\|A\|_2 = \sqrt{2/d}$ when boundedness, symmetry and efficiency all hold, because the Shapley value is the only linear operator to satisfy all three properties [46, 54].

**Lemma 11.** *When the linear operator $A$ satisfies the **boundedness**, **symmetry** and **efficiency** properties, the spectral norm is $\|A\|_2 = \sqrt{2/d}$.*

## 4 Feature attribution robustness: main results

We now shift our attention to the robustness properties of entire feature attribution algorithms. These results build on those presented in Section 3, combining them to provide general results that apply to existing methods, and to new methods that combine their implementation choices arbitrarily.

Our first main result relates to the robustness to input perturbations.

**Theorem 1.** *The robustness of removal-based explanations to input perturbations is given by the following meta-formula,*

$$\|\phi(f, x) - \phi(f, x')\|_2 \leq g(\text{removal}) \cdot h(\text{summary}) \cdot \|x - x'\|_2,$$

*where the factors for each method are defined as follows:*

$$g(\text{removal}) = \begin{cases} L & \text{if removal} = \textbf{\textit{baseline}} \text{ or } \textbf{\textit{marginal}} \\ L + 2BM & \text{if removal} = \textbf{\textit{conditional}}, \end{cases}$$

$$h(\text{summary}) = \begin{cases} 2\sqrt{d} & \text{if summary} = \textbf{\textit{Shapley}}, \textbf{\textit{Banzhaf}}, \text{ or } \textbf{\textit{leave-one-out}} \\ \sqrt{d} & \text{if summary} = \textbf{\textit{mean when included}}. \end{cases}$$

We therefore observe that the explanation functions themselves are Lipschitz continuous, but with a constant that combines properties of the original model with implementation choices of the attribution method. The model's inherent robustness interacts with the removal choice to yield the $g(\text{removal})$ factor, and the summary technique is accounted for by $h(\text{summary})$.

Our second result takes a similar form, but relates to the robustness to model perturbations.

**Theorem 2.** *The robustness of removal-based explanations to model perturbations is given by the following meta-formula,*

$$\|\phi(f, x) - \phi(f', x)\|_2 \leq h(\text{summary}) \cdot \|f - f'\|,$$

*where the functional distance and factor associated with the summary technique are specified as follows:*

$$\|f - f'\| = \begin{cases} \|f - f'\|_\infty & \text{if removal} = \textbf{\textit{baseline}} \text{ or } \textbf{\textit{marginal}} \\ \|f - f'\|_{\infty, \mathcal{X}} & \text{if removal} = \textbf{\textit{conditional}}, \end{cases}$$

$$h(\text{summary}) = \begin{cases} 2\sqrt{d} & \text{if summary} = \textbf{\textit{Shapley}}, \textbf{\textit{Banzhaf}}, \text{ or } \textbf{\textit{leave-one-out}} \\ \sqrt{d} & \text{if summary} = \textbf{\textit{mean when included}}. \end{cases}$$

Similarly, we see here that the attribution difference is determined by the strength of the model perturbation, as measured by a functional distance metric that depends on the feature removal technique. This represents a form of Lipschitz continuity with respect to the model $f$. These results address the case with either an input perturbation or model perturbation, but we can also account for simultaneous input and model perturbations, as shown in Corollary 1 in Appendix E. The remainder of the paper empirically verifies these results and discusses their practical implications.

## 5 Summary of experiments

Due to space constraints, we defer our experiments to Appendix A but briefly describe the results here. First, we empirically validate our results using synthetic data and a logistic regression classifier, where

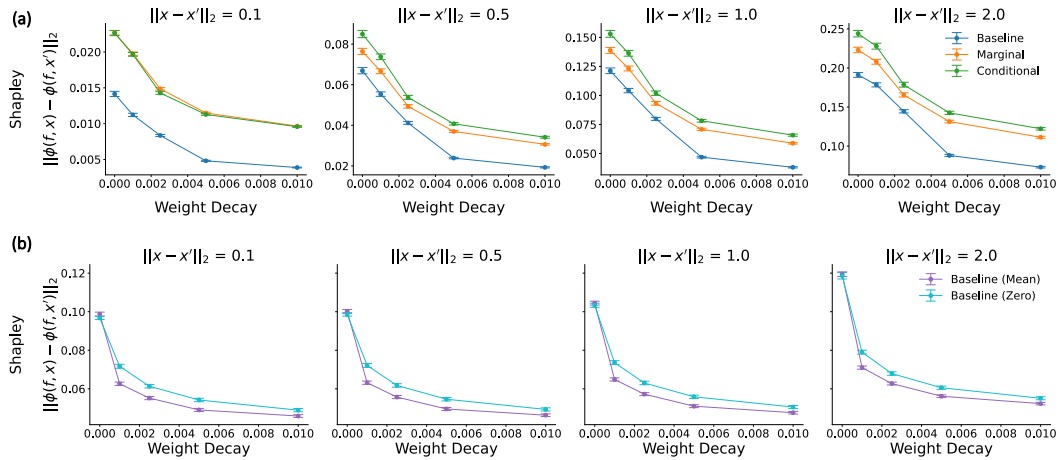

Figure 1: Shapley attribution difference for networks trained with increasing weight decay, under input perturbations with varying perturbation norms. The results include (a) the wine quality dataset with FCNs and baseline, marginal, and conditional feature removal; and (b) MNIST with CNNs and baseline feature removal with either training set means or zeros. Error bars show the mean and $95\%$ confidence intervals across explicand-perturbation pairs.

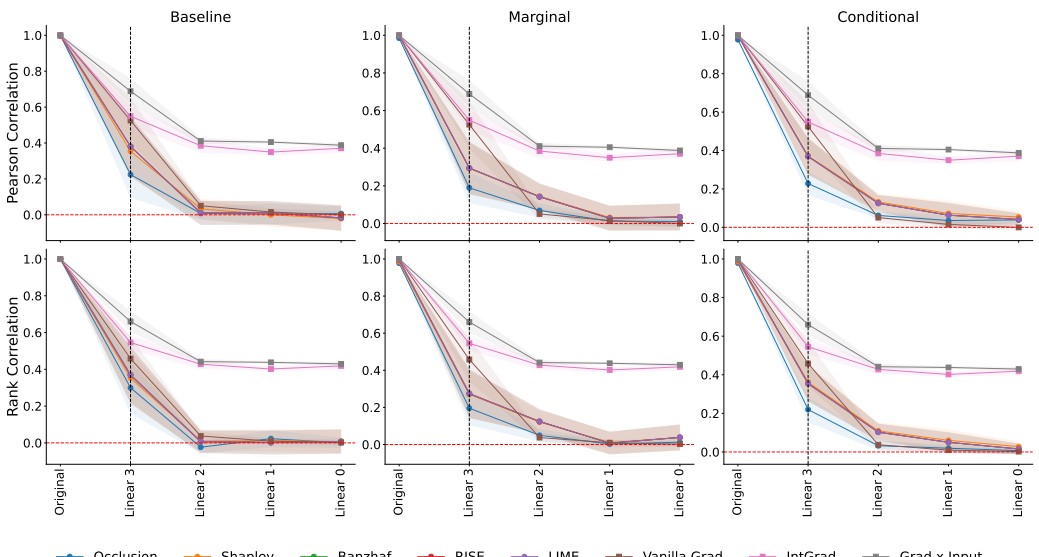

Figure 2: Sanity checks for attributions using cascading randomization for the FCN trained on the wine quality dataset. Attribution similarity is measured by Pearson correlation and Spearman rank correlation. We show the mean and $95\%$ confidence intervals across 10 random seeds.

the model's Lipschitz constant and the upper bound for the total variation constant $M$ can be readily computed. This allows us to verify the robustness bound under input perturbations (Theorem 1). Using the same setup, we construct logistic regression classifiers where the functional distance on the data manifold is small, and we show that feature removal with the conditional distribution is indeed more robust to model perturbations than baseline or marginal removal, as implied by Theorem 2.

Next, we demonstrate two practical implications of our findings with the UCI wine quality [14, 24], MNIST [44], CIFAR-10 [42] and Imagenette datasets [19, 36]. (1) Removal-based attributions are more robust under input perturbations for networks trained with weight decay, which encourages a smaller Lipschitz constant. Formally, this is because a network's Lipschitz constant is upper bounded by the product of its layers' spectral norms [61], which are respectively upper bounded by each layer's Frobenius norm. These results are shown for Shapley value attributions in Figure 1 for the wine quality and MNIST datasets, where we see a consistent effect of weight decay across feature removal approaches. We speculate that a similar effect may be observed with other techniques that encourage

Lipschitz regularity [8, 30, 63]. (2) Viewing the sanity check with parameter randomization proposed by Adebayo et al. [2] as a form of model perturbation, we show that removal-based attributions empirically pass the sanity check, in contrast to certain gradient-based methods [55, 60]. Figure 2 shows results for the wine quality dataset, where the similarity for removal-based methods decays more rapidly than for IntGrad and Grad $\times$ Input. Results for the remaining datasets are in Appendix A.

## 6 Discussion

Our analysis focused on the robustness properties of removal-based feature attributions to perturbations in both the model $f$ and input $x$. For input perturbations, Theorem 1 shows that the attribution change is proportional to the input perturbation strength. The Lipchitz constant is determined by the attribution's removal and summary technique, as well as the model's inherent robustness. We find that both implementation choices play an important role, but our analysis in Section 3 reveals that the associated factors in our main result can only be so small for meaningful attributions, implying that attribution robustness in some sense reduces to model robustness. Fortunately, there are many existing works on improving a neural network's Lipschitz regularity [8, 30, 63].

For model perturbations, Theorem 2 shows that the attribution difference is proportional to functional distance between the perturbed and original models, as measured by the Chebyshev distance (or infinity norm). Depending on the removal approach, the distance may depend on only a subdomain of the input space. One practical implication is that attributions should remain similar under minimal model changes, e.g., accurate model distillation or quantization. Another implication is that bringing the model closer to its correct form (e.g., the Bayes classifier) results in attributions closer to those of the correct model; several recent works have demonstrated this by exploring ensembled attributions [23, 31, 37, 48], which are in some cases mathematically equivalent to attributions of an ensembled model.

Our robustness results provide a new lens to compare different removal-based attribution methods. In terms of the summary technique, our main proof technique relies on the associated operator norm, which does not distinguish between several approaches (Shapley, leave-one-out, Banzhaf); however, our results for the spectral norm suggest that the Banzhaf value is in some sense most robust. This result echoes recent work [64] and has implications for other uses of game-theoretic credit allocation, but we also find that the Shapley value is most robust within a specific class of summary techniques. Our findings regarding the removal approach are more complex: the conditional distribution approach has been praised by some works for being more informative than baseline or marginal [17, 27], but this comes at a cost of *worse* robustness to input perturbations. On the other hand, it leads to *improved* robustness to model perturbations, which has implications for fooling attributions with imperceptible off-manifold model changes [58].

Next, we make connections between current understanding of gradient-based methods and our findings for removal-based methods. Overall, the notion of Lipschitz continuity is important for robustness guarantees under input perturbations. Our results for removal-based attributions rely on Lipschitz continuity of the model itself, whereas gradient-based attributions rely on Lipschitz continuity of the model's *gradient* (i.e., Lipschitz smoothness) [22]. Practically, both are computationally intensive to determine for realistic networks [30, 63]. Empirically, weight decay is an effective approach for generating robust attributions, for both removal-based methods (our results) and gradient-based methods (see Dombrowski et al. [22]), as the Frobenius norms of a neural network's weights upper bound both the network's Lipschitz constant and Hessian norm. Under model perturbations, removal-based attributions are more robust when features are removed such that $x_S$ stay on the data manifold, whereas gradient-based attributions are more robust when gradients are projected on the tangent space of the data manifold [21].

Finally, we consider limitations of our analysis. Future work may consider bounding attribution differences via different distance measures (e.g., $\ell_p$ norms for $p \neq 2$). The main limitation to our results is that they are conservative. By relying on global properties of the model, i.e., the Lipschitz constant and global prediction bounds, we arrive at worst-case bounds that are in many cases overly conservative, as reflected in our experiments. An alternative approach can focus on localized versions of our bounds. Analogous to recent work on certified robustness [38, 39, 52, 57, 62], tighter bounds may be achievable with specific perturbation strengths, unlike our approach that provides bounds simultaneously for any perturbation strength via global continuity properties.

## Acknowledgements

We thank members of the Lee Lab and Jonathan Hayase for helpful discussions. This work was funded by the National Science Foundation [DBI-1552309 and DBI-1759487]; and the National Institutes of Health [R35 GM 128638 and R01 NIA AG 061132].

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

# A   Experiments

This section conducts a range of experiments to provide empirical validation for our theoretical results, and to examine their practical implications for common machine learning models. Code is available at `https://github.com/suinleelab/removal-robustness`.

## A.1   Synthetic data with input perturbation

First, we empirically validate our results from Section 4 using synthetic data and a logistic regression classifier, where the model's Lipschitz constant $L$ and the data distribution's total variation constant $M$ can be readily computed for the bound in Theorem 1.

**Setup.** For each input $x \in \mathbb{R}^4$ in the synthetic dataset, features are generated from a multivariate Gaussian distribution with zero mean, $\boldsymbol{x}_3, \boldsymbol{x}_4$ independent from each other and from $\boldsymbol{x}_1, \boldsymbol{x}_2$, and with a correlation $\rho$ between $\boldsymbol{x}_1, \boldsymbol{x}_2$. That is, the covariance matrix $\Sigma \in \mathbb{R}^{4 \times 4}$ has the following form:

$$\Sigma = \begin{pmatrix} 1 & \rho & 0 & 0 \\ \rho & 1 & 0 & 0 \\ 0 & 0 & 1 & 0 \\ 0 & 0 & 0 & 1 \end{pmatrix}.$$

The binary label $\boldsymbol{y} \in \{0, 1\}$ for each input is sampled from a Bernoulli distribution with probability $\sigma(\beta^\top \boldsymbol{x})$, where $\sigma(\cdot)$ is the sigmoid function and $\beta^\top = [5, 0, 3, 1]$. Unless stated otherwise, we consider $\rho = 1$, zeros are used for baseline feature removal, and the exact Gaussian distributions are used for marginal and conditional feature removal with 20,000 samples.

A logistic regression classifier was trained with 500 samples, and removal-based feature attributions were computed for 50 test samples (i.e., explicands) with and without input perturbation. In our experiment, perturbations are randomly sampled from standard Gaussian distributions with $\ell_2$ norms normalized to values ranging between 0 to 2 with a total of 50 different values. For each perturbation norm, 50 perturbations are generated and applied to each explicand, resulting in 2,500 total explicand-perturbation pairs. As illustrated in Figure 3, a line is drawn to map the maximum $\ell_2$ difference between original and perturbed attributions at each perturbation norm. The slope of such a line estimates the dependency of the upper bound of attribution differences with respect to the perturbation norm. Theorem 1 guarantees a specific slope, and this experiment aims to verify this property.

Furthermore, note that the sigmoid function $\sigma(\cdot)$ is $1/4$-Lipschitz because its largest derivative is $1/4$. A logistic regression classifier is a composition of a linear function and sigmoid function $\sigma(\cdot)$, so the model is $L$-Lipschitz with $L = \|\hat{\beta}\|_2/4$, where $\hat{\beta}$ is the trained model coefficients. Finally, because the input features follow a Gaussian distribution, the total variation constant $M$ derived in Example 1 applies here. Hence, we have specific values for all terms in Theorem 1 bound, which allows a direct comparison between theoretical and empirical results.

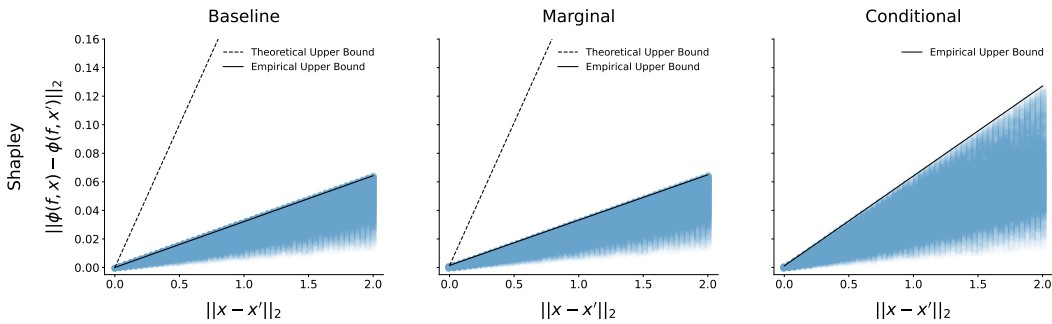

Figure 3: Shapley attribution differences under input perturbation with various perturbation norms in our synthetic data experiment. Lines bounding the maximum attribution difference at each perturbation norm are shown as empirical upper bounds. Theoretical upper bounds are included for baseline and marginal feature removal and omitted for conditional feature removal because it is infinite for $\rho = 1$.

**Results.** As shown in Figure 3, for all three feature removal approaches, the theoretical upper bounds for the Shapley value are indeed greater than the corresponding empirical upper bounds. We also observe that the empirical upper bound with conditional feature removal exceeds those with baseline and marginal feature removal, as expected due to our results involving the conditional distribution's total variation distance (Lemma 2).

Moreover, we vary the Lipschitz constant $L$ of the trained logistic regression classifier by a factor of $\alpha = \{0.05, 0.1, 0.5, 1, 2\}$, which is done by multiplying the trained coefficients $\hat{\beta}$ by $\alpha$. Figure 4 shows that the perturbation norm-dependent slope of the empirical bound increases linearly with $\alpha$, aligning with our theoretical results. Also, Example 1 suggests that increasing $\rho$ would lead to a larger upper bound for conditional feature removal, which is observed in the empirical bound with $\rho = \{0, 0.25, 0.5, 0.75, 1\}$ (see Figure 4 right).

Overall, these empirical results validate our theoretical bound from Theorem 1. We note that gaps between our theoretical and empirical bounds exist, likely because we use the global Lipschitz constant to provide guarantee under all input perturbation strengths, instead of local constants for specific input regions and perturbation strengths, and also due to looseness in the total variation constant $M$. For example, our bound for multivariate Gaussian distributions in Example 1 suggests that $M = \infty$ when $\rho = 1$, which is far more conservative than our empirical results (see Figure 3 right); this is due to looseness in Pinsker's inequality (Appendix B). Similar results are observed for Occlusion, Banzhaf, RISE, and LIME attributions, and these are included in Appendix I.

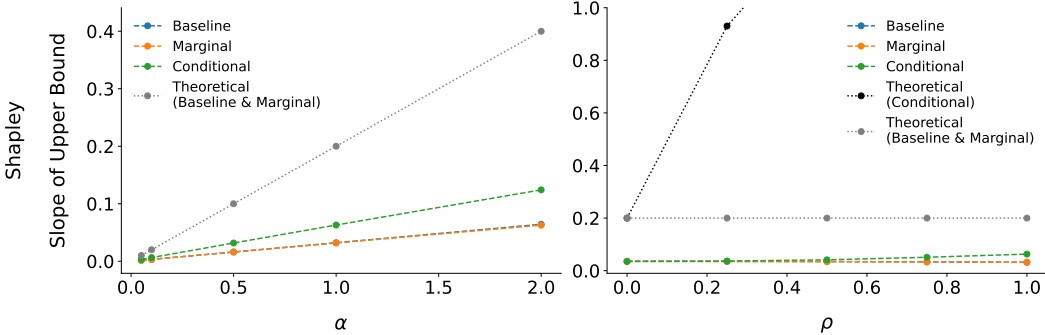

Figure 4: Slopes of empirical and theoretical upper bounds with respect to the input perturbation norm, for Shapley value attribution difference in the synthetic experiment. The parameter $\alpha$ varies the logistic regression Lipschitz constant, and $\rho$ varies the correlation between the two synthetic features $\boldsymbol{x}_1, \boldsymbol{x}_2$.

## A.2 Synthetic data with model perturbation

Here, we construct a logistic regression classifier as well as a perturbed version such that their functional distance on the data manifold is small. We use this construction to empirically verify that removing features with the conditional distribution leads to attributions that are more robust against model perturbation, as suggested by Theorem 2.

**Setup.** The synthetic data generation follows that of Appendix A.1. A logistic regression classifier $f$ is constructed with coefficients $[5, 0, 3, 1]$, and its perturbed counterpart $f'$ has coefficients $[0, 5, 3, 1]$. Because the synthetic features $\boldsymbol{x}_1, \boldsymbol{x}_2$ are identical with the default value $\rho = 1$, we have the functional distance $\|f - f'\|_{\infty, \mathcal{X}} = 0$, so our theoretical bound suggests that $\|\phi(f, x) - \phi(f', x)\|_2 = 0$ for conditional feature removal. When we consider smaller values for $\rho$, we still expect better robustness for conditional rather than marginal removal due to Lemma 4, because significant probability mass lies in a subdomain where $f$ and $f'$ are functionally close (near the hyperplane where $\boldsymbol{x}_1 = \boldsymbol{x}_2$). Feature attributions are computed for 100 explicands with the same feature removal settings as in Appendix A.1.

**Results.** As suggested by our theoretical bound for model perturbation when $\rho = 1$, the Shapley attribution difference between $f, f'$ is near zero when features are removed using their conditional distribution, whereas the attribution difference can be large with baseline and marginal feature removal (Figure 5). When we decrease $\rho$ in the synthetic data generation, the models $f, f'$ become

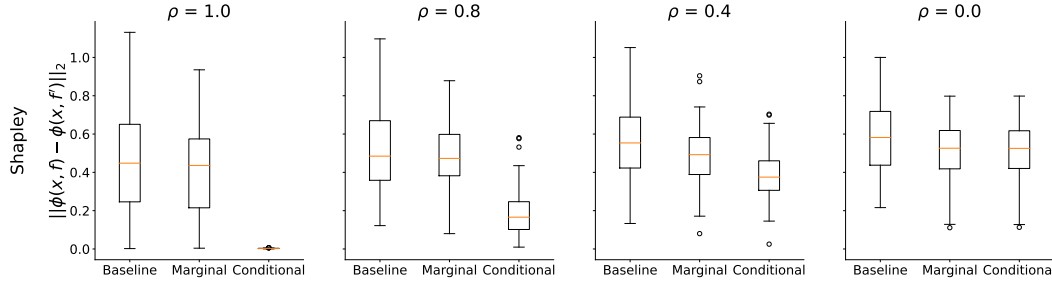

Figure 5: Shapley value attribution differences between the logistic regression classifiers $f, f'$ as constructed in Appendix A.2 with baseline, marginal, and conditional feature removal, and varying correlation $\rho$.

more likely to generate different outputs, and we observe that the robustness gap between conditional vs. baseline and marginal feature removal reduces. These empirical results corroborate Lemma 4. Similar results are observed for Occlusion, Banzhaf, RISE, and LIME attributions and are included in Appendix I. It is worth noting that the arbitrarily large bound for baseline and marginal feature removal underpins the adversarial attacks on LIME and SHAP demonstrated by Slack et al. [58]. Our theory and synthetic experiments for model perturbation are similar to those in Frye et al. [27], but we provide a unified analysis for removal-based attribution methods instead of focusing on the Shapley value, and we consider cases where models disagree even on the data manifold (Lemma 4).

### A.3 Implication I: robust attributions for regularized neural networks

Our theoretical upper bound for input perturbation decreases with the model's Lipschitz constant (Theorem 1), suggesting that training a model with better Lipschitz regularity can help obtain more robust removal-based feature attributions. Weight decay is one way to achieve this, as it encourages lower Frobenius norm for the individual layers in a neural network, and the product of these Frobenius norms upper bounds the network's Lipschitz constant [66]. We therefore examine whether neural networks trained with weight decay produce attributions that are more robust against input perturbation.

**Datasets.** The UCI white wine quality dataset [14, 24] consists of $4,898$ samples of white wine, each with 11 input features for predicting the wine quality (whether the rating $> 6$). Two subsets of $500$ samples are randomly chosen as the validation and test set. For datasets with larger input dimension, we use MNIST [44], CIFAR-10 [42], and ten classes of ImageNet [19] (i.e., Imagenette[3]). From the official training set of MNIST and CIFAR-10, $10,000$ images are randomly chosen as the validation set.

**Setup.** Fully connected networks (FCNs) and convolutional networks (CNNs) are trained with a range of weight decay values for the wine quality dataset and MNIST, respectively, where we use the values $\{0, 0.001, 0.0025, 0.005, 0.01\}$. ResNet-18 networks [33] are trained for CIFAR-10, with weight decay values $\{0.0075, 0.01, 0.025, 0.05, 0.075\}$. ResNet-50 networks pre-trained on ImageNet are fine-tuned for Imagenette, with weight decay values $\{0.005, 0.01, 0.025, 0.05\}$. The test set accuracy does not differ much with these weight decay values for the wine quality dataset, MNIST, and CIFAR-10 (Figure 20 and Figure 21), but large weight decay values can worsen the performance of ResNet-50 on Imagenette (Figure 21). Details on model architectures and other hyperparameters are in Appendix J. Removal-based attributions are computed for the $500$ test samples and all features in the wine quality dataset, and for 100 test images with non-overlapping patches of size $4 \times 4$ as features in MNIST and CIFAR-10, and of size $16 \times 16$ as features in Imagenette (see Appendix G for a discussion of how our theory extends to the use of feature groups). For the wine quality dataset, removal-based attributions are computed exactly with all $2^{11}$ feature subsets. For MNIST, certain attributions must be estimated rather than calculated exactly: Occlusion attributions are computed exactly, whereas Shapley, Banzhaf, LIME, and RISE are estimated with $10,000$ random feature sets, and Shapley and Banzhaf use a least squares regression similar to KernelSHAP [45].

---

[3]`https://github.com/fastai/imagenette`

For CIFAR-10 and Imagenette, we focus on Occlusion, RISE, and Shapley values, computed with the same settings as for MNIST, except RISE and Shapley values are estimated with 20,000 random feature sets for Imagenette due to its large number of features.

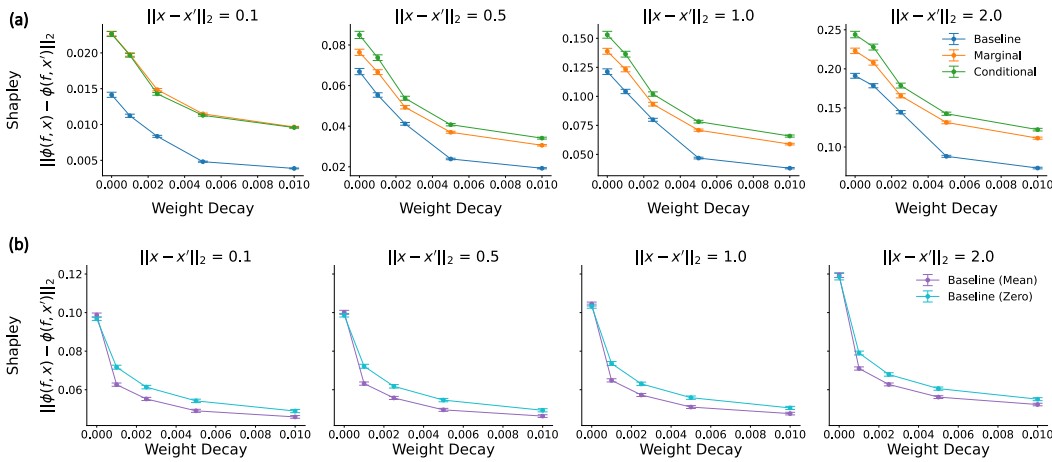

Figure 6: Identical to Figure 1 and shown here again for readability. Shapley attribution difference for networks trained with increasing weight decay, under input perturbations with perturbation norms $\{0.1, 0.5, 1, 2\}$. The results include (a) the wine quality dataset with FCNs and baseline (replacing with training set minimums), marginal, and conditional feature removal; and (b) MNIST with CNNs and baseline feature removal with either training set means or zeros. Error bars show the mean and $95\%$ confidence intervals across explicand-perturbation pairs.

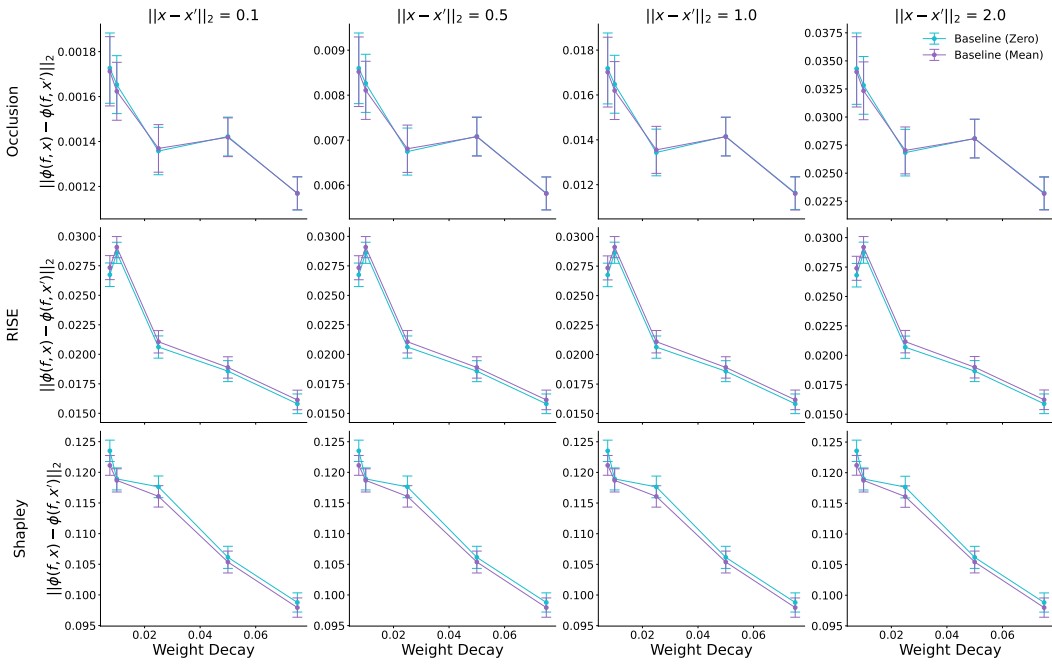

Figure 7: Attribution difference for ResNet-18 networks trained on CIFAR-10 with increasing weight decay, under input perturbations with perturbation norms $\{0.1, 0.5, 1, 2\}$. Error bars show the means and $95\%$ confidence intervals across explicand-perturbation pairs.

When using baseline feature removal, features are replaced with the training set minimums for the wine quality dataset, with the training set means or zeros for MNIST and CIFAR-10, and with zeros (identical to the training set means due to normalization) for Imagenette. The wine quality training set is used for removing features with their marginal distributions, while a conditional variational autoencoder (CVAE) is trained following the procedure in Frye et al. [27] to approximate feature removal with the conditional distribution (details in Appendix J). For both marginal and conditional feature removal in the wine quality dataset, 250 samples are used. Marginal and conditional feature removal are computationally expensive and less common for image data, so we use only baseline feature removal as is often done in practice. Finally, input perturbations are generated as in Appendix A.1, with 10 random perturbations applied per explicand and with perturbation norms $\{0.1, 0.5, 1, 2\}$ for the wine quality dataset, MNIST, and CIFAR-10. For Imagenette, 5 random perturbations with norms $\{50, 100, 200, 400\}$ are applied per explicand to obtain similar perturbation strength per input.

**Results.** Figure 6 shows that Shapley value attribution differences decrease with increasing weight decay, across different feature removal techniques, for both the wine quality dataset and MNIST, and for all perturbation norms in $\{0.1, 0.5, 1, 2\}$. This demonstrates the practical implication that removal-based attributions can be made more robust against input perturbation by training with weight decay. Interestingly, weight decay has also been shown to improve the robustness of gradient-based attributions [22]. In Appendix K, similar results are observed for Occlusion, Banzhaf, RISE, and LIME attributions, and also when we measure explanation similarity using Pearson or Spearman rank correlation rather than $\ell_2$ distance. For ResNet-18 networks trained on CIFAR-10 and ResNet-50 networks trained on Imagenette, similar trends are observed for Occlusion, RISE, and Shapley attributions (Figure 7 and Figure 8).

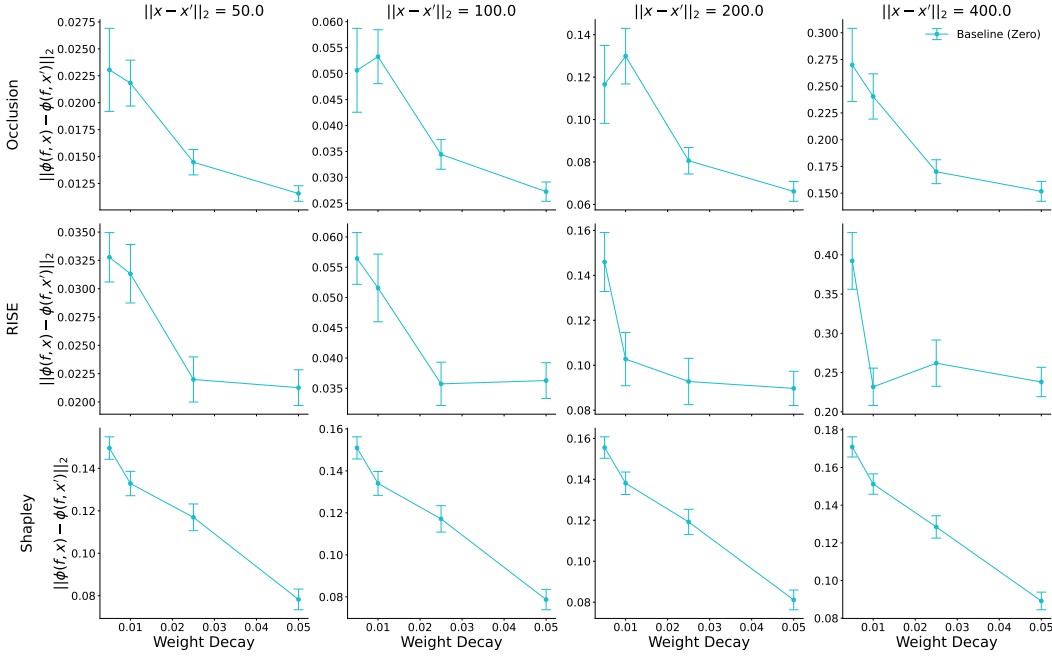

Figure 8: Attribution difference for ResNet-50 networks trained on Imagenette with increasing weight decay, under input perturbations with perturbation norms $\{50, 100, 200, 400\}$. Error bars show the means and $95\%$ confidence intervals across explicand-perturbation pairs.

## A.4 Implication II: sanity checks for removal-based attributions

The robustness bound for model perturbation in Theorem 2 suggests that the attribution difference between an intact and perturbed model should be small if the model perturbation is small. Similarly, we expect that if the perturbation intensifies, the attribution differences may become larger. One way to perturb a neural network with increasing intensity is to sequentially randomize its parameters from the output to input layer, a procedure known as *cascading randomization* introduced by Adebayo

et al. [2] for sanity checking feature attributions. Here, we study whether removal-based attributions indeed pass the parameter randomization check, as suggested by our theoretical bound.

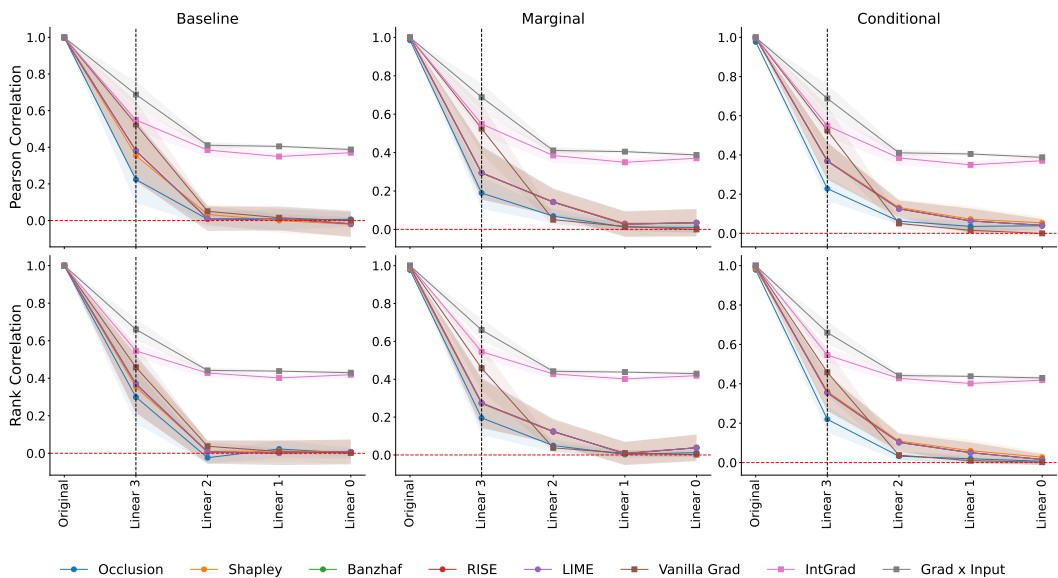

Figure 9: Identical to Figure 2 and shown here again for readability. Sanity checks for attributions using cascading randomization for the FCN trained on the wine quality dataset. Attribution similarity is measured by Pearson correlation and Spearman rank correlation. We show the mean and 95% confidence intervals across 10 random seeds.

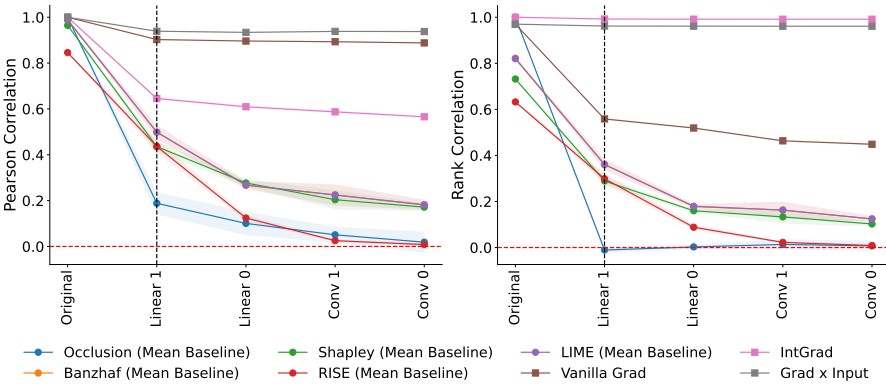

Figure 10: Sanity checks for attributions using cascading randomization for the CNN trained on MNIST. Attribution similarity is measured by Pearson correlation and Spearman rank correlation. We show the mean and 95% confidence intervals across 10 random seeds.

**Datasets.** We use the same datasets and splits as in Appendix A.3.

**Setup.** The best performing networks from Appendix A.3 are used here. The settings for removal-based attributions are the same as in Appendix A.3, except 50,000 random feature sets are used for more stable Shapley estimations in Imagenette. Parameters for each network are randomized in a cascading fashion from the output to input layer as in Adebayo et al. [2], with attributions computed after randomizing each layer and compared to the original attributions using Pearson correlation and Spearman rank correlation. Randomization is performed using the layer's default initialization. Feature attributions with the original model are computed twice and compared to capture variations due to sampling in feature removal and in attribution estimation using feature sets. This experiment is run 10 times with different random seeds for the wine quality dataset, MNIST, and CIFAR-10, and

3 times for Imagenette. Gradient-based attribution methods such as Vanilla Gradients [56], Integrated Gradients [60], and Grad × Input [55] are included as references, as they were the focus of the original work on sanity checks [2].

**Results.** As suggested by our robustness bound for model perturbation, the original and perturbed removal-based attributions become dissimilar as more parameters are randomized, leading to near-zero correlations when the entire network is re-initialized (Figure 9). This shows that removal-based attributions reliably pass the model randomization sanity check. In contrast, we observe that for Integrated Gradients and Grad × Input, the correlation between intact and perturbed attributions remains around 0.5 even when the entire network is randomized. Similarly, for the CNN trained on MNIST, the original and perturbed removal-based attributions have lower correlations as more parameters are randomized, whereas the correlations decrease only slightly for Integrated Gradients and Grad × Input (Figure 10). Similar trends are observed for the ResNet-18 trained on CIFAR-10 and ResNet-50 trained on Imagenette (Figure 11, Figure 12, and Figure 29).

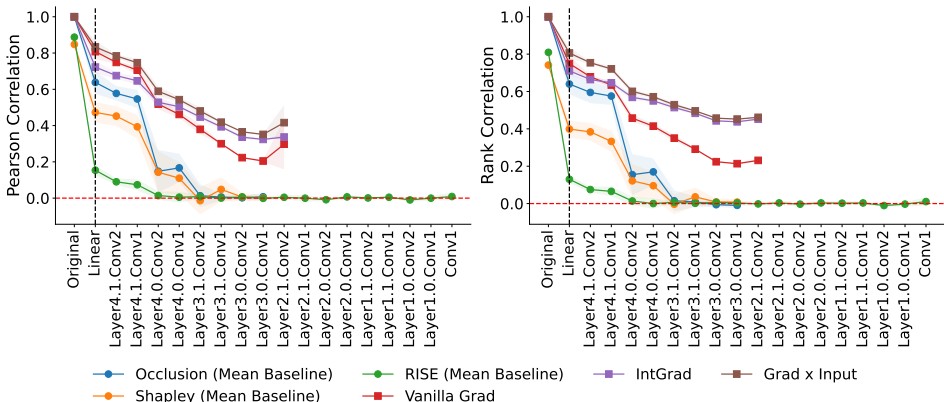

Figure 11: Sanity checks for attributions using cascading randomization for the ResNet-18 trained on CIFAR-10. Features are removed by replacing them with training set means. Attribution similarity is measured by Pearson correlation and Spearman rank correlation. We show the means and 95% confidence intervals across 10 random seeds. Missing points correspond to undefined correlations due to constant attributions under model randomization.

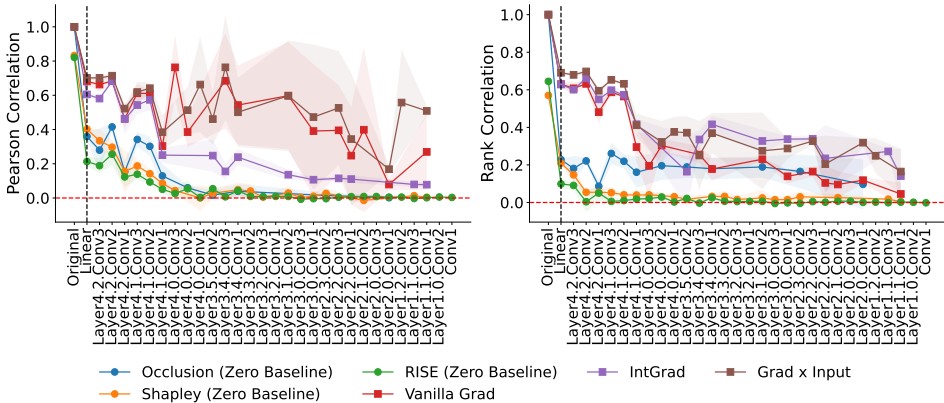

Figure 12: Sanity checks for attributions using cascading randomization for the ResNet-50 trained on Imagenette. Features are removed by replacing them with zeros (identical to training set means due to normalization). Attribution similarity is measured by Pearson correlation and Spearman rank correlation. We show the means and 95% confidence intervals across 3 random seeds. Missing points correspond to undefined correlations due to constant attributions under model randomization.

# B  Proofs: prediction robustness to input perturbations

We now re-state and prove our results from Section 3.1.

**Lemma 1.** *When removing features using either the **baseline** or **marginal** approaches, the prediction function $f(x_S)$ for any feature set $x_S$ is L-Lipschitz continuous:*

$$|f(x_S) - f(x'_S)| \leq L \cdot \|x_S - x'_S\|_2 \quad \forall\, x_S, x'_S \in \mathbb{R}^{|S|}.$$

*Proof.* The key insight for this result is that removing features using either the baseline or marginal approach is equivalent to using a distribution $q(\boldsymbol{x}_{\bar{S}})$ that does not depend on the observed features. Because of that, we have:

$$
\begin{aligned}
|f(x_S) - f(x'_S)| &= \left| \int f(x_S, x_{\bar{S}}) q(x_{\bar{S}}) dx_{\bar{S}} - \int f(x'_S, x_{\bar{S}}) q(x_{\bar{S}}) dx_{\bar{S}} \right| \\
&\leq \int |f(x_S, x_{\bar{S}}) - f(x'_S, x_{\bar{S}})| \cdot q(x_{\bar{S}}) dx_{\bar{S}} \\
&\leq L \cdot \|x_S - x'_S\|_2 \cdot \int q(x_{\bar{S}}) dx_{\bar{S}} \\
&= L \cdot \|x_S - x'_S\|_2.
\end{aligned}
$$

$\square$

**Lemma 2.** *When removing features using the **conditional** approach, the prediction function $f(x_S)$ for a feature set $x_S$ satisfies*

$$|f(x_S) - f(x'_S)| \leq L \cdot \|x_S - x'_S\|_2 + 2B \cdot d_{TV}\Big(p(\boldsymbol{x}_{\bar{S}} \mid x_S), p(\boldsymbol{x}_{\bar{S}} \mid x'_S)\Big),$$

*where the total variation distance is defined via the $L^1$ functional distance as*

$$d_{TV}\Big(p(\boldsymbol{x}_{\bar{S}} \mid x_S), p(\boldsymbol{x}_{\bar{S}} \mid x'_S)\Big) \equiv \frac{1}{2}\Big\|p(\boldsymbol{x}_{\bar{S}} \mid x_S) - p(\boldsymbol{x}_{\bar{S}} \mid x'_S)\Big\|_1.$$

*Proof.* In this case, we must be careful to account for the different distributions used when marginalizing out the removed features.

$$
\begin{aligned}
|f(x_S) - f(x'_S)| &= \left| \int f(x_S, x_{\bar{S}}) p(x_{\bar{S}} \mid x_S) dx_{\bar{S}} - \int f(x'_S, x_{\bar{S}}) p(x_{\bar{S}} \mid x'_S) dx_{\bar{S}} \right| \\
&\leq \int \big| f(x_S, x_{\bar{S}}) p(x_{\bar{S}} \mid x_S) - f(x'_S, x_{\bar{S}}) p(x_{\bar{S}} \mid x'_S) \big| dx_{\bar{S}} \\
&\leq \int \big| f(x_S, x_{\bar{S}}) p(x_{\bar{S}} \mid x_S) - f(x'_S, x_{\bar{S}}) p(x_{\bar{S}} \mid x_S) \big| dx_{\bar{S}} \\
&\quad + \int \big| f(x'_S, x_{\bar{S}}) p(x_{\bar{S}} \mid x_S) - f(x'_S, x_{\bar{S}}) p(x_{\bar{S}} \mid x'_S) \big| dx_{\bar{S}} \\
&\leq \int |f(x_S, x_{\bar{S}}) - f(x'_S, x_{\bar{S}})| \cdot p(x_{\bar{S}} \mid x_S) dx_{\bar{S}} \\
&\quad + \int |f(x'_S, x_{\bar{S}})| \cdot \big| p(x_{\bar{S}} \mid x_S) - p(x_{\bar{S}} \mid x'_S) \big| dx_{\bar{S}} \\
&\leq L \cdot \|x_S - x'_S\|_2 \cdot \int p(x_{\bar{S}} \mid x_S) dx_{\bar{S}} + B \cdot \int \big| p(x_{\bar{S}} \mid x_S) - p(x_{\bar{S}} \mid x'_S) \big| dx_{\bar{S}} \\
&= L \cdot \|x_S - x'_S\|_2 + 2B \cdot d_{TV}\Big(p(x_{\bar{S}} \mid x_S), p(x_{\bar{S}} \mid x'_S)\Big).
\end{aligned}
$$

$\square$

**Example 1.** *For a Gaussian random variable $\boldsymbol{x} \sim \mathcal{N}(\mu, \Sigma)$ with mean $\mu \in \mathbb{R}^d$ and covariance $\Sigma \in \mathbb{R}^{d \times d}$, Assumption 3 holds with $M = \frac{1}{2}\sqrt{\lambda_{\max}(\Sigma^{-1}) - \lambda_{\min}(\Sigma^{-1})}$. If $\boldsymbol{x}$ is assumed to be standardized, this captures the case where independent features yield $M = 0$. Intuitively, it also means that if one dimension is roughly a linear combination of the others, or if $\lambda_{\max}(\Sigma^{-1})$ is large, the conditional distribution can change quickly as a function of the conditioning variables.*

*Proof.* Consider a Gaussian random variable $\boldsymbol{x}$ with mean $\mu \in \mathbb{R}^d$ and positive definite covariance $\Sigma \in \mathbb{R}^{d \times d}$. We can partition the variable into two parts, denoted $\boldsymbol{x}_1$ and $\boldsymbol{x}_2$, and then partition the variable and parameters as follows:

$$\boldsymbol{x} = \begin{pmatrix} \boldsymbol{x}_1 \\ \boldsymbol{x}_2 \end{pmatrix} \qquad \mu = \begin{pmatrix} \mu_1 \\ \mu_2 \end{pmatrix} \qquad \Sigma = \begin{pmatrix} \Sigma_{11} & \Sigma_{12} \\ \Sigma_{21} & \Sigma_{22} \end{pmatrix}.$$

We can then consider the distribution for $\boldsymbol{x}_1$ conditioned on a specific $x_2$ value. The conditional distribution is $\boldsymbol{x}_1 \mid x_2 \sim \mathcal{N}(\bar{\mu}, \bar{\Sigma})$ with the parameters defined as

$$\bar{\mu} = \mu_1 + \Sigma_{12}\Sigma_{22}^{-1}(x_2 - \mu_2) \qquad \bar{\Sigma} = \Sigma_{11} - \Sigma_{12}\Sigma_{22}^{-1}\Sigma_{21}.$$

Our goal is to understand the total variation distance when conditioning on different values $x_2, x_2'$. Pinsker's inequality gives us the following upper bound based on the KL divergence:

$$d_{TV}\left(p(\boldsymbol{x}_1 \mid x_2), p(\boldsymbol{x}_1 \mid x_2')\right) \leq \sqrt{\frac{1}{2}D_{\mathrm{KL}}\left(p(\boldsymbol{x}_1 \mid x_2) \,\|\, p(\boldsymbol{x}_1 \mid x_2')\right)}.$$

The KL divergence between Gaussian distributions has a closed-form solution, and in the case with equal covariance matrices reduces to the following:

$$D_{\mathrm{KL}}\left(p(\boldsymbol{x}_1 \mid x_2) \,\|\, p(\boldsymbol{x}_1 \mid x_2')\right) = \frac{1}{2}(x_2 - x_2')^\top \Sigma_{22}^{-1}\Sigma_{21}\left(\Sigma_{11} - \Sigma_{12}\Sigma_{22}^{-1}\Sigma_{21}\right)^{-1}\Sigma_{12}\Sigma_{22}^{-1}(x_2 - x_2')$$

To provide an upper bound on this term based on $\|x_2 - x_2'\|_2$, it suffices to find the maximum eigenvalue of the above matrix. To characterize the matrix, we observe that it is present in the formula for the block inversion of the joint covariance matrix:

$$\Sigma^{-1} = \begin{pmatrix} \left(\Sigma_{11} - \Sigma_{12}\Sigma_{22}^{-1}\Sigma_{21}\right)^{-1} & -\left(\Sigma_{11} - \Sigma_{12}\Sigma_{22}^{-1}\Sigma_{21}\right)^{-1}\Sigma_{12}\Sigma_{22}^{-1} \\ -\Sigma_{22}^{-1}\Sigma_{21}\left(\Sigma_{11} - \Sigma_{12}\Sigma_{22}^{-1}\Sigma_{21}\right)^{-1} & \Sigma_{22}^{-1} + \Sigma_{22}^{-1}\Sigma_{21}\left(\Sigma_{11} - \Sigma_{12}\Sigma_{22}^{-1}\Sigma_{21}\right)^{-1}\Sigma_{12}\Sigma_{22}^{-1} \end{pmatrix}$$

We therefore have the following, because the matrix we're interested in is a principal minor:

$$\lambda_{\max}\left(\Sigma_{22}^{-1} + \Sigma_{22}^{-1}\Sigma_{21}\left(\Sigma_{11} - \Sigma_{12}\Sigma_{22}^{-1}\Sigma_{21}\right)^{-1}\Sigma_{12}\Sigma_{22}^{-1}\right) \leq \lambda_{\max}(\Sigma^{-1}).$$

Weyl's inequality also gives us:

$$\lambda_{\min}(\Sigma_{22}^{-1}) + \lambda_{\max}\left(\Sigma_{22}^{-1}\Sigma_{21}\left(\Sigma_{11} - \Sigma_{12}\Sigma_{22}^{-1}\Sigma_{21}\right)^{-1}\Sigma_{12}\Sigma_{22}^{-1}\right)$$
$$\leq \lambda_{\max}\left(\Sigma_{22}^{-1} + \Sigma_{22}^{-1}\Sigma_{21}\left(\Sigma_{11} - \Sigma_{12}\Sigma_{22}^{-1}\Sigma_{21}\right)^{-1}\Sigma_{12}\Sigma_{22}^{-1}\right)$$

Combining this with the previous inequality, we arrive at the following:

$$\lambda_{\max}\left(\Sigma_{22}^{-1}\Sigma_{21}\left(\Sigma_{11} - \Sigma_{12}\Sigma_{22}^{-1}\Sigma_{21}\right)^{-1}\Sigma_{12}\Sigma_{22}^{-1}\right) \leq \lambda_{\max}(\Sigma^{-1}) - \lambda_{\min}(\Sigma_{22}^{-1})$$

For the subtractive term, we can write:

$$\lambda_{\min}(\Sigma_{22}^{-1}) = \frac{1}{\lambda_{\max}(\Sigma_{22})} \geq \frac{1}{\lambda_{\max}(\Sigma)} = \lambda_{\min}(\Sigma^{-1}).$$

We therefore have the following upper bound on the KL divergence:

$$D_{\mathrm{KL}}\left(p(\boldsymbol{x}_1 \mid x_2) \,\|\, p(\boldsymbol{x}_1 \mid x_2')\right) \leq \frac{1}{2} \cdot \|x_2 - x_2'\|_2^2 \cdot \left(\lambda_{\max}(\Sigma^{-1}) - \lambda_{\min}(\Sigma^{-1})\right).$$

Combining this with Pinsker's inequality, we arrive at the final result:

$$d_{TV}\left(p(\boldsymbol{x}_1 \mid x_2), p(\boldsymbol{x}_1 \mid x_2'))\right) \leq \frac{1}{2} \cdot \|x_2 - x_2'\|_2 \cdot \sqrt{\lambda_{\max}(\Sigma^{-1}) - \lambda_{\min}(\Sigma^{-1})}.$$

This means that we have $M = \frac{1}{2}\sqrt{\lambda_{\max}(\Sigma^{-1}) - \lambda_{\min}(\Sigma^{-1})}$.

Notice that this does not imply $M = 0$ when the features $\boldsymbol{x}$ are independent: it only captures this for the case of isotropic variance. However, if we assume that $\boldsymbol{x}$ is standardized, so that each dimension has variance equal to 1, $M = 0$ is implied for independent features because we have $\Sigma = I$.

$\square$

**Lemma 3.** *Under Assumption 3, the prediction function $f(x_S)$ defined using the **conditional** approach is Lipschitz continuous with constant $L + 2BM$ for any feature set $x_S$.*

*Proof.* The result follows directly from Lemma 2 and Assumption 3:

$$
\begin{aligned}
|f(x_S) - f(x_S')| &\leq L \cdot \|x_S - x_S'\|_2 + 2B \cdot d_{TV}\left(p(x_{\bar{S}} \mid x_S), p(x_{\bar{S}} \mid x_S')\right) \\
&\leq L \cdot \|x_S - x_S'\|_2 + 2BM \cdot \|x_S - x_S'\|_2 \\
&= (L + 2BM) \cdot \|x_S - x_S'\|_2.
\end{aligned}
$$

$\square$

# C   Proofs: prediction robustness to model perturbations

We now re-state and prove our results from Section 3.2.

**Lemma 4.** *For two models $f, f' : \mathbb{R}^d \mapsto \mathbb{R}$ and a subdomain $\mathcal{X}' \subseteq \mathbb{R}^d$, the prediction functions $f(x_S), f'(x_S)$ for any feature set $x_S$ satisfy*

$$|f(x_S) - f'(x_S)| \leq \|f - f'\|_{\infty, \mathcal{X}'} \cdot Q_{x_S}(\mathcal{X}') + 2B \cdot \left(1 - Q_{x_S}(\mathcal{X}')\right),$$

*where $Q_{x_S}(\mathcal{X}')$ is the probability of imputed samples lying in $\mathcal{X}'$ based on the distribution $q(\boldsymbol{x}_{\bar{S}})$:*

$$Q_{x_S}(\mathcal{X}') \equiv \mathbb{E}_{q(\boldsymbol{x}_{\bar{S}})}\left[\mathbb{I}\left\{(x_S, \boldsymbol{x}_{\bar{S}}) \in \mathcal{X}'\right\}\right].$$

*Proof.* The main idea here is that when integrating over the difference in function outputs, we can separate the integral over $(x_S, \boldsymbol{x}_{\bar{S}})$ values falling in the domains $\mathcal{X}'$ and $\mathbb{R}^d \setminus \mathcal{X}'$. Because we have a single value for the observed variables $x_S$, the distribution $q(\boldsymbol{x}_{\bar{S}})$ is identical for both terms regardless of the removal technique.

$$
\begin{aligned}
|f(x_S) - f'(x_S)| &= \left| \int f(x_S, x_{\bar{S}})q(x_{\bar{S}})dx_{\bar{S}} - \int f'(x_S, x_{\bar{S}})q(x_{\bar{S}})dx_{\bar{S}} \right| \\
&\leq \int \left| f(x_S, x_{\bar{S}}) - f'(x_S, x_{\bar{S}}) \right| \cdot q(x_{\bar{S}})dx_{\bar{S}} \\
&= \int \left| f(x_S, x_{\bar{S}}) - f'(x_S, x_{\bar{S}}) \right| \cdot \mathbb{I}\left\{(x_S, \boldsymbol{x}_{\bar{S}}) \in \mathcal{X}'\right\} \cdot q(x_{\bar{S}})dx_{\bar{S}} \\
&\quad + \int \left| f(x_S, x_{\bar{S}}) - f'(x_S, x_{\bar{S}}) \right| \cdot \mathbb{I}\left\{(x_S, \boldsymbol{x}_{\bar{S}}) \notin \mathcal{X}'\right\} \cdot q(x_{\bar{S}})dx_{\bar{S}} \\
&\leq \|f - f'\|_{\infty, \mathcal{X}'} \int \mathbb{I}\left\{(x_S, \boldsymbol{x}_{\bar{S}}) \in \mathcal{X}'\right\} \cdot q(x_{\bar{S}})dx_{\bar{S}} \\
&\quad + 2B \cdot \int \mathbb{I}\left\{(x_S, \boldsymbol{x}_{\bar{S}}) \notin \mathcal{X}'\right\} \cdot q(x_{\bar{S}})dx_{\bar{S}} \\
&= \|f - f'\|_{\infty, \mathcal{X}'} \cdot Q_{x_S}(\mathcal{X}') + 2B \cdot \left(1 - Q_{x_S}(\mathcal{X}')\right)
\end{aligned}
$$

$\square$

**Lemma 5.** *When removing features using the **conditional** approach, the prediction functions $f(x_S), f'(x_S)$ for two models $f, f'$ and any feature set $x_S$ satisfy*

$$|f(x_S) - f'(x_S)| \leq \|f - f'\|_{\infty, \mathcal{X}}.$$

*Proof.* The result follows directly from Lemma 4 when we set $\mathcal{X}' = \mathcal{X}$. Note that when we remove features using the conditional distribution, we have $Q_{x_S}(\mathcal{X}) = 1$ for all $x_S$ lying on the data manifold, or where there exists $x_{\bar{S}}$ such that $p(x_S, x_{\bar{S}}) > 0$. $\square$

**Lemma 6.** *When removing features using the **baseline** or **marginal** approach, the prediction functions $f(x_S), f'(x_S)$ for two models $f, f'$ and any feature set $x_S$ satisfy*

$$|f(x_S) - f'(x_S)| \leq \|f - f'\|_{\infty}.$$

*Proof.* The result follows directly from Lemma 4 when we set $\mathcal{X}' = \mathbb{R}^d$, because we have $Q_{x_S}(\mathbb{R}^d) = 1$ for all $x_S \in \mathbb{R}^{|S|}$. $\square$

# D  Proofs: summary technique robustness

We now re-state and prove our results from Section 3.3.

**Proposition 1.** *The attributions for each method can be calculated by applying a linear operator* $A \in \mathbb{R}^{d \times 2^d}$ *to a vector* $v \in \mathbb{R}^{2^d}$ *representing the predictions with each feature set, or*

$$\phi(f, x) = Av,$$

*where the linear operator A for each method is listed in Table 3, and v is defined as* $v_S = f(x_S)$ *for each* $S \subseteq [d]$ *based on the chosen feature removal technique.*

*Proof.* This result follows directly from the definition of each method. Following the approach of Covert et al. [17], we disentangle between each method's feature removal implementation and how it generates attribution scores. Assuming a fixed input $x$ where $f(x_S)$ denotes the prediction with a feature set, each method's attributions are defined as follows.

- **Occlusion** [67]: this method simply compares the prediction given all features and the prediction with a single missing feature. The attribution score $\phi_i(f, x)$ is defined as

$$\phi_i(f, x) = f(x) - f(x_{[d] \setminus \{i\}}).$$

  We refer to this summary technique as "leave-one-out" in Table 3.

- **Shapley value** [45, 54]: this method calculates the impact of removing a single feature, and then averages this over all possible preceding subsets. The attribution score $\phi_i(f, x)$ is defined as

$$\phi_i(f, x) = \frac{1}{d} \sum_{S \subseteq [d] \setminus \{i\}} \binom{d-1}{|S|}^{-1} \Big( f(x_{S \cup \{i\}}) - f(x_S) \Big).$$

- **Banzhaf value** [9, 13]: this method is similar to the Shapley value, but it applies a uniform weighting when averaging over all preceding subsets. The attribution score $\phi_i(f, x)$ is defined as

$$\phi_i(f, x) = \frac{1}{2^{d-1}} \sum_{S \subseteq [d] \setminus \{i\}} \Big( f(x_{S \cup \{i\}}) - f(x_S) \Big).$$

- **RISE** [51]: this method is similar to the Banzhaf value, but it does not subtract the value for subsets where a feature is not included. The attribution score $\phi_i(f, x)$ is defined as

$$\phi_i(f, x) = \frac{1}{2^{d-1}} \sum_{S \subseteq [d] \setminus \{i\}} f(x_{S \cup \{i\}}).$$

  We refer to this summary technique as "mean when included" in Table 3

- **LIME** [53]: this method is more flexible than the others and generally involves solving a regression problem that treats the features as inputs and the predictions as labels. LIME can be viewed as a removal-based explanation when the features are represented in a binary fashion, indicating whether they are provided to the model or not [17]. There is also a choice of weighting kernel $w(S) \geq 0$, of a regularization term, and of whether to fit an intercept term.

  We assume that the regularization is omitted and that we fit an intercept term. Denoting the model's coefficients as $(\beta_0, \ldots, \beta_d)$, the problem is the following:

$$\min_{\beta_0, \ldots, \beta_d} \sum_{S \subseteq [d]} w(S) \Big( \beta_0 + \sum_{i \in S} \beta_i - f(x_S) \Big)^2.$$

  We can rewrite this problem in a simpler fashion to derive its solution. First, we let $W \in [0, 1]^{2^d \times 2^d}$ be a diagonal matrix containing the weights $w(S)$ for each subset. Next, we let

the predictions be represented by a vector $v \in \mathbb{R}^{2^d}$ where $v_S = f(x_S)$, and we let the matrix $Z \in \{0,1\}^{2^d \times d}$ enumerate all subsets in a binary fashion. Finally, to accommodate the intercept term, we let $\beta = (\beta_0, \ldots, \beta_d) \in \mathbb{R}^{d+1}$ represent the parameters, and we prepend a columns of ones to $Z$, resulting in $Z' \in \{0,1\}^{2^d \times (d+1)}$. The problem then becomes:

$$\min_{\beta} (Z'\beta - v)^\top W (Z'\beta - v') = \left\| Z'\beta - v \right\|_W^2.$$

Assuming that $Z'^\top W Z'$ is invertible, which is guaranteed when $w(S) > 0$ for all $S \subseteq [d]$, this problem has a closed-form solution:

$$\beta^* = (Z'^\top W Z')^{-1} Z'^\top W v.$$

LIME's attribution scores are given by $\phi_i(f, x) = \beta_i^*$, for $i = 1, \ldots, d$, where we discard the intercept term. We can therefore conclude that each attribution is a linear function of $v$, with coefficients given by all but the first row of $(Z'^\top W Z')^{-1} Z'^\top W$. If we denote all but the first row of $(Z'^\top W Z')^{-1}$ as $(Z'^\top W Z')^{-1}_{[1:]}$, then we have $A = (Z'^\top W Z')^{-1}_{[1:]} Z'^\top W$.

The formula for the attributions remains roughly the same if we do not fit an intercept term: if the intercept is zero, we have $A = (Z^\top W Z)^{-1} Z^\top W$. We can similarly allow for ridge regularization: given a penalty parameter $\lambda > 0$, we then have $A = (Z^\top W Z + \lambda I_{2^d})^{-1} Z^\top W$.

These coefficients depend on the specific choice of weighting kernel $w(S)$, and previous work has shown that leave-one-out, Shapley and Banzhaf values all correspond to specific weighting kernels [17]. Our analysis of those methods therefore pertains to LIME, at least in certain special cases. However, LIME often uses a heuristic exponential kernel in practice, which is more difficult to characterize analytically. We therefore omit the specific entries and norms for LIME in Table 3, but show norms for the exponential kernel in our computed results.

We refer to this summary technique as "weighted least squares" in Table 3.

We can therefore see that in each method, $\phi_i(f, x)$ is a linear function of the predictions for each $S \subseteq [d]$. Across all methods, we can let the predictions be represented by a vector $v \in \mathbb{R}^{2^d}$ and the coefficients by a matrix $A \in \mathbb{R}^{d \times 2^d}$. $\qquad \square$

**Lemma 7.** *The difference in attributions given the same summary technique $A$ and different model outputs $v, v'$ can be bounded as*

$$\|Av - Av'\|_2 \leq \|A\|_2 \cdot \|v - v'\|_2 \qquad or \qquad \|Av - Av'\|_2 \leq \|A\|_{1,\infty} \cdot \|v - v'\|_\infty,$$

*where $\|A\|_2$ is the spectral norm, and the operator norm $\|A\|_{1,\infty}$ is the square root of the sum of squared row 1-norms, with values for each $A$ given in Table 3.*

*Proof.* We first derive the two matrix inequalities. The first inequality follows from the definition of the spectral norm, which is defined as the maximum singular value:

$$\|A\|_2 \equiv \sup_{u \neq 0} \frac{\|Au\|_2}{\|u\|_2} = \sigma_{\max}(A) = \sqrt{\lambda_{\max}(AA^\top)}.$$

We prove the second bound using Holder's inequality as follows, where $A_i$ denotes the $i$th row:

$$\|Au\|_2 = \sqrt{\sum_{i=1}^d (A_i^\top u)^2} \leq \sqrt{\sum_{i=1}^d \|A_i\|_1^2 \cdot \|u\|_\infty^2} = \|u\|_\infty \cdot \sqrt{\sum_{i=1}^d \|A_i\|_1^2} = \|u\|_\infty \cdot \|A\|_{1,\infty}.$$

In the special case where all of $A$'s row 1-norms are equal, or where we have $\|A_i\|_1 = \|A_j\|_1$ for all $i, j \in [d]$, we can use the simplified expression $\|A\|_{1,\infty} = \|A_1\|_1 \cdot \sqrt{d}$.

Next, we derive the specific norm values for each linear operator in Table 3.

Beginning with the operator norm $\|A\|_{1,\infty}$, we remark that several methods have equal 1-norms across all rows: following their interpretation as *semivalues* [46], leave-one-out, Shapley and Banzhaf all have $\|A_i\|_1 = 2$ for all $i \in [d]$. Similarly, RISE has $\|A_i\|_1 = 1$ for all $i \in [d]$. This yields the norms $\|A\|_{1,\infty} = 2\sqrt{d}$ for leave-one-out, Banzhaf and Shapley, and $\|A\|_{1,\infty} = \sqrt{d}$ for RISE.

For the spectral norm, we consider each method separately. We find that for all methods, it is simplest to calculate $AA^\top$ and then $\lambda_{\max}(AA^\top)$ to find the maximum singular value $\sigma_{\max}(A)$. Note that the entries of the matrix are determined by the inner product $(AA^\top)_{ij} = A_i^\top A_j$.

For leave-one-out, we derive $AA^\top$ as follows:

$$(AA^\top)_{ij} = \begin{cases} 2 & \text{if } i = j \\ 1 & \text{otherwise.} \end{cases}$$

The matrix $AA^\top$ is therefore the sum of a rank-one component and an identity term, and we have $AA^\top = \mathbf{1}_d\mathbf{1}_d + I_d$. Recognizing that $\mathbf{1}_d$ is the leading eigenvector, we get the following result:

$$\sigma_{\max}(A) = \sqrt{\lambda_{\max}(AA^\top)} = \sqrt{\frac{(\mathbf{1}_d\mathbf{1}_d + I_d)\mathbf{1}_d}{\|\mathbf{1}_d\|_2}} = \sqrt{\frac{\mathbf{1}_d d + \mathbf{1}_d}{\|\mathbf{1}_d\|_2}} = \sqrt{d+1}.$$

For the Banzhaf value, we derive $AA^\top$ as follows:

$$(AA^\top)_{ij} = \begin{cases} 1/2^{2d-2} & \text{if } i = j \\ 0 & \text{otherwise.} \end{cases}$$

The max eigenvalue is therefore $\lambda_{\max}(AA^\top) = 1/2^{d-2}$, which yields $\|A\|_2 = 1/2^{d/2-1}$.

For the mean when included technique used by RISE, we derive $AA^\top$ as follows:

$$(AA^\top)_{ij} = \begin{cases} 1/2^{d-1} & \text{if } i = j \\ 1/2^d & \text{otherwise.} \end{cases}$$

The matrix $AA^\top$ is therefore the sum of a rank-one component and an identity term, or we have $AA^\top = \mathbf{1}_d\mathbf{1}_d/2^d + I_d/2^d$. Recognizing that $\mathbf{1}_d$ is the leading eigenvector, we get the following result:

$$\sigma_{\max}(A) = \sqrt{\lambda_{\max}(AA^\top)} = \sqrt{\frac{(\mathbf{1}_d\mathbf{1}_d/2^d + I_d/2^d)\mathbf{1}_d}{\|\mathbf{1}_d\|_2}} = \sqrt{\frac{\mathbf{1}_d d/2^d + \mathbf{1}_d/2^d}{\|\mathbf{1}_d\|_2}} = \sqrt{\frac{d+1}{2^d}}.$$

For the Shapley value, we derive $AA^\top$ by first considering the diagonal entries:

$$(AA^\top)_{ii} = A_i^\top A_i = \sum_{S:i\in S} \left( \frac{(|S|-1)!(d-|S|)!}{d!} \right)^2 + \sum_{S:i\notin S} \left( \frac{|S|!(d-|S|-1)!}{d!} \right)^2$$

$$= \sum_{k=0}^{d} \left[ \binom{d-1}{k-1} \left( \frac{(k-1)!(d-k)!}{d!} \right)^2 + \binom{d-1}{k} \left( \frac{k!(d-k-1)!}{d!} \right)^2 \right]$$

$$= \frac{1}{d} \sum_{k=1}^{d} \frac{(k-1)!(d-k)!}{d!} + \frac{1}{d} \sum_{k=0}^{d-1} \frac{k!(d-k-1)!}{d!}$$

$$= \sum_{k=1}^{d-1} \frac{(k-1)!(d-k-1)!}{d!} + \underbrace{\frac{2}{d^2}}_{k=0,k=d}$$

$$= \frac{1}{d(d-1)} \sum_{k=1}^{d-1} \binom{d-2}{k-1}^{-1} + \frac{2}{d^2}$$

We next consider the off-diagonal entries:

$$(AA^\top)_{ij} = A_i^\top A_j = \sum_{S:i\in S, j\in S} \frac{(|S|-1)!(d-|S|)!}{d!} \frac{(|S|-1)!(d-|S|)!}{d!}$$

$$- \sum_{S:i\in S, j\notin S} \frac{(|S|-1)!(d-|S|)!}{d!} \frac{|S|!(d-|S|-1)!}{d!}$$

$$- \sum_{S:i\notin S, j\in S} \frac{|S|!(d-|S|-1)!}{d!} \frac{(|S|-1)!(d-|S|)!}{d!}$$

$$+ \sum_{S:i\notin S, j\notin S} \frac{|S|!(d-|S|-1)!}{d!} \frac{|S|!(d-|S|-1)!}{d!}$$

$$= \sum_{k=0}^{d} \binom{d-2}{k-2} \frac{(k-1)!(d-k)!}{d!} \frac{(k-1)!(d-k)!}{d!}$$

$$- 2 \sum_{k=0}^{d} \binom{d-2}{k-1} \frac{(k-1)!(d-k)!}{d!} \frac{k!(d-k-1)!}{d!}$$

$$+ \sum_{k=0}^{d} \binom{d-2}{k} \frac{k!(d-k-1)!}{d!} \frac{k!(d-k-1)!}{d!}$$

$$= \sum_{k=2}^{d} \frac{(k-1)}{d(d-1)} \frac{(k-1)!(d-k)!}{d!}$$

$$- 2 \sum_{k=1}^{d-1} \frac{1}{d(d-1)} \frac{k!(d-k)!}{d!}$$

$$+ \sum_{k=0}^{d-2} \frac{(d-k-1)}{d(d-1)} \frac{k!(d-k-1)!}{d!}$$

We first focus on the case where $d > 2$, and we later return to the case where $d = 2$. We can group terms that share the same $k$ values to simplify the above:

$$A_i^\top A_j = \frac{1}{d(d-1)} \left[ -\sum_{k=2}^{d-2} \frac{(k-1)!(d-k-1)!}{(d-1)!} + \underbrace{\frac{2(d-1)}{d}}_{k=0,k=d} - \underbrace{\frac{2}{d-1}}_{k=1,k=d-1} \right]$$

$$= -\frac{1}{d(d-1)} \sum_{k=2}^{d-2} \frac{(k-1)!(d-k-1)!}{(d-1)!} + \frac{2}{d^2} - \frac{2}{d(d-1)^2}$$

$$= -\frac{1}{d(d-1)^2} \sum_{k=2}^{d-2} \binom{d-2}{k-1}^{-1} + \frac{2}{d^2} - \frac{2}{d(d-1)^2}.$$

By convention, we let the summation above evaluate to zero for $d = 3$. We therefore have the following result for $AA^\top$:

$$(AA^\top)_{ij} = \begin{cases} \frac{1}{d(d-1)} \sum_{k=1}^{d-1} \binom{d-2}{k-1}^{-1} + \frac{2}{d^2} & \text{if } i = j \\ -\frac{1}{d(d-1)^2} \sum_{k=2}^{d-2} \binom{d-2}{k-1}^{-1} + \frac{2}{d^2} - \frac{2}{d(d-1)^2} & \text{otherwise.} \end{cases}$$

From this, we can see that $AA^\top$ is the sum of a rank-one component and an identity term, or that we have $AA^\top = \mathbf{1}_d \mathbf{1}_d b + I_d(c-b)$ with values for $b = (AA^\top)_{ij}$ and $c = (AA^\top)_{ii}$ given above. This matrix has two eigenvalues, $c - b$ and $db + (c-b)$, and we must consider which of the two is larger. Under our assumption of $d > 2$, we can observe that $b > 0$:

$$b = (AA^\top)_{ij} = \frac{2}{d^2} - \frac{1}{d(d-1)^2} \sum_{k=2}^{d-2} \binom{d-2}{k-1}^{-1} - \frac{2}{d(d-1)^2}$$

$$\geq \frac{2}{d^2} - \frac{1}{d(d-1)^2} \sum_{k=2}^{d-2} \binom{d-2}{1}^{-1} - \frac{2}{d(d-1)^2}$$

$$= \frac{2}{d^2} - \frac{d-3}{d(d-1)^2(d-2)} - \frac{2}{d(d-1)^2}$$

$$= \frac{2(d-1)^2(d-2) - d(d-3) - 2d(d-2)}{d^2(d-1)^2(d-2)}$$

It can be verified that the numerator of the above fraction is positive for $d > 2$. This implies that $db + (c-b)$ is the larger eigenvalue. We can therefore calculate $\lambda_{\max}(AA^\top)$ as follows:

$$\lambda_{\max}(AA^\top) = db + (c-b) = c + (d-1)b$$

$$= \frac{2}{d^2} + \underbrace{\frac{2}{d(d-1)}}_{k=1,k=d-1} + \frac{2(d-1)}{d^2} - \frac{2}{d(d-1)}$$

$$= \frac{2(d-1) + 2(d-1)^2}{d^2(d-1)}$$

$$= \frac{2}{d}$$

On the other hand, when we have $d = 2$ we have the following entries for $AA^\top$:

$$(AA^\top)_{ij} = \begin{cases} 1 & \text{if } i = j \\ 0 & \text{otherwise.} \end{cases}$$

This yields $\lambda_{\max}(AA^\top) = 1$, which coincides with the formula $\lambda_{\max}(AA^\top) = 2/d$ derived above. Finally, this let us conclude that the spectral norm for the Shapley value is the following:

$$\|A\|_2 = \sqrt{\lambda_{\max}(AA^\top)} = \sqrt{\frac{2}{d}}.$$

$\square$

**Lemma 8.** *When the linear operator $A$ satisfies the **boundedness** property, we have $\|A\|_{1,\infty} = 2\sqrt{d}$.*

*Proof.* Following Theorem 1 in Monderer and Samet [46], solution concepts satisfying linearity and the boundedness property (also known as the Milnor axiom) are *probabilistic values*. This means that each attribution is the average of a feature's marginal contributions, or that for each feature $i \in [d]$ there exists a distribution $p_i(S)$ with support on the power set of $[d] \setminus \{i\}$, such that the attributions are given by

$$\phi_i(f, x) = \mathbb{E}_{p_i(S)}\big[f(x_{S \cup \{i\}}) - f(x_S)\big] = \sum_{S \subseteq [d] \setminus \{i\}} p_i(S)\big(f(x_{S \cup \{i\}}) - f(x_S)\big). \tag{D.0.1}$$

The corresponding matrix $A \in \mathbb{R}^{d \times 2^d}$ therefore has rows $A_i$ with entries $A_{iS} \geq 0$ when $i \in S$ and $A_{iS} \leq 0$ when $i \notin S$, where the positive and negative entries sum to 1 and -1 respectively. We can therefore conclude that each row satisfies $\|A_i\|_1 = 2$, and that $\|A\|_{1,\infty} = 2\sqrt{d}$.

$\square$

**Lemma 9.** *When the linear operator $A$ satisfies the **boundedness** and **symmetry** properties, the spectral norm is bounded as follows,*

$$\frac{1}{2^{d/2-1}} \leq \|A\|_2 \leq \sqrt{d+1},$$

*with $1/2^{d/2-1}$ achieved by the Banzhaf value and $\sqrt{d+1}$ achieved by leave-one-out.*

*Proof.* Following Monderer and Samet [46], solution concepts satisfying linearity, boundedness and symmetry are *semivalues*. This means that they are *probabilistic values* for which the probability distribution $p_i(S)$ depends only on the cardinality $|S|$ (see the definition of probabilistic values above). As a result, we can parameterize the summary technique by how much weight it places on each cardinality.

Let $\alpha \in \mathbb{R}^d$ denote a probability distribution over the cardinalities $k = 1, \ldots, d$. We can write the attributions generated by any semivalue as follows:

$$\phi_i(f, x) = \sum_{k=1}^d \alpha_k \sum_{\substack{S \subseteq [d] \setminus \{i\} \\ |S| = k-1}} \binom{d-1}{k-1}^{-1} \big(f(x_{S \cup \{i\}}) - f(x_S)\big).$$

Note that this resembles eq. (D.0.1), only with coefficients that are shared between all subsets with the same cardinality. Due to each attribution being a linear combination of the predictions, we can associate a matrix $A \in \mathbb{R}^{d \times 2^d}$ with each semivalue, similar to Proposition 1. Furthermore, we can let $A$ be a linear combination of basis elements corresponding to each cardinality. We define the basis matrix $A^{(k)} \in \mathbb{R}^{d \times 2^d}$ for each cardinality $k = 1, \ldots, d$ as follows:

$$A_{iS}^{(k)} = \begin{cases} \binom{d-1}{|S|-1}^{-1} & \text{if } i \in S \text{ and } |S| = k \\ -\binom{d-1}{|S|}^{-1} & \text{if } i \notin S \text{ and } |S| = k-1 \\ 0 & \text{otherwise.} \end{cases}$$

With this, we can write the matrix associated with any semivalue as a function of the cardinality weights $\alpha$:

$$A(\alpha) = \sum_{k=1}^{d} \alpha_k A^{(k)}.$$

An important insight is that by composing a linear function with the convex spectral norm function [11], we can see that $\|A(\alpha)\|_2$ is a convex function of $\alpha$. This lets us find both the upper and lower bound of $\|A(\alpha)\|_2$ subject to $\alpha$ being a valid probability distribution.

For the lower bound, we refer readers to Theorem C.3 from Wang and Jia [64], who prove that the Banzhaf value achieves the minimum with $\alpha^*$ such that $\alpha_k^* = \binom{d-1}{k-1}/2^{d-1}$ and spectral norm $\|A(\alpha^*)\|_2 = 1/2^{d/2-1}$. Notice that this is a valid probability distribution because we have $\sum_{k=1}^{d} \binom{d-1}{k-1} = 2^{d-1}$. Also note that the proof in [64] parameterizes $\alpha$ slightly differently.

For the upper bound, we can leverage convexity to make the following argument. The fact that spectral norm is convex with $\alpha$ representing probabilities lets us write the following for any $\alpha$ value:

$$\sigma_{\max}\left(A(\alpha)\right) = \sigma_{\max}\left(\sum_{k=1}^{d} \alpha_k \cdot A^{(k)}\right) \leq \sum_{k=1}^{d} \alpha_k \cdot \sigma_{\max}\left(A^{(k)}\right) \leq \max_k \left\{\sigma_{\max}\left(A^{(k)}\right)\right\}.$$

Finding the upper bound therefore reduces to finding the basis element with the largest spectral norm. For each basis element $A^{(k)}$, we can derive the spectral norm by first finding the maximum eigenvalue of $A^{(k)}A^{(k)^\top} \in \mathbb{R}^{d \times d}$. In deriving this matrix, we first consider the diagonal entries:

$$
\begin{aligned}
\left(A^{(k)}A^{(k)^\top}\right)_{ii} &= \sum_{S:i\in S,|S|=k} \binom{d-1}{|S|-1}^{-2} + \sum_{S:i\notin S,|S|=k-1} \binom{d-1}{|S|}^{-2} \\
&= \binom{d-1}{k-1}\binom{d-1}{k-1}^{-2} + \binom{d-1}{k-1}\binom{d-1}{k-1}^{-2} \\
&= 2\binom{d-1}{k-1}^{-1}.
\end{aligned}
$$

Next, we consider the off-diagonal entries:

$$
\begin{aligned}
\left(A^{(k)}A^{(k)^\top}\right)_{ij} &= \sum_{S:i\in S,j\in S,|S|=k} \binom{d-1}{|S|-1}^{-2} + \sum_{S:i\notin S,j\notin S,|S|=k-1} \binom{d-1}{|S|}^{-2} \\
&= \binom{d-2}{k-2}\binom{d-1}{k-1}^{-2} + \binom{d-2}{k-1}\binom{d-1}{k-1}^{-2} \\
&= \frac{k-1}{d-1}\binom{d-1}{k-1}^{-1} + \frac{d-k}{d-1}\binom{d-1}{k-1}^{-1} \\
&= \binom{d-1}{k-1}^{-1}.
\end{aligned}
$$

To summarize, the entries of this matrix are the following:

$$
\left(A^{(k)}A^{(k)^\top}\right)_{ij} = \begin{cases} 2\binom{d-1}{k-1}^{-1} & \text{if } i = j \\ \binom{d-1}{k-1}^{-1} & \text{otherwise.} \end{cases}
$$

We therefore see that $A^{(k)}A^{(k)\top}$ is equal to a rank-one matrix plus an identity matrix, or $A^{(k)}A^{(k)\top} = b\mathbf{1}_d\mathbf{1}_d^\top + (c - b)I_d$, with values for $b = \left(A^{(k)}A^{(k)\top}\right)_{ij}$ and $c = \left(A^{(k)}A^{(k)\top}\right)_{ii}$ given above. Recognizing that $b > 0$ and therefore that $\mathbf{1}_d$ is the leading eigenvector, we have the following expression for the maximum eigenvalue:

$$
\begin{aligned}
\lambda_{\max}\left(A^{(k)}A^{(k)\top}\right) &= c + (d-1)b \\
&= \left(A^{(k)}A^{(k)\top}\right)_{ii} + (d-1)\cdot\left(A^{(k)}A^{(k)\top}\right)_{ij} \\
&= (d+1)\binom{d-1}{k-1}^{-1}.
\end{aligned}
$$

Finally, we must consider how to maximize this as a function of $k$. Setting $k = 1$ or $k = d$ achieves the same value of $d + 1$, which yields the following spectral norm:

$$
\|A^{(d)}\|_2 = \sqrt{\lambda_{\max}\left(A^{(d)}A^{(d)\top}\right)} = \sqrt{d+1}.
$$

The case with $k = d$ corresponds to leave-one-out (see Lemma 7), and $k = 1$ corresponds to an analogous *leave-one-in* approach that has been discussed in prior work [15, 17]. $\qquad\square$

**Lemma 10.** *When the linear operator $A$ satisfies the **boundedness** and **efficiency** properties, the spectral norm is bounded as follows,*

$$
\sqrt{\frac{2}{d}} \le \|A\|_2 \le \sqrt{2 + 2\cdot\cos\left(\frac{\pi}{d+1}\right)},
$$

*with $\sqrt{2/d}$ achieved by the Shapley value.*

*Proof.* Following Monderer and Samet [46], solution concepts satisfying linearity, boundedness and efficiency are *random-order values* (or *quasivalues*). This means that the attributions are the average marginal contributions across a distribution of feature orderings. Reasoning about such orderings requires new notation, so we use $\pi$ to denote an ordering over $[d]$, and $\pi(i)$ to denote the position of index $i$ within the ordering. We also let $\mathrm{Pre}(\pi, i)$ denote the set of all elements preceding $i$ in the ordering, or $\mathrm{Pre}(\pi, i) \equiv \{j : \pi(j) < \pi(i)\}$. Similarly, we let $\mathrm{Pre}(\pi, i, j)$ be an indicator of whether $j$ directly precedes $i$ in the ordering, or $\mathrm{Pre}(\pi, i, j) \equiv \mathbb{I}\{\pi(j) = \pi(i) - 1\}$. Finally, we use $\Pi$ to denote the set of all orderings, where $|\Pi| = d!$.

Now, given a distribution over orderings $p(\pi)$, we can write the attributions for any random-order value as follows:

$$
\begin{aligned}
\phi_i(f, x) &= \sum_{\pi\in\Pi} p(\pi)\Big(f(x_{\mathrm{Pre}(\pi,i)\cup\{i\}}) - f(x_{\mathrm{Pre}(\pi,i)})\Big) \\
&= \sum_{S\subseteq[d]\setminus\{i\}}\Big(\sum_{\pi:\mathrm{Pre}(\pi,i)=S} p(\pi)\Big)\Big(f(x_{S\cup\{i\}}) - f(x_S)\Big).
\end{aligned}
$$

Based on this formulation, we can associate a matrix $A \in \mathbb{R}^{d\times 2^d}$ with each random-order value based on the coefficients applied to each prediction. The entries are defined as follows:

$$
A_{iS} = \begin{cases} \sum_{\pi:\mathrm{Pre}(\pi,i)=S\setminus\{i\}} p(\pi) & \text{if } i \in S \\ -\sum_{\pi:\mathrm{Pre}(\pi,i)=S} p(\pi) & \text{otherwise.} \end{cases}
$$

Furthermore, we can view $A$ as a linear combination of basis elements $A^{(\pi)} \in \mathbb{R}^{d\times 2^d}$ corresponding to each individual ordering. We define the basis matrix $A^{(\pi)}$ associated with each $\pi \in \Pi$ as follows:

$$A_{iS}^{(\pi)} = \begin{cases} 1 & \text{if } \mathrm{Pre}(\pi, i) = S \setminus \{i\} \\ -1 & \text{if } \mathrm{Pre}(\pi, i) = S \\ 0 & \text{otherwise.} \end{cases}$$

With this, we can write the linear operator $A$ corresponding to a distribution $p(\pi)$ as the following linear combination:

$$A(p) = \sum_{\pi \in \Pi} p(\pi) \cdot A^{(\pi)}.$$

By composing a linear operation with the convex spectral norm function [11], we can see that $\|A(p)\|_2$ is a convex function of the probabilities $p(\pi)$ assigned to each ordering. This will allow us to derive a form for both the upper and lower bound on $\|A(p)\|_2$.

Beginning with the upper bound, we can use convexity to write the following:

$$\sigma_{\max}\big(A(p)\big) = \sigma_{\max}\Big( \sum_{\pi \in \Pi} p(\pi) \cdot A^{(\pi)} \Big) \leq \sum_{\pi \in \Pi} p(\pi) \cdot \sigma_{\max}\big(A^{(\pi)}\big) \leq \max_{\pi \in \Pi} \big\{ \sigma_{\max}\big(A^{(\pi)}\big) \big\}.$$

Finding the upper bound therefore reduces to finding the basis element with the largest spectral norm. For an arbitrary ordering $\pi$, we can derive the spectral norm via the matrix $A^{(\pi)} A^{(\pi)^\top}$, whose entries are given by:

$$\Big( A^{(\pi)} A^{(\pi)^\top} \Big)_{ij} = \begin{cases} 2 & \text{if } i = j \\ -1 & \text{if } i \neq j \text{ and } \mathrm{Pre}(\pi, i, j) \\ -1 & \text{if } i \neq j \text{ and } \mathrm{Pre}(\pi, j, i) \\ 0 & \text{otherwise.} \end{cases}$$

Within these matrices, the diagonal entries are always equal to 2, most rows contain two $-1$'s, and two rows (corresponding to the first and last element in the ordering $\pi$) contain a single $-1$. We can see that the matrices $A^{(\pi)} A^{(\pi)^\top}$ are identical up to a permutation of the rows and columns, or that they are similar matrices. They therefore share the same eigenvalues.

Without loss of generality, we choose to focus on the ordering $\pi'$ that places the indices in their original order, or $\pi'(i) = i$ for $i \in [d]$. The corresponding matrix $A^{(\pi')} A^{(\pi')^\top}$ has the following form:

$$A^{(\pi')} A^{(\pi')^\top} = \begin{pmatrix} 2 & -1 & 0 & 0 & \cdots & 0 \\ -1 & 2 & -1 & 0 & \cdots & 0 \\ 0 & -1 & 2 & -1 & \cdots & 0 \\ 0 & 0 & \ddots & \ddots & \ddots & \vdots \\ \vdots & \vdots & \ddots & -1 & 2 & -1 \\ 0 & \cdots & \cdots & 0 & -1 & 2 \end{pmatrix}$$

This is a tridiagonal Toeplitz matrix, a type of matrix whose eigenvalues have a known closed-form solution [49]. The maximum eigenvalue is the following:

$$\lambda_{\max}\Big( A^{(\pi')} A^{(\pi')^\top} \Big) = 2 + 2\sqrt{(-1) \cdot (-1)} \cos\Big( \frac{\pi}{d+1} \Big)$$

$$= 2 + 2 \cdot \cos\Big( \frac{\pi}{d+1} \Big).$$

We therefore have the following upper bound on the spectral norm for a random-order value:

$$\|A\|_2 \leq \sqrt{2 + 2 \cdot \cos\left(\frac{\pi}{d+1}\right)}.$$

This value tends asymptotically towards 2, and it is achieved by any random-order value that places all its probability mass on a single ordering. Such a *single-order value* approach has not been discussed in the literature, but it is a special case of a proposed variant of the Shapley value [26].

Next, we consider the lower bound. We now exploit the convexity of $\|A(p)\|_2$ to argue that a specific distribution achieves a lower spectral norm than any other distribution.

Consider an arbitrary distribution $p(\pi)$ over orderings, with the associated linear operator $A(p)$ and spectral norm $\|A(p)\|_2$. Our proof technique is to find other distributions that result in different linear operators $A$ that share the same spectral norm. For any ordering $\pi' \in \Pi$, we can imagine permuting the distribution $p(\pi)$ in the following sense: we generate a new distribution $p_{\pi'}(\pi)$ such that $p_{\pi'}(\pi) = p(\pi(\pi'))$, where $\pi(\pi')$ is a modified ordering with $\pi(\pi')(i) = \pi(\pi'(i))$. That is, the probability mass is reassigned based on a permutation of the indices.

Crucially, the linear operator $A(p_{\pi'})$ is different from $A(p)$ but shares the same spectral norm. This is because $A(p_{\pi'})A(p_{\pi'})^\top$ is similar to $A(p)A(p)^\top$, or they are equal up to a permutation of the rows and columns. We therefore have $\|A(p_{\pi'})\|_2 = \|A(p)\|_2$, and due to convexity we have

$$\left\|A\big(p/2 + p_{\pi'}/2\big)\right\|_2 \leq \frac{1}{2}\|A(p_{\pi'})\|_2 + \frac{1}{2}\|A(p)\|_2 = \|A(p)\|_2.$$

This does not imply that $p/2 + p_{\pi'}/2$ is a minimizer, but we can now generalize this technique to identify a lower bound. Rather than generating a single permuted distribution, we propose generating all possible permuted distributions. That is, we consider the set of all $d!$ permutations $\pi' \in \Pi$ and generate the corresponding distributions $\{p_{\pi'} : \pi' \in \Pi\}$. Notice that when we take the mean of these distributions, the result is identical regardless of the original distributions $p(\pi)$, and that it assigns uniform probability mass to all orderings, because we have:

$$\frac{1}{d!}\sum_{\pi' \in \Pi} p_{\pi'}(\pi) = \frac{1}{d!}\sum_{\pi' \in \Pi} p(\pi') = \frac{1}{d!}.$$

We can then exploit convexity to write the following inequality:

$$\left\|A\big(\sum_{\pi' \in \Pi} p_{\pi'}/d!\big)\right\|_2 \leq \|A(p)\|_2.$$

We can make this argument for any original distribution $p$, so this provides a global lower bound. The random-order value with uniform weighting is the Shapley value, whose spectral norm was derived in Lemma 7. We therefore conclude with the following lower bound for random-order values, which is achieved by the Shapley value:

$$\|A\|_2 \geq \sqrt{\frac{2}{d}}.$$

$\square$

**Lemma 11.** *When the linear operator $A$ satisfies the **boundedness**, **symmetry** and **efficiency** properties, the spectral norm is $\|A\|_2 = \sqrt{2/d}$.*

*Proof.* The result follows from the spectral norm result in Lemma 7, and the fact that the Shapley value is the only solution concept to satisfy linearity, boundedness, symmetry and efficiency [46]. $\square$

# E  Proofs: main results

We now re-state and prove our results from Section 4.

**Theorem 1.** *The robustness of removal-based explanations to input perturbations is given by the following meta-formula,*

$$\|\phi(f, x) - \phi(f, x')\|_2 \leq g(\text{removal}) \cdot h(\text{summary}) \cdot \|x - x'\|_2,$$

*where the factors for each method are defined as follows:*

$$g(\text{removal}) = \begin{cases} L & \text{if removal} = \textbf{\textit{baseline}} \text{ or } \textbf{\textit{marginal}} \\ L + 2BM & \text{if removal} = \textbf{\textit{conditional}}, \end{cases}$$

$$h(\text{summary}) = \begin{cases} 2\sqrt{d} & \text{if summary} = \textbf{\textit{Shapley}}, \textbf{\textit{Banzhaf}}, \text{ or } \textbf{\textit{leave-one-out}} \\ \sqrt{d} & \text{if summary} = \textbf{\textit{mean when included}}. \end{cases}$$

*Proof.* Given two inputs $x$ and $x'$, the results in Lemma 1 and Lemma 3 imply the following bound for any feature set $x_S$,

$$|f(x'_S) - f(x'_S)| \leq L' \cdot \|x_S - x'_S\|_2,$$

where we have $L' = L$ when using the baseline or marginal approach and $L' = L + 2BM$ when using the conditional approach. This bound depends on the specific subset, we also have a bound across all subsets because $\|x_S - x'_S\|_2 \leq \|x - x'\|_2$ for all $S \subseteq [d]$. In terms of the corresponding vectors $v, v' \in \mathbb{R}^{2^d}$, this implies the following bound:

$$\|v - v'\|_\infty = \max_{S \subseteq [d]} |v_S - v'_S| \leq L' \cdot \|x - x'\|_2.$$

Next, Lemma 7 shows that we have the following bound on the difference in attributions:

$$\|\phi(f, x) - \phi(f, x')\|_2 = \|Av - Av'\|_2 \leq \|A\|_{1,\infty} \cdot \|v - v'\|_\infty,$$

where the norm $\|A\|_{1,\infty}$ depends on the chosen summary technique. Combining these bounds, we arrive at the following result:

$$\|\phi(f, x) - \phi(f, x')\|_2 \leq \|A\|_{1,\infty} \cdot L' \cdot \|x - x'\|_2.$$

Substituting in the specific values for $L'$ and $\|A\|_{1,\infty}$ completes the proof.

$\square$

**Theorem 2.** *The robustness of removal-based explanations to model perturbations is given by the following meta-formula,*

$$\|\phi(f, x) - \phi(f', x)\|_2 \leq h(\text{summary}) \cdot \|f - f'\|,$$

*where the functional distance and factor associated with the summary technique are specified as follows:*

$$\|f - f'\| = \begin{cases} \|f - f'\|_\infty & \text{if removal} = \textbf{\textit{baseline}} \text{ or } \textbf{\textit{marginal}} \\ \|f - f'\|_{\infty,\mathcal{X}} & \text{if removal} = \textbf{\textit{conditional}}, \end{cases}$$

$$h(\text{summary}) = \begin{cases} 2\sqrt{d} & \text{if summary} = \textbf{\textit{Shapley}}, \textbf{\textit{Banzhaf}}, \text{ or } \textbf{\textit{leave-one-out}} \\ \sqrt{d} & \text{if summary} = \textbf{\textit{mean when included}}. \end{cases}$$

*Proof.* Given two models $f$ and $f'$, the results in Lemmas 5 and 6 imply the following bound for any feature set $x_S$,

$$|f(x_S) - f'(x_S)| \leq \|f - f'\|,$$

where the specific norm is $\|f - f'\|_\infty$ when using the baseline or marginal approach and $\|f - f'\|_{\infty,\mathcal{X}}$ when using the conditional approach. In terms of the corresponding vectors $v, v' \in \mathbb{R}^{2^d}$, this implies the following bound:

$$\|v - v'\|_\infty = \max_{S \subseteq [d]} |v_S - v'_S| \leq \|f - f'\|.$$

Next, Lemma 7 showed that we have the following bound on the difference in attributions:

$$\|\phi(f, x) - \phi(f', x)\|_2 = \|Av - Av'\|_2 \leq \|A\|_{1,\infty} \cdot \|v - v'\|_\infty,$$

where the norm $\|A\|_{1,\infty}$ depends on the chosen summary technique. Combining these bounds, we arrive at the following result:

$$\|\phi(f, x) - \phi(f', x)\|_2 \leq \|A\|_{1,\infty} \cdot \|f - f'\|.$$

Substituting in the specific values for $\|f - f\|$ and $\|A\|_{1,\infty}$ completes the proof.

$\square$

Next, we present a corollary that considers the case of simultaneous input and model perturbations. While this case has received less attention in the literature, we include this result for completeness because it is easy to analyze given our previous analysis.

**Corollary 1.** *The robustness of removal-based explanations to simultaneous input and model perturbations is given by the following meta-formula,*

$$\|\phi(f, x) - \phi(f', x')\|_2 \leq g(\text{removal}) \cdot h(\text{summary}) \cdot \|x - x'\|_2 + h(\text{summary}) \cdot \|f - f'\|,$$

*where the functional distance and factors associated with the removal and summary technique are given in Theorems 1 and 2.*

*Proof.* This result follows from triangle inequality:

$$\|\phi(f, x) - \phi(f', x')\|_2 \leq \|\phi(f, x) - \phi(f, x')\|_2 + \|\phi(f, x') - \phi(f', x')\|_2.$$

The bounds for the first and second terms are given by Theorem 1 and Theorem 2, respectively. $\square$

# F  The role of sampling when removing features

In this section, we present results analyzing the impact of sampling when removing features. Sampling is often required in practice to integrate across the distribution $q(\boldsymbol{x}_{\bar{S}})$ for removed features (see eq. (1)), and this can create further discrepancy between attributions for similar inputs.

## F.1  Theory

Here, we provide a probabilistic bound for the difference between the model prediction $f(x_S)$ with a subset of features $x_S$ and its estimator $\hat{f}(x_S)$ with $m$ independent samples from $q(\boldsymbol{x}_{\bar{S}})$.

**Lemma 12.** *Let $\hat{f}(x_S) \equiv \frac{1}{m} \sum_{i=1}^{m} f(x_S, x_{\bar{S}}^{(i)})$ be the empirical estimator of $f(x_S)$ in eq. (1), where $x_{\bar{S}}^{(i)} \overset{i.i.d.}{\sim} q(\boldsymbol{x}_{\bar{S}})$. For any $\delta \in (0, 1)$ and $\varepsilon > 0$, if the sample size $m$ satisfies $m \geq \frac{2B^2 \log(2/\delta)}{\varepsilon^2}$, then $\mathbb{P}\left(|\hat{f}(x_S) - f(x_S)| \leq \varepsilon\right) \geq 1 - \delta.$*

*Proof.* With Assumption 2, we have $-\frac{B}{m} \leq \frac{f(x_S, x_{\bar{S}}^{(i)})}{m} \leq \frac{B}{m}$. Also, $\mathbb{E}[\hat{f}(x_{\bar{S}})] = f(x_{\bar{S}})$ because $x_{\bar{S}}^{(i)}$ are sampled from $q(\boldsymbol{x}_{\bar{S}})$. Applying Hoeffding's inequality [35], we obtain the probabilistic bound

$$\mathbb{P}(|\hat{f}(x_S) - f(x_S)| \geq \varepsilon) \leq 2 \exp\left(-\frac{2\varepsilon^2}{\sum_{i=1}^{m} \left(\frac{2B}{m}\right)^2}\right)$$

$$= 2 \exp\left(-\frac{m\varepsilon^2}{2B^2}\right) \leq 2 \exp\left(-\frac{\left(\frac{2B^2 \log(2/\delta)}{\varepsilon^2}\right)\varepsilon^2}{2B^2}\right) = \delta,$$

and hence for the complement event we have

$$\mathbb{P}(|\hat{f}(x_S) - f(x_S)| \leq \varepsilon) \geq 1 - \delta.$$

$\square$

Lemma 12 can be applied to bound the difference between the original and perturbed attributions when the integral in eq. (1) is estimated via sampling. For example, under input perturbations, the difference $|\hat{f}(x_S) - \hat{f}(x'_S)|$ decomposes into $|\hat{f}(x_S) - f(x_S)| + |f(x_S) - f(x'_S)| + |f(x'_S) - \hat{f}(x'_S)|$ by the triangle inequality. The first and last terms can be bounded with high probability using Lemma 12 and the union bound, and the middle term is bounded using results from Section 3.1 (e.g., Lemma 1 or Lemma 3). Intuitively, by combining probabilistic bounds over all the feature subsets, it is possible to derive a final probabilistic upper bound with the following meta-formula:

$$\|\phi(f, x) - \phi(f, x')\|_2 \leq g(\text{removal}) \cdot h(\text{summary}) \cdot \|x - x'\|_2 + g_{\text{sampling}}(\text{removal}), \quad \text{(F.1.1)}$$

where $g_{\text{sampling}}(\text{removal})$ is a function of $\delta, \varepsilon, m$ when removal = **marginal** or **conditional**, and $g_{\text{sampling}}(\textbf{baseline}) = 0$ because no sampling is needed. Similarly, for our bound under model perturbations, the term $g_{\text{sampling}}(\text{removal})$ can be included to account for sampling in feature removal.

## F.2  Empirical observations

In eq. (F.1.1), the sampling-dependent term $g_{\text{sampling}}(\text{removal})$ does not depend on the input perturbation strength $\|x - x'\|_2$, so we empirically estimate $g_{\text{sampling}}(\text{removal})$ with the intercept of the empirical upper bound (as illustrated in Figure 3) when $\|x - x'\|_2 = 0$. As shown in Figure 13 (left), $g_{\text{sampling}}(\textbf{baseline})$ is indeed zero, whereas $g_{\text{sampling}}(\textbf{marginal})$ and $g_{\text{sampling}}(\textbf{conditional})$ empirically decrease with increasing sample size for feature removal as expected. We observe that the sampling error tends to be lower for the conditional approach compared to the marginal approach, potentially due to lower variances in the conditional versus marginal distributions. Finally, Figure 13 (right) confirms that the perturbation-dependent term $g(\text{removal}) \cdot h(\text{summary}) \cdot \|x - x'\|_2$ does not depend on the removal sample size.

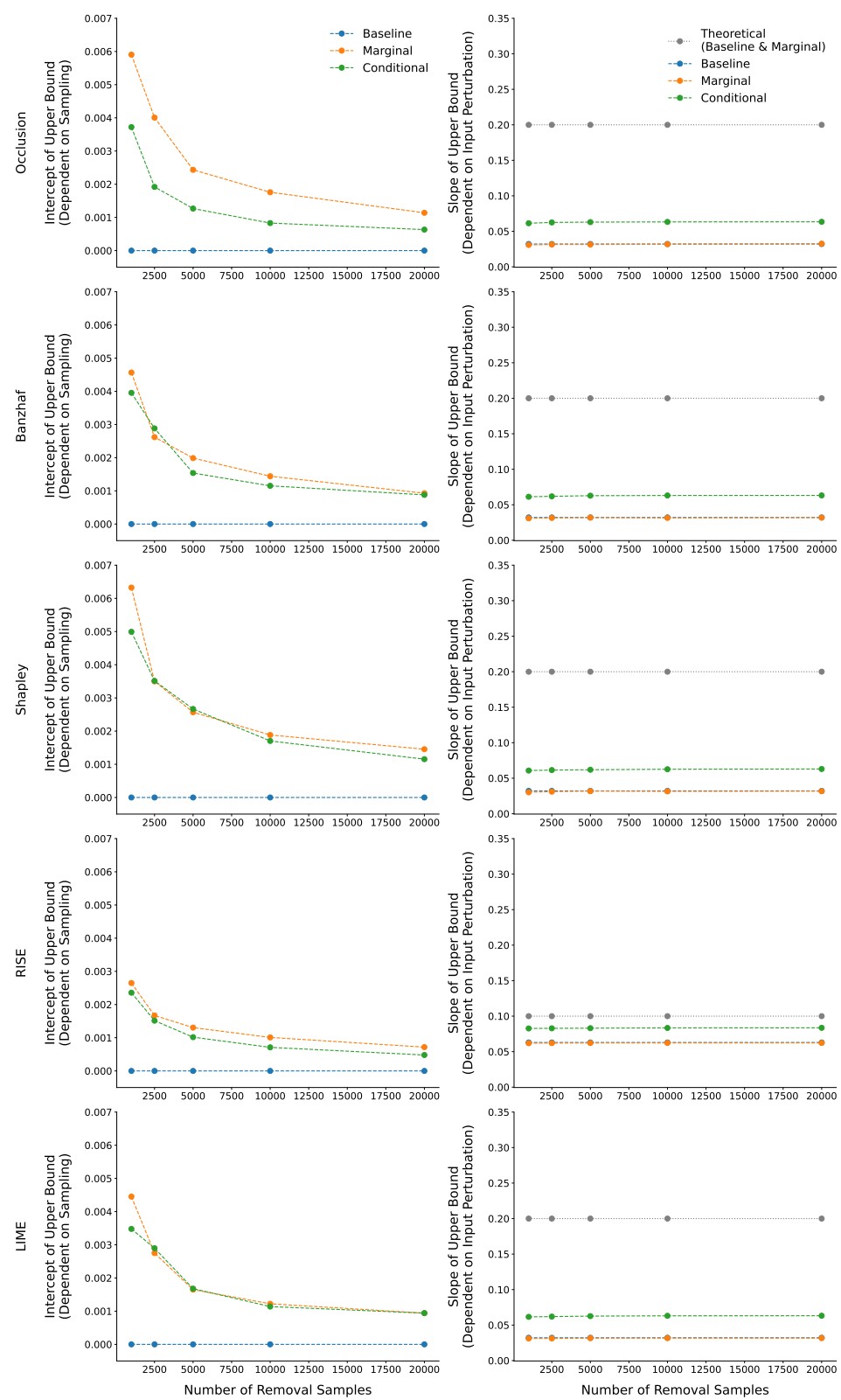

Figure 13: Intercepts and slopes of empirical and theoretical upper bounds with respect to the input perturbation norm, for Occlusion, Banzhaf, Shapley, RISE, and LIME attribution differences in the synthetic experiment in Appendix A.1, and with varying sample size for estimating feature removal.

# G Feature grouping

Rather than generate attribution scores for each feature $x_i$, another option is to partition the feature set and generate group-level scores. This is common in practice for images [12, 16, 53], where removing individual pixels is unlikely to produce meaningful prediction differences, and reducing the number of features makes attributions less computationally intensive. For example, LIME pre-computes superpixels using a basic segmentation algorithm [53], and ViT Shapley uses grid-shaped patches defined by a vision transformer [16].

When computing group-level attributions, we require a set of feature index groups that we denote as $\{G_1, \ldots, G_g\}$, where each group satisfies $G_i \subseteq [d]$. We assume that the $g$ groups are non-overlapping and cover all the indices, so that we have $G_i \cap G_j = \{\}$ for $i \neq j$ and $G_1 \cup \ldots \cup G_g = [d]$. Under this approach, the attributions are a vector of the form $\phi(f, x) \in \mathbb{R}^g$, and we compute them by querying the model only with subsets that are unions of the groups. For this, we let $T \subseteq [g]$ denote a subset of group indices and denote the corresponding union of groups as $G_T = \bigcup_{i \in T} G_i \subseteq [d]$. Using this notation and taking the Shapley value as an example, the attribution for the $i$th group $G_i$ is defined as

$$\phi_i(f, x) = \frac{1}{g} \sum_{T \subseteq [g] \setminus \{i\}} \binom{g-1}{|T|}^{-1} \left( f(x_{G_{T \cup \{i\}}}) - f(x_{G_T}) \right),$$

where $f(x_{G_T})$ is defined based on the chosen feature removal approach.

To generalize our earlier results to this setting, we first remark that querying the model with each subset induces a vector $v \in \mathbb{R}^{2^g}$ containing the predictions for each union $G_T$ with $T \subseteq [g]$. The attributions are then calculated according to $\phi(f, x) = Av$, where $A \in \mathbb{R}^{g \times 2^g}$ is the linear operator associated with each summary technique (see Table 3). Our approach for bounding the attribution difference under either input or model perturbation is similar here, because we can use the same inequalities from Lemma 7, or $||A(v - v')||_2 \leq ||A||_2 \cdot ||v - v'||_2$ and $||A(v - v')||_2 \leq ||A||_{1,\infty} \cdot ||v - v'||_\infty$, depending on which norm is available for the vectors of predictions $||v - v'||$.

Following the same proof techniques used previously, we can guarantee the following bounds in the feature grouping setting.

**Input perturbation.** Lemma 1 and Lemma 3 guarantee that $f(x_S)$ is Lipschitz continuous for any subset $S \subseteq [d]$. These results automatically apply to the specific subsets induced by feature grouping. If $f(x_S)$ is $L'$-Lipschitz continuous for all $S \subseteq [d]$, we therefore have the following inequality,

$$|f(x_{G_T}) - f(x'_{G_T})| \leq L' \cdot ||x_{G_T} - x'_{G_T}||_2,$$

for all group subsets $T \subseteq [g]$. As a result, we can guarantee that the induced vectors $v, v' \in \mathbb{R}^{2^g}$ for two inputs $x, x'$ have the following bound:

$$||v - v'||_\infty \leq L' \cdot ||x - x'||_2.$$

Finally, we see that Theorem 1 only needs to be changed by replacing $d$ with $g$ in the $h(\text{summary})$ term. This bound may even be tighter when we use feature groups, because the Lipschitz constant may be smaller when we restrict our attention to a smaller number of subsets.

**Model perturbation.** Lemma 5 and Lemma 6 guarantee that the predictions $f(x_S)$ and $f'(x_S)$ for two models $f, f'$ can differ by no more than the functional difference $||f - f'||$, where the specific norm depends on the feature removal approach. These results apply to all subsets $S \subseteq [d]$, so they automatically apply to those induced by feature grouping. We therefore have the following inequality,

$$|f(x_{G_T}) - f'(x_{G_T})| \leq ||f - f'||,$$

for all $x$ and all group subsets $T \subseteq [g]$. For the induced vectors $v, v' \in \mathbb{R}^{2^g}$, we also have

$$||v - v'||_\infty \leq ||f - f'||.$$

Finally, we see that Theorem 2 applies as well, where the only change required is replacing $d$ with $g$ in the $h(\text{summary})$ term.

# H  LIME weighted least squares linear operator

In this section, we discuss the linear operator $A \in \mathbb{R}^{d \times 2^d}$ associated with LIME [53]. We begin with a theoretical characterization that focuses on LIME's default implementation choices, and we then corroborate these points with empirical results.

## H.1  Theory

The observation that LIME is equivalent to a linear operator $A \in \mathbb{R}^{d \times 2^d}$ is discussed in Appendix D. We pointed out there that LIME's specific operator depends on several implementation choices, including the choice of weighting kernel, regularization and intercept term. Here, we consider that no regularization is used, that an intercept term is fit, and we focus on the default weighting kernel in LIME's popular implementation.[4] The default is the following exponential kernel,

$$w(S) = \exp\left(-\frac{\left(1 - \sqrt{|S|/d}\right)^2}{\sigma^2}\right), \tag{H.1.1}$$

where $\sigma^2 > 0$ is a hyperparameter. The user can set this parameter arbitrarily, but LIME also considers default values for each data type: $\sigma^2 = 0.25$ for images, $\sigma^2 = 25$ for language, and $\sigma^2 = 0.75\sqrt{d}$ for tabular data. We find that these choices approach two limiting cases, $\sigma^2 \to \infty$ and $\sigma^2 \to 0$, where LIME reduces to other summary techniques; namely, we find that these limiting cases respectively recover the Banzhaf value [9] and leave-one-out [67] summary techniques. We argue this first from a theoretical perspective, and Appendix H.2 then provides further empirical evidence.

**Limiting case $\sigma^2 \to \infty$.** In this case, it is easy to see from eq. (H.1.1) that we have $w(S) \to 1$ for all $S \subseteq [d]$. The weighted least squares objective for LIME then reduces to an unweighted least squares problem, which according to results from Hammer and Holzman [32] recovers the Banzhaf value. We might expect that the default settings of $\sigma^2$ approach this outcome in practice (e.g., $\sigma^2 = 25$), and we verify this in our empirical results.

**Limiting case $\sigma^2 \to 0$.** As we let $\sigma^2$ approach 0, increasing weight is applied to $w(S)$ when the cardinality $|S|$ is large. Similar to a softmax activation, the differences between cardinalities become exaggerated with low temperatures; for very low temperatures, the weight applied to $|S| = d$ is much larger than the weight applied to $|S| = d - 1$, which is much larger than the weight for $|S| = d - 2$, etc. In this regime, applying weight only to the subset $S = [d]$ leaves the problem undetermined, so the small residual weight on $|S| = d - 1$ plays an important role. The problem begins to resemble one where only subsets with $|S| \geq d - 1$ are taken into account, which yields a unique solution: following Covert et al. [17], this reduces LIME to leave-one-out (Occlusion). More formally, if we denote $A_{\text{LIME}}(\sigma^2)$ as the linear operator resulting from a specific $\sigma^2$ value, we argue that $\lim_{\sigma^2 \to 0} A_{\text{LIME}}(\sigma^2)$ is the operator associated with leave-one-out. We do not prove this result due to the difficulty of deriving $A_{\text{LIME}}(\sigma^2)$ for a specific $\sigma^2$ value, but Appendix H.2 provides empirical evidence for this claim.

## H.2  Empirical analysis

Here, we denote the LIME linear operator with the exponential kernel as $A_{\text{LIME}}$ instead of $A_{\text{LIME}}(\sigma^2)$ for brevity. Recall from Appendix D that the LIME linear operator has the form $A_{\text{LIME}} = (Z'^\top W Z')^{-1}_{[1:]} Z'^\top W$, where $Z' = \{0, 1\}^{2^d \times (d+1)}$ is a binary matrix with all ones in the first column and the other columns indicating whether each feature is in each subset, and $W \in \mathbb{R}^{2^d \times 2^d}$ is a diagonal matrix with $W_{S,S} = w(S)$ for all $S \in [d]$ and zeros elsewhere.

To empirically compare the linear operator of LIME to those of the Banzhaf value and leave-one-out (Occlusion), we first derive closed-form expressions of $Z'^\top W Z'$ and $Z'^\top W$ as in the following proposition.

---

[4] https://github.com/marcotcr/lime

**Proposition 2.** *Define*

$$a_0 \equiv \sum_{k=0}^{d} \binom{d}{k} \exp\left(-\left(1 - \sqrt{k/d}\right)^2 / \sigma^2\right),$$

$$a_1 \equiv \sum_{k=1}^{d} \binom{d-1}{k-1} \exp\left(-\left(1 - \sqrt{k/d}\right)^2 / \sigma^2\right), \text{ and}$$

$$a_2 \equiv \sum_{k=2}^{d} \binom{d-2}{k-2} \exp\left(-\left(1 - \sqrt{k/d}\right)^2 / \sigma^2\right).$$

*Then* $(Z'^\top W Z')$ *is a* $(d+1) \times (d+1)$ *matrix with the following form:*

$$(Z'^\top W Z') = \begin{pmatrix} a_0 & B \\ B^\top & C \end{pmatrix},$$

*where* $B = (a_1, ..., a_1) \in \mathbb{R}^{1 \times d}$ *, and* $C$ *is a* $d \times d$ *matrix with* $a_1$ *on the diagonal entries and* $a_2$ *on the off-diagonal entries. Furthermore, for all* $S \in [d]$

$$(Z'^\top W)_{iS} = \begin{cases} w(S), & \text{if } i = 1 \\ \mathbb{1}\{i - 1 \in S\}w(S) & \text{otherwise.} \end{cases}$$

*Proof.* We first find an expression for each element in $Z'^\top W Z'$. Consider the first diagonal entry:

$$(Z'^\top W Z')_{11} = Z_1'^\top W Z_1' = \mathbf{1}_{2^d}^\top W \mathbf{1}_{2^d}$$

$$= \sum_{S \subseteq [d]} w(S)$$

$$= \sum_{k=0}^{d} \binom{d}{k} \exp\left(-\left(1 - \sqrt{k/d}\right)^2 / \sigma^2\right) = a_0.$$

For the rest of the diagonal entries, we have:

$$(Z'^\top W Z')_{ii} = Z_i'^\top W Z_i' = \sum_{S: i-1 \in S} w(S)$$

$$= \sum_{k=1}^{d} \binom{d-1}{k-1} \exp\left(-\left(1 - \sqrt{k/d}\right)^2 / \sigma^2\right) = a_1,$$

for $i = 2, ..., d + 1$. For the off-diagonal entries, we have:

$$(Z'^\top W Z')_{1i} = (Z'^\top W Z')_{i1} = \mathbf{1}_{2^d}^\top W Z_i' = \sum_{S: i-1 \in S} w(S) = a_1,$$

for $i = 2, ..., d + 1$. Finally, consider the rest of the off-diagonal entries:

$$(Z'^\top W Z')_{ij} = Z_i'^\top W Z_j' = \sum_{S: i-1 \in S, j-1 \in S} w(S)$$

$$= \sum_{k=2}^{d} \binom{d-2}{k-2} \exp\left(-\left(1 - \sqrt{k/d}\right)^2 / \sigma^2\right) = a_2,$$

for $i, j = 2, ..., d + 1$ such that $i \neq j$. Taken all these together, $Z'^\top W Z'$ has the closed form expression as shown in the proposition.

We now proceed to find an expression for each element in $Z'^\top W$, which has the first row entries:

$$(Z'^\top W)_{1S} = \mathbf{1}_{2^d}^\top W_S = w(S),$$

for all $S \subseteq [d]$, whereas the rest of the rows have entries:

$$(Z'^\top W)_{iS} = Z_i'^\top W_S = \mathbb{1}\{i - 1 \in S\}w(S),$$

for $i = 2, ..., d + 1$. Hence the closed form expression of $Z'^\top W$ in the proposition follows. □

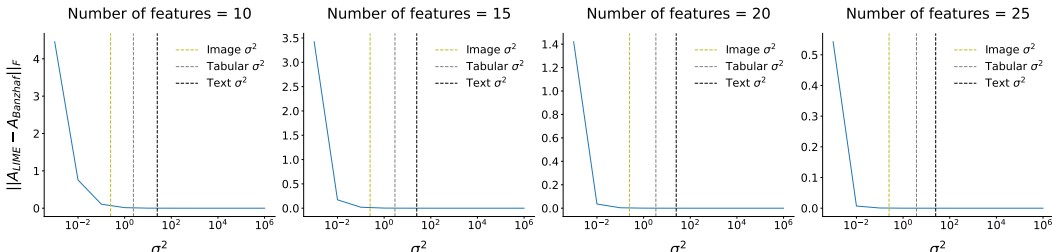

Figure 14: Frobenius norm of the difference between the LIME linear operator $A_{\text{LIME}}$ and the Banzhaf linear operator $A_{\text{Banzhaf}}$, with varying LIME exponential kernel hyperparameter $\sigma^2$ and number of features. We include reference lines for the default LIME hyperparameters: $\sigma^2 = 0.25$ for images, $\sigma^2 = 0.75\sqrt{d}$ for tabular data, and $\sigma^2 = 25$ for text data.

With closed form expressions for $Z'^\top W Z'$ and $Z'^\top W$, we numerically compute the LIME linear operator $A_{\text{LIME}} = (Z'^\top W Z')^{-1}_{[1:]} Z'^\top W$ and compare it to the linear operators of the Banzhaf value ($A_{\text{Banzhaf}}$) and Occlusion ($A_{\text{Occlusion}}$) with both large and small $\sigma^2$ values.

As shown in Figure 14, as $\sigma^2$ increases, the Frobenius norm difference between the LIME and Banzhaf linear operators approaches zero, verifying our theoretical argument for the limiting case of $\sigma^2 \to \infty$. Furthermore, we observe that $A_{\text{LIME}}$ and $A_{\text{Banzhaf}}$ have near zero difference for all default $\sigma^2$ values when there are $\{10, 15, 20, 25\}$ features. As $\sigma^2$ approaches zero, the Frobenius norm difference between the LIME and Occlusion linear operators empirically approaches zero (Figure 15), confirming our intuition for the limiting case where $\sigma^2 \to 0$. Based on our computational results, LIME with the default $\sigma^2$ values indeed has similar spectral norm $\|\cdot\|_2$ and the operator norm $\|\cdot\|_{1,\infty}$ compared to the Banzhaf value (Figure 16).

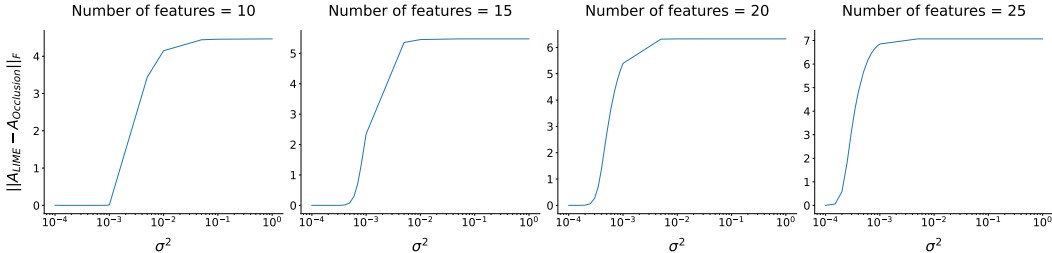

Figure 15: Frobenius norm of the difference between the LIME linear operator $A_{\text{LIME}}$ and the Occlusion linear operator $A_{\text{Occlusion}}$, with varying LIME parameter $\sigma^2$ and number of features.

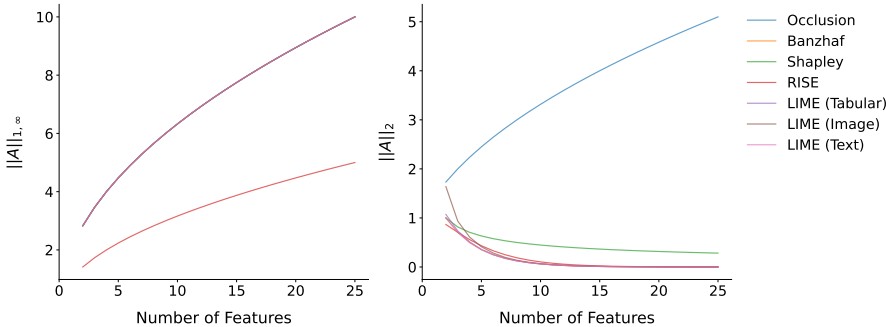

Figure 16: Computed norms of summary technique linear operators. Note that all lines except for RISE overlap on the left-hand-side showing the operator norm.

# I Additional results for synthetic experiments

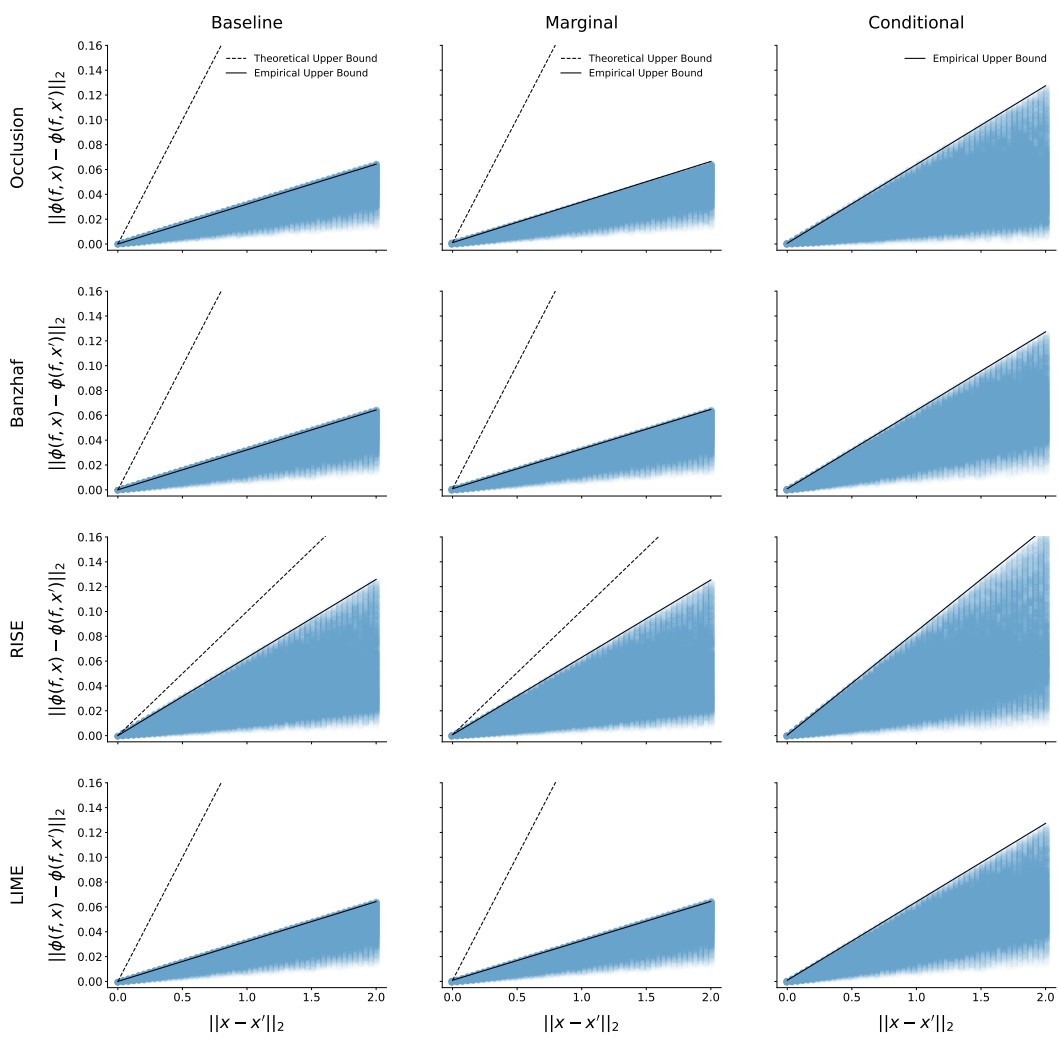

Figure 17: Occlusion, Banzhaf, RISE, and LIME attribution differences under input perturbation with various perturbation norms in our synthetic data experiment. Lines bounding the maximum attribution difference at each perturbation norm are shown as empirical upper bounds. Theoretical upper bounds are included for baseline and marginal feature removal, and omitted for conditional feature removal because it is infinite for $\rho = 1$.

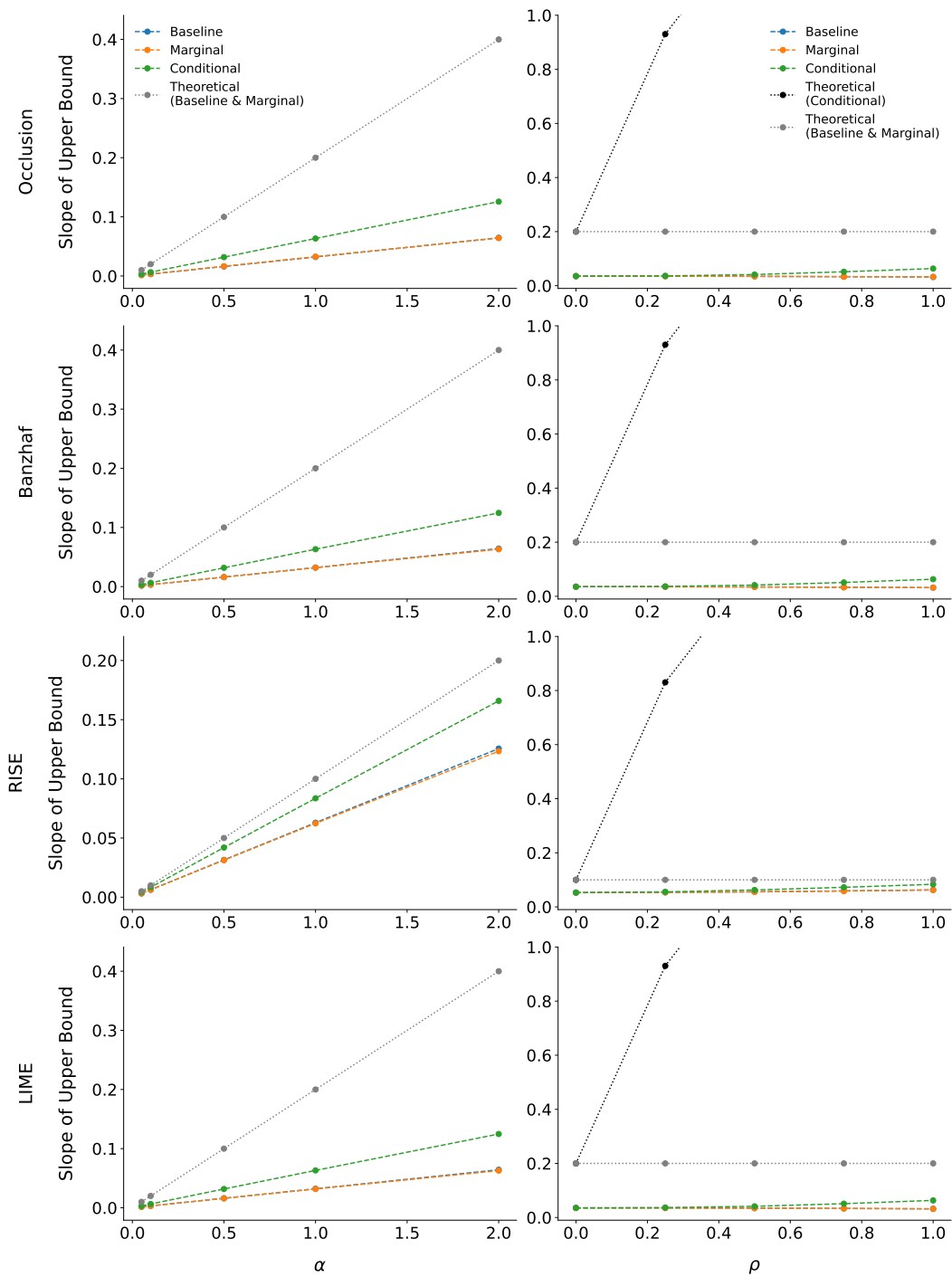

Figure 18: Slopes of empirical and theoretical upper bounds with respect to the input perturbation norm, for Occlusion, Banzhaf, RISE, and LIME attribution differences in the synthetic experiment. The parameter $\alpha$ varies the logistic regression Lipschitz constant, and $\rho$ varies the correlation between the two synthetic features $x_1, x_2$.

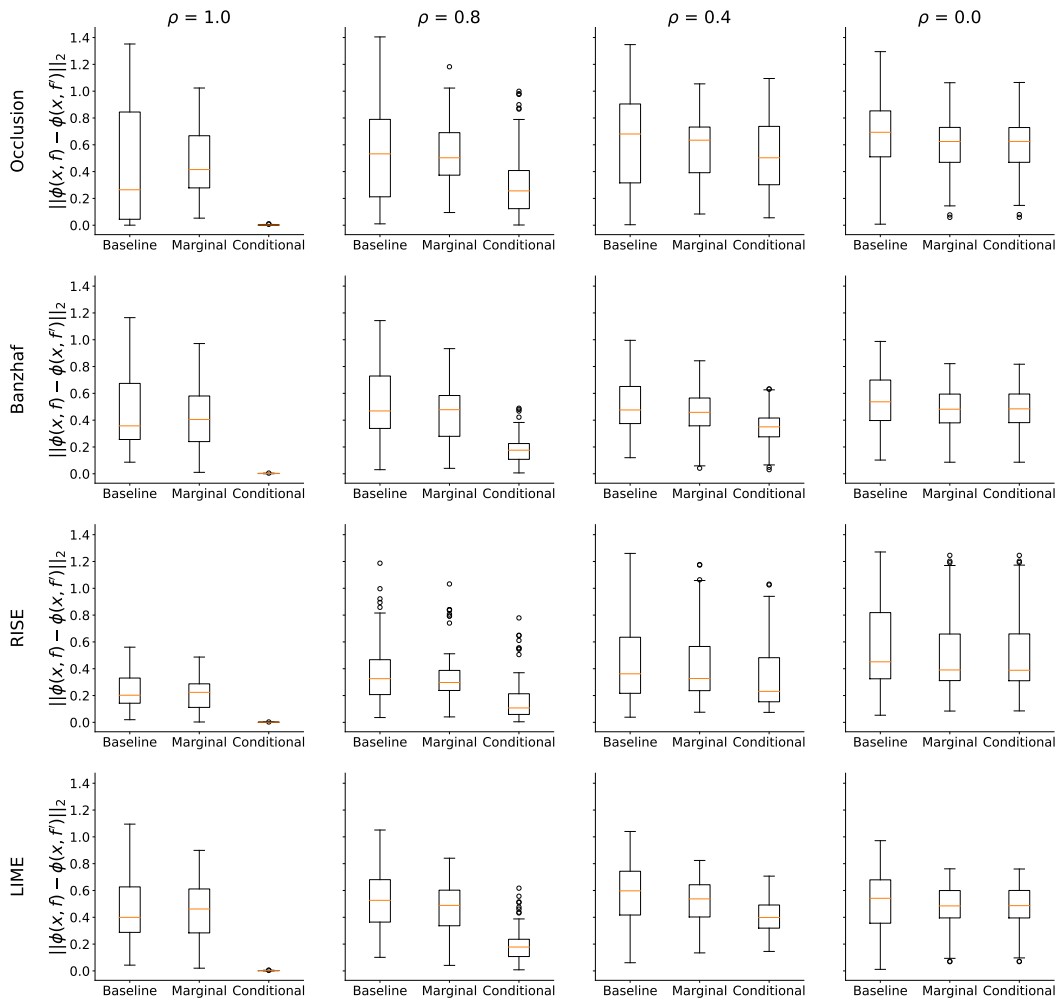

Figure 19: Occlusion, Banzhaf, RISE, and LIME attribution differences between the logistic regression classifiers $f, f'$ as constructed in Appendix A.2 with baseline, marginal, and conditional feature removal, and varying correlation $\rho$.

## J   Implementation details

**Fully connected networks.** For the wine quality dataset, we trained FCNs with varying levels of weight decay. The architecture consists of 3 hidden layers of width 128 with ReLU activations and was trained with Adam for 50 epochs with learning rate 0.001. We also trained a conditional VAE (CVAE) following the procedure in Frye et al. [27] to approximate conditional feature removal. The CVAE encoder and decoder both consist of 2 hidden layers of width 64 with ReLU activations and latent dimension 8. We trained the CVAE using a prior regularization strength of 0.5 for at most 500 epochs with Adam and learning rate 0.0001, with early stopping if the validation mean squared error did not improve for 100 consecutive epochs. All models were trained with NVIDIA GeForce RTX 2080 Ti GPUs with 11GB memory.

**Convolutional networks.** We trained CNNs for MNIST with varying levels of weight decay. Following the LeNet-like architecture in Adebayo et al. [2], our network has the following structure: Input $\rightarrow$ Conv($5 \times 5$, 32) $\rightarrow$ MaxPool($2 \times 2$) $\rightarrow$ Conv($5 \times 5$, 64) $\rightarrow$ MaxPool($2 \times 2$) $\rightarrow$ Flatten $\rightarrow$ Linear(1024, 1024) $\rightarrow$ Linear(1024, 10). We trained the model with Adam for 200 epochs with learning rate 0.001. The same GPUs for FCN training were used to train the CNNs.

**ResNets.** We trained ResNet-18 networks for CIFAR-10 from scratch with Adam for 500 epochs with learning rate 0.00005. The same GPUs for FCN training were used to train the ResNet-18 networks. We used a ResNet-50 network pre-trained on ImageNet and fine-tuned it for Imagenette with Adam for 20 epochs with learning rate 0.0005. The ResNet-50 networks were trained with NVIDIA Quadro RTX6000 GPUs with 24GB memory.

# K   Additional results for practical implications

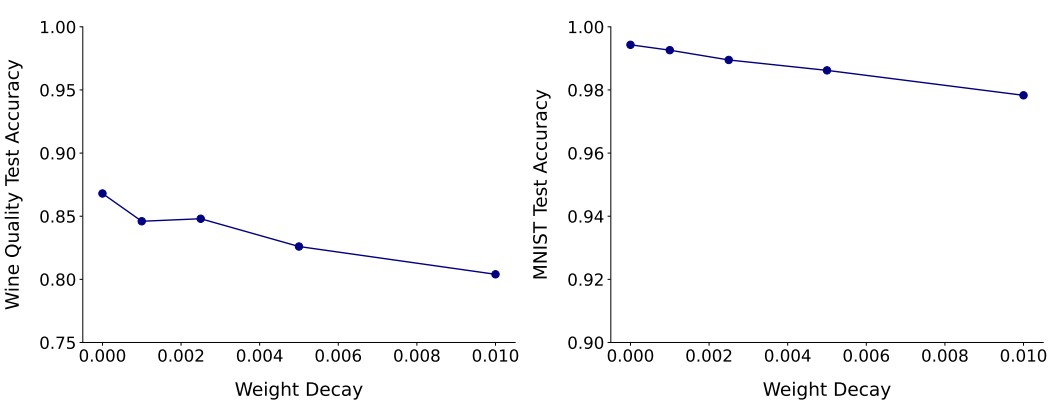

Figure 20: Test accuracy of FCN trained on the wine quality dataset and CNN trained on MNIST with varying degree of weight decay.

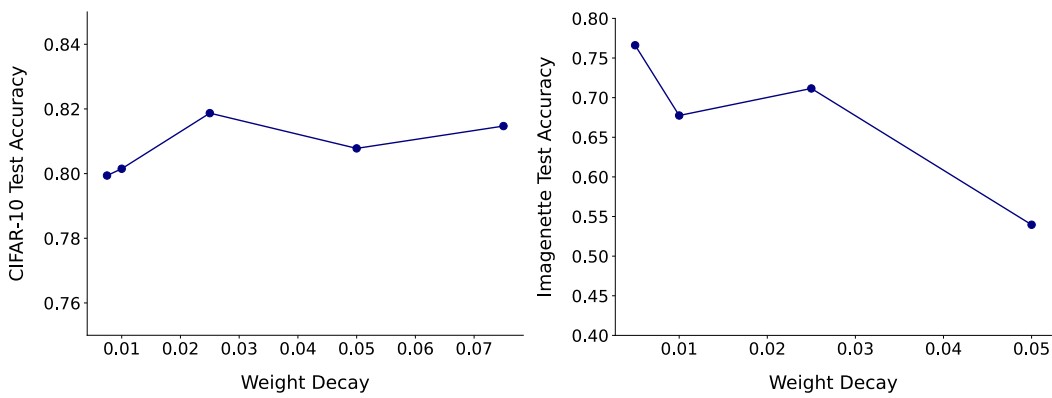

Figure 21: Test accuracy of ResNet-18 trained on CIFAR-10 and ResNet-50 fine-tuned on Imagenette with varying degree of weight decay.

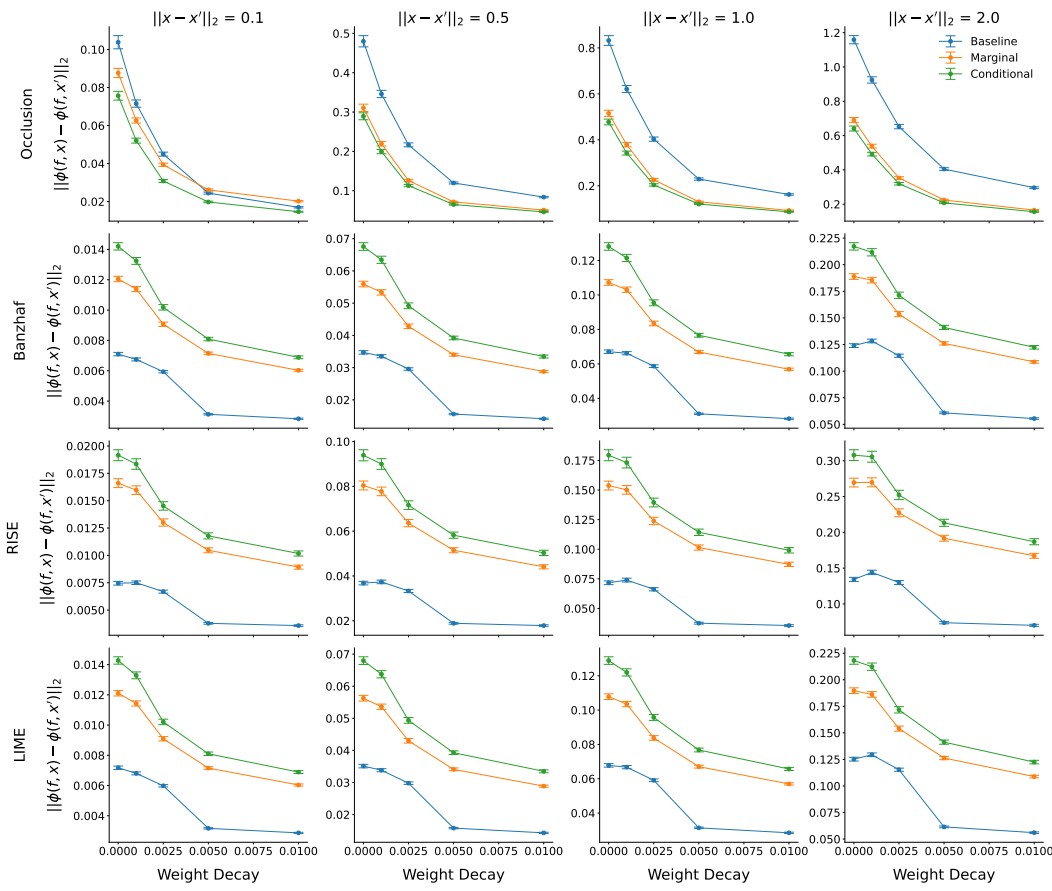

Figure 22: Attribution difference for networks trained with increasing weight decay, under input perturbations with perturbation norms $\{0.1, 0.5, 1, 2\}$. The results include the wine quality dataset with FCNs and baseline (replacing with training set minimums), marginal, and conditional feature removal. Error bars show the mean and $95\%$ confidence intervals across explicand-perturbation pairs.

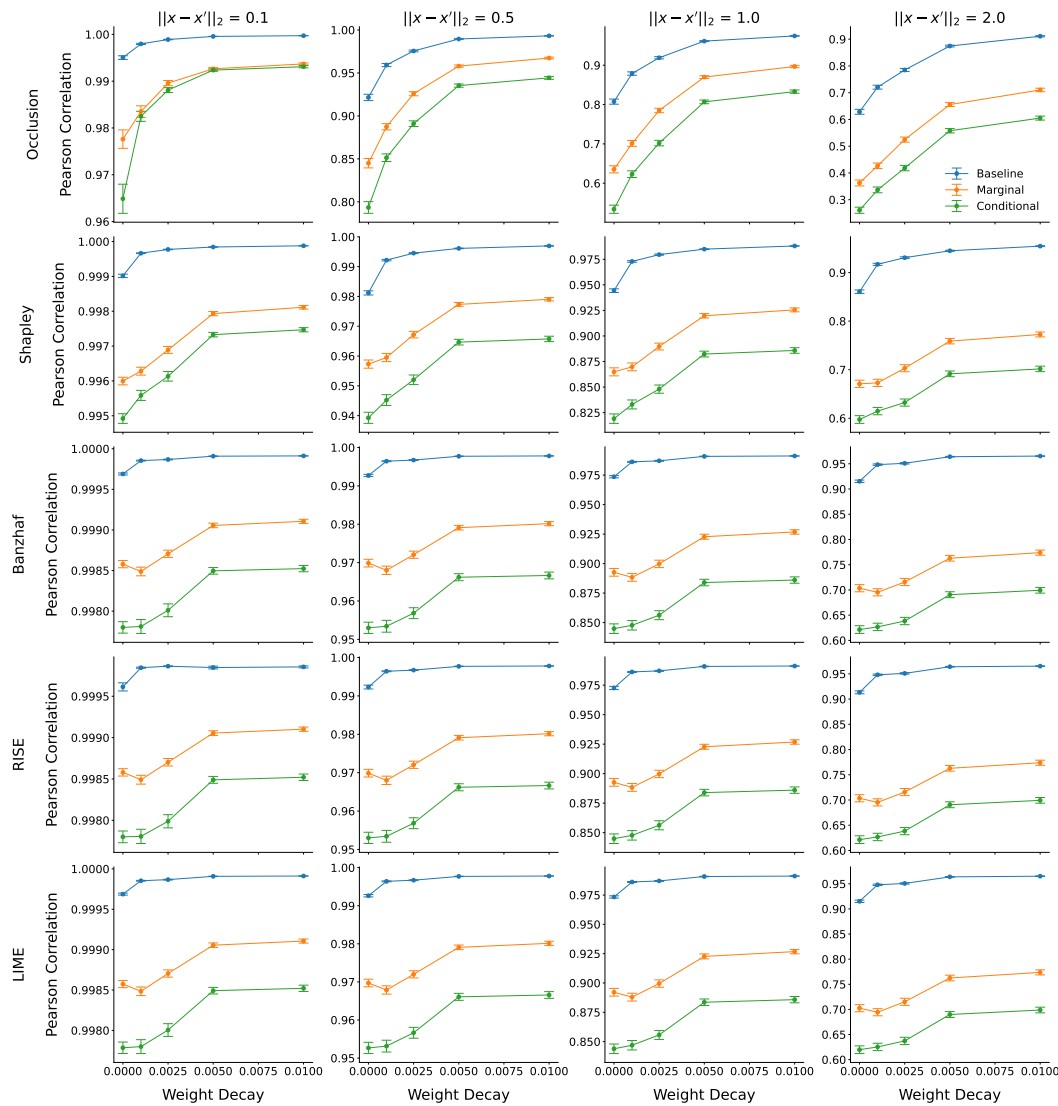

Figure 23: Pearson correlation of attributions for networks trained with increasing weight decay, under input perturbations with perturbation norms $\{0.1, 0.5, 1, 2\}$. The results include the wine quality dataset with FCNs and baseline (replacing with training set minimums), marginal, and conditional feature removal. Error bars show the means and $95\%$ confidence intervals across explicand-perturbation pairs.

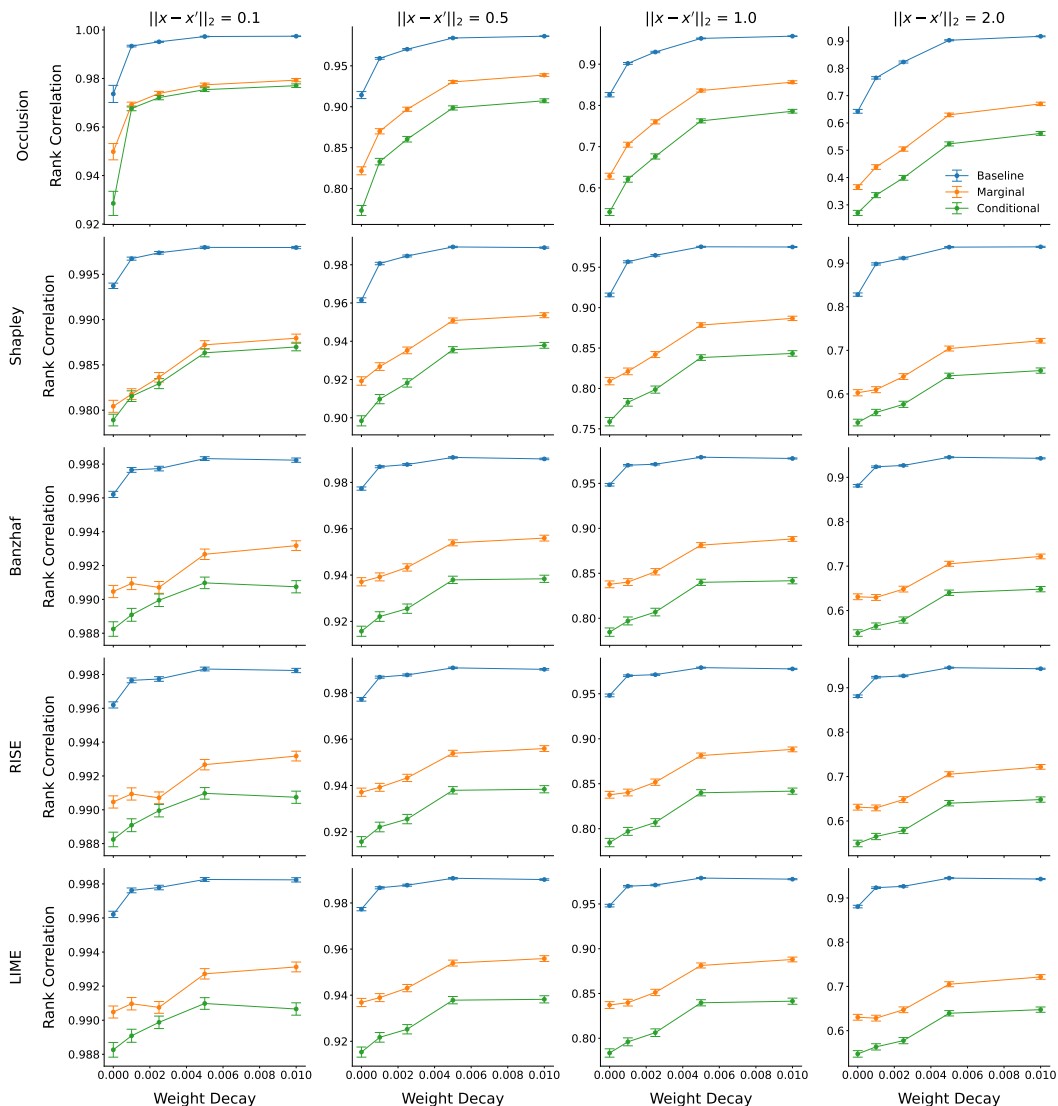

Figure 24: Spearman rank correlation of attributions for networks trained with increasing weight decay, under input perturbations with perturbation norms $\{0.1, 0.5, 1, 2\}$. The results include the wine quality dataset with FCNs and baseline (replacing with training set minimums), marginal, and conditional feature removal. Error bars show the means and $95\%$ confidence intervals across explicand-perturbation pairs.

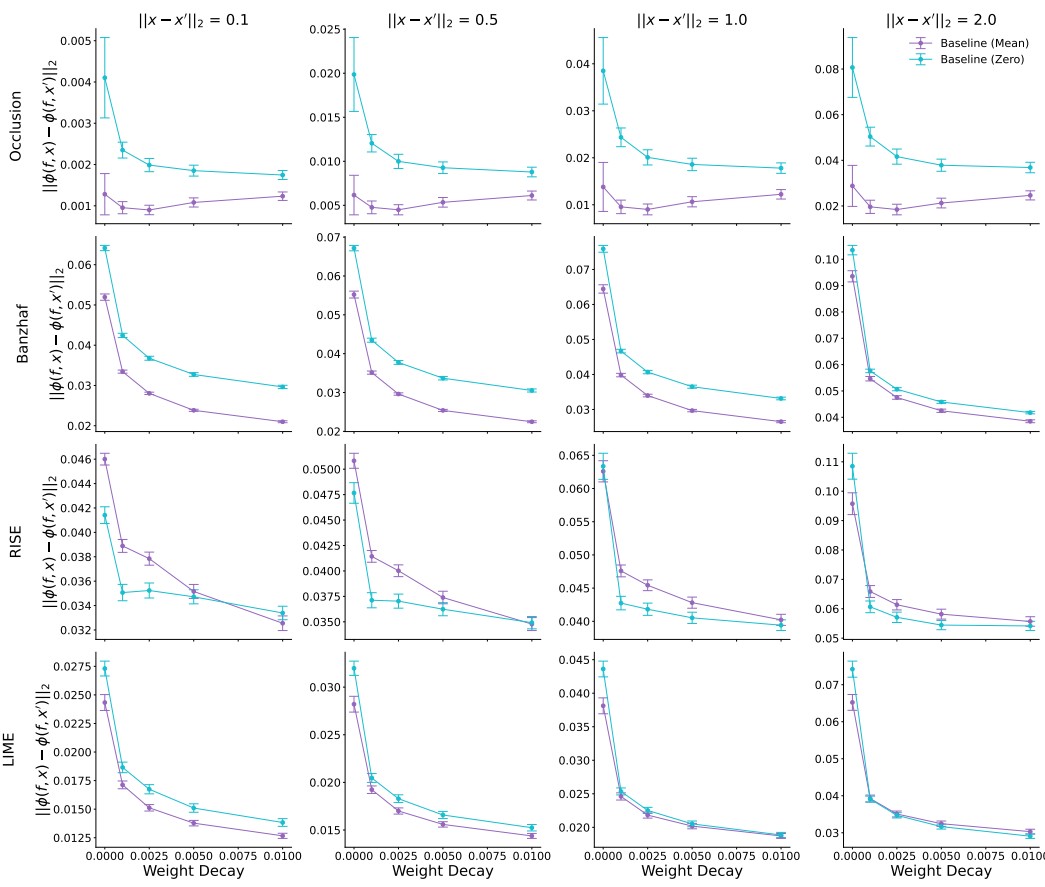

Figure 25: Attribution difference for networks trained with increasing weight decay, under input perturbations with perturbation norms $\{0.1, 0.5, 1, 2\}$. The results include MNIST with CNNs and baseline feature removal with either training set means or zeros. Error bars show the means and $95\%$ confidence intervals across explicand-perturbation pairs.

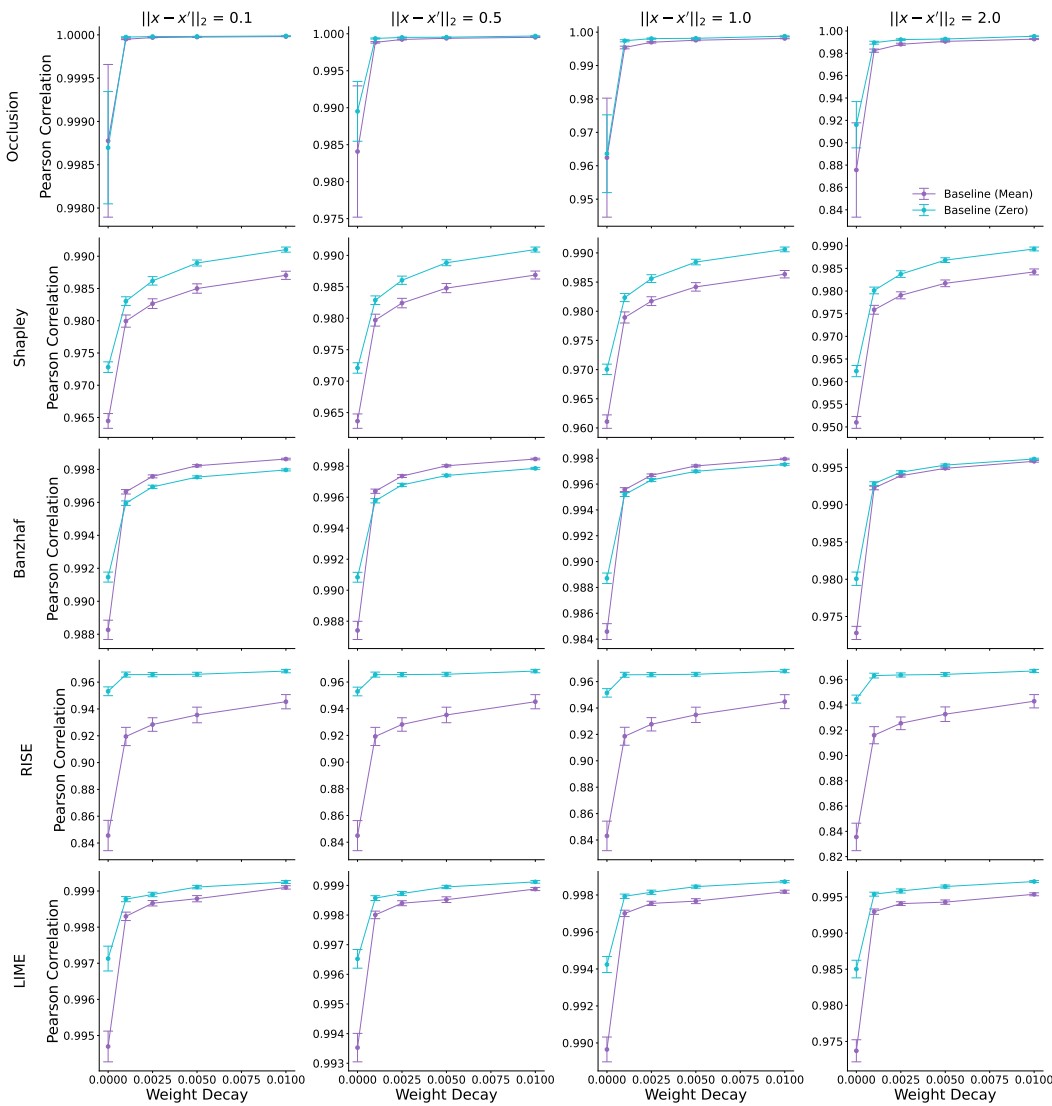

Figure 26: Pearson correlation of attributions for networks trained with increasing weight decay, under input perturbations with perturbation norms $\{0.1, 0.5, 1, 2\}$. The results include MNIST with CNNs and baseline feature removal with either training set means or zeros. Error bars show the means and $95\%$ confidence intervals across explicand-perturbation pairs.

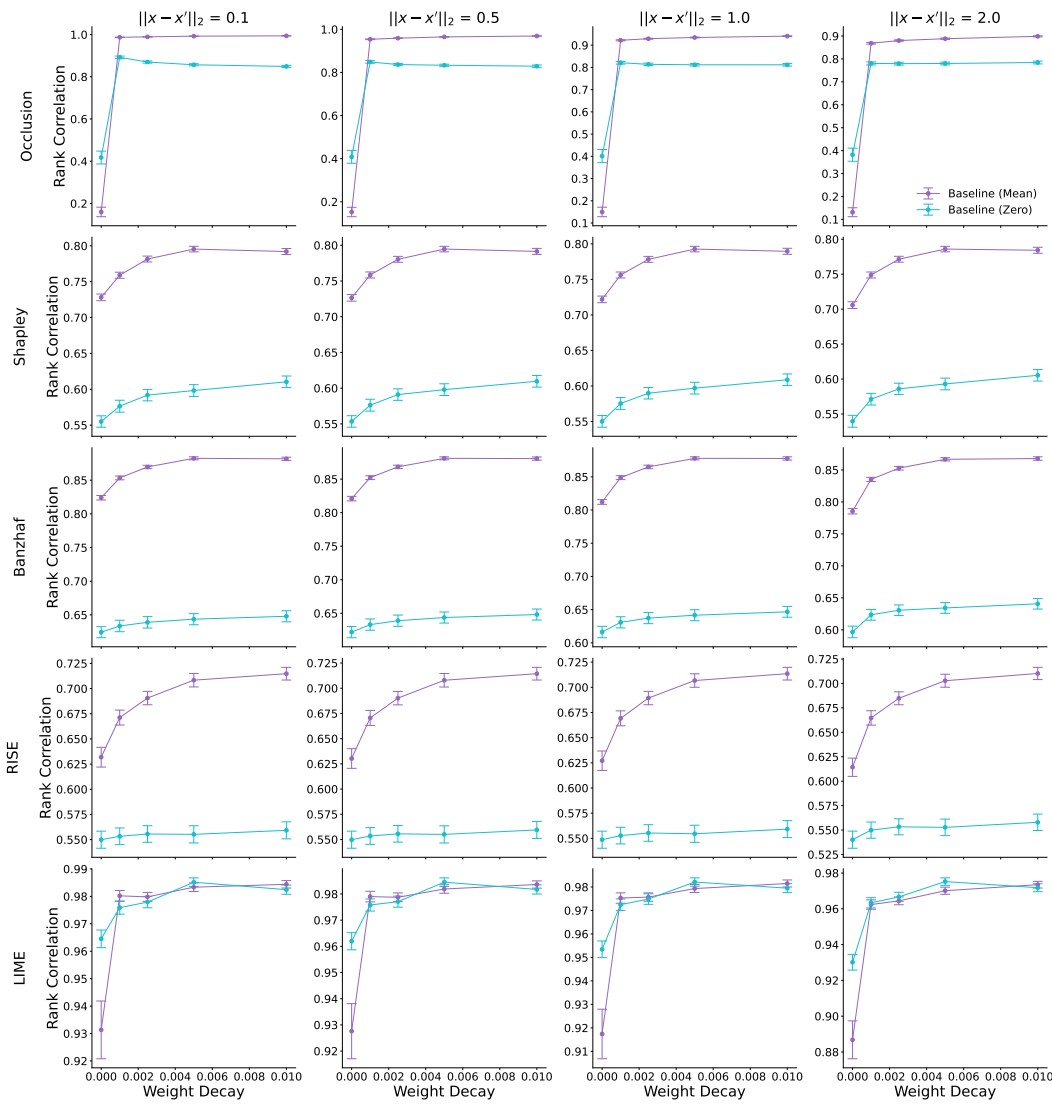

Figure 27: Spearman rank correlation of attributions for networks trained with increasing weight decay, under input perturbations with perturbation norms $\{0.1, 0.5, 1, 2\}$. The results include MNIST with CNNs and baseline feature removal with either training set means or zeros. Error bars show the means and $95\%$ confidence intervals across explicand-perturbation pairs.

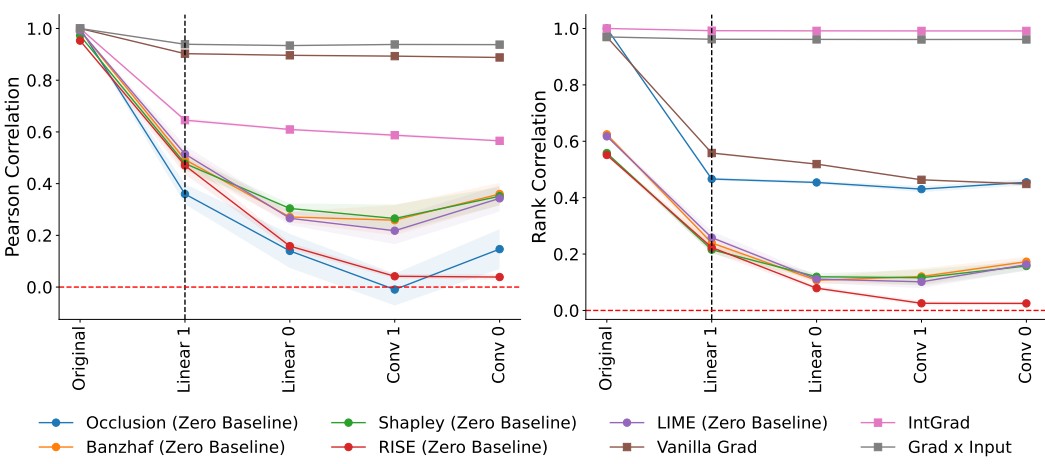

Figure 28: Sanity checks for attributions using cascading randomization for the CNN trained on MNIST. Features are removed by replacing them with zeros. Attribution similarity is measured by Pearson correlation and Spearman rank correlation. We show the means and 95% confidence intervals across 10 random seeds.

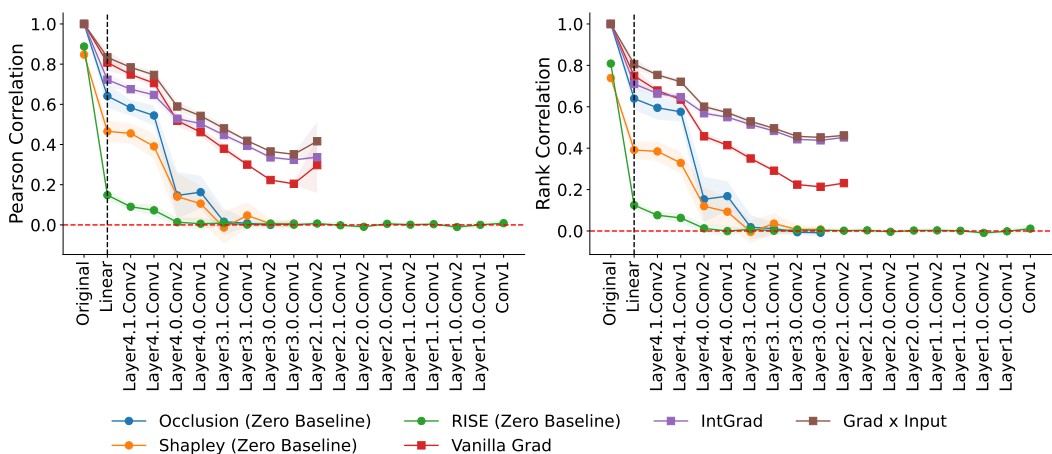

Figure 29: Sanity checks for attributions using cascading randomization for the ResNet-18 trained on CIFAR-10. Features are removed by replacing them with zeros. Attribution similarity is measured by Pearson correlation and Spearman rank correlation. We show the means and 95% confidence intervals across 10 random seeds. Missing points correspond to undefined correlations due to constant attributions under model randomization.

