# OpenReview forum: "On the Robustness of Removal-Based Feature Attributions"
_NeurIPS.cc/2023/Conference — NeurIPS 2023 poster_

### Official Review · Reviewer_dzDD · 2023-07-04

**Soundness:** 4 excellent
**Presentation:** 3 good
**Contribution:** 2 fair
**Rating:** 6
**Confidence:** 4

**Summary:**

This paper derives robustness results regarding a general class of feature attribution methods referred to as “removal-based feature attributions”, which includes occlusion methods, but also Shapley values and LIME explanations. The authors study Lipschitz-continuity properties of these methods with respect to changes in the modeled function and the inputs. They find that, under Lipschitz assumptions on the classifiers output, the removal-based attributions are also Lipschitz-continuous w.r.t to deviation in the inputs and the model.

**Strengths:**

**Generality.** The class of feature attributions studied in this work is kept quite general and it thus covers many relevant implementations such as Occlusion, LIME, and Shapley values.

**Technical soundness.** I checked the proofs of some main results and was not able to identify any flaws This makes this work is a rigorous technical contribution in my view.

**Good exposition and presentation.** The formalization and notation were introduced well, and the results were good to follow. The proofs that I checked were sufficiently detailed to convince me of the correctness of the theoretical claims. The tables help to get an overview over the various results.

**Interesting results on robust summarization techniques.** I particularly like the results on the robustness of the summarization techniques and the statements of the most robust aggregation techniques under each set of axioms (Lemma 8 - Lemma 11). The do not rely on any assumptions and can be operationalized directly, for instance for robust data valuation as done in the recent work mentioned (Wang & Jia, 2023).



**Weaknesses:**

**How robust should it be?** While the results are interesting, they are hard to interpret, because there is not optimal robustness. As the authors correctly state, too small robustness can be considered as invariance as in the sense of Adebayo et al. On the contrary, low robustness allows for adversarial attributions. Therefore, the computation of the robustness score allows for no clear interpretation.

**Given the formalization done in prior work, the main results are not unexpected.** First, I would like to underline that central parts of the formalization in this work were transferred from Covert et al., 2021. For instance, the distinction between the removal strategy and the summarization technique and the representation of the methods in this scheme was introduced in this work. I think this should be pronounced more clearly. Further, we assume the model to be Lipschitz-continuous (Assumption 1), the probability divergence when changing the set of present features to be Lipschitz as well (Assumption 3), and we consider only attributions that can be represented as linear operator of the model outputs, it is not very surprising that we arrive at Lipschitz-continuity of the attributions w.r.t. to inputs and the model. While I see that intuitive results also need to be rigorously proven, I am uncertain about how many new insights are added.

**Implications and Experiments.** The practical implications of the results still remain unclear to some extent. Taking a practitioner’s perspective, I wonder about what novel insights can be gained from this work. The finding that regularized networks have more robust (or less noisy) attributions, has already been established (e.g., by Shah et al, 2021, Dombrowski et al.). On a sidenote, the Lipschitz-continuity does not imply that the method needs to pass the parameter randomization test, as a constant function would still be Lipschitz-continuous. I would be more interested in some constructive way to operationalize the findings, for instance to prevent “adversarial” explanations as hinted in the text? Can they help us construct better techniques? I think these questions are interesting and to this end, to me, it is seems unfortunate that the experiments already conducted and some further experimentation towards this direction did not make it into the main paper.

**Minor Points**

A line of related work uses removal strategies to benchmark feature attributions (e.g., Tomsett et al., 2020; Rong et al., 2022), which could be discussed as potential a related work. I wonder if the results in this work bear some connection with attribution evaluation problem, which could be a possible application.

Table 2 seems not to be referenced in the text and thus appears a bit context-less.

--------------------------------------

**Summary.**
A rigorously executed paper with general theoretic results on the robustness of attribution methods. However, the practical implications remain a bit unclear and there are no experiments in the main paper. Overall, the paper seems just above the bar of acceptance to me. I would be willing to further increase my score if the authors can convince me of the practical relevance and impact of their results.


**References**

Ian C Covert, Scott Lundberg, and Su-In Lee. Explaining by removing: A uniﬁed framework for model explanation. The Journal of Machine Learning Research, 22(1):9477–9566, 2021.

Shah, Harshay, Prateek Jain, and Praneeth Netrapalli. "Do input gradients highlight discriminative features?" Advances in Neural Information Processing Systems 34 (2021): 2046-2059.

Tomsett, Richard, et al. "Sanity checks for saliency metrics." Proceedings of the AAAI conference on artificial intelligence. Vol. 34. No. 04. 2020.

Rong, Yao, et al. A Consistent and Efficient Evaluation Strategy for Attribution Methods. In International Conference on Machine Learning (pp. 18770-18795). PMLR, 2022

Wang, Jiachen T., and Ruoxi Jia. "Data banzhaf: A robust data valuation framework for machine learning." International Conference on Artificial Intelligence and Statistics. PMLR, 2023.



**Questions:**

I would appreciate if the authors could elaborate on the following points:

* How can the results be operationalized in practice? Can something constructive be derived from them?
* How would authors quantify the degree of robustness that is desirable (e.g., not invariant, not noisy)
* Do the results bear any connections to the benchmarking of feature attributions with is frequently done through removals (e.g., Tomsett et al., 2020; Rong et al, 2022)?

A discussion of these points in the paper could help to strengthen this work and I would be willing to reconsider my overall assessment based on the authors' replies.


**Limitations:**

I think the technical limitations are sufficiently clear, and mostly concern the Lipschitz-continuity assumptions and the bounded domains.

---

> ### Author Rebuttal · Authors · 2023-08-09
>
> > “Given the formalization done in prior work, the main results are not unexpected.”
>
> Regarding the “removal-based explanation” formalization from Covert et al., 2021, we do not intend to frame this as a contribution, hence its exposition in the Background (Section 2.2). Regarding the assumptions required for our analysis and whether the results are surprising, our goal is to understand these methods, so it is not a weakness in our analysis to have identified these necessary conditions. Viewed differently, these assumptions perhaps reflect how difficult it is to achieve robust attributions. To highlight several insights that are worthy and perhaps “surprising” technical contributions, these include: identifying the role of Lipschitz-like continuity in the conditional distribution and its role in making the conditional removal approach less robust; bridging two notions of implementation invariance (weak and strong) through the lens of robustness to model perturbation; and deriving the most robust summary techniques within classes of game-theoretic options.
>
> > “The finding that regularized networks have more robust (or less noisy) attributions, has already been established (e.g., by Shah et al., 2021, Dombrowski et al.).”
>
> The novelty lies in the different reasons why network regularization works, which are implied by our theoretical analysis, and which should be a topic of sufficient interest for readers. The previous works demonstrating this effect specifically study gradient-based methods [R1, R10], where weight decay plays a role in bounding a network’s Hessian. Here, we use weight decay because of its role in bounding the network’s Lipschitz constant. Furthermore, the role of Lipschitz continuity in our analysis implies that we can use other techniques to improve the robustness: beyond weight decay regularization, we can also use recent techniques for Lipschitz-constrained networks [R11-R13]. To help readers appreciate these points, we will be sure to clarify them in our revised Discussion.
>
> > “The Lipschitz continuity does not imply that the method needs to pass the parameter randomization test.”
>
> The parameter randomization test is a form of model perturbation, and Lipschitz continuity has no role in our analysis of robustness to model perturbations.
>
> > “How can the results be operationalized in practice? Can something constructive be derived from them?”
>
> From a practical perspective, our theory suggests (i) how to choose an attribution with strong robustness, and (ii) how to train a model that achieves more robust explanations. Regarding the latter point and specifically robustness to input perturbations, by largely reducing this problem to the model's Lipschitz constant, our work shows how to leverage a range of solutions developed in the robust learning literature, such as regularizing or constraining the network’s Lipschitz constant.
>
> There has been an ongoing debate in the literature about whether features should be removed using baseline values/marginal distribution or conditional distribution, as discussed and summarized in Chen et al. [R14]. Our analysis provides constructive guidelines for this choice from a robustness perspective: that is, if input perturbation is a concern, then baseline or marginal removal provides superior robustness. However, if model perturbation is a concern (e.g., “fairwashing” a model’s dependency on features that encode social biases), then conditional removal may be a better choice.
>
> > “How would authors quantify the degree of robustness that is desirable (e.g., not invariant, not noisy)?”
>
> This is an interesting question that we think is somewhat open-ended in terms of how to address it. The perspective we explored is how to choose an attribution that’s as robust as possible while maintaining a specific sense of being meaningful, and our results in Section 3.3 provide several options for how "meaningful" can be defined (see Definition 2). Briefly, we found that different combinations of constraints on the attribution lead to different optimal choices, for example the Banzhaf or Shapley value, which are both known in the literature. However, we acknowledge that there could be other definitions for “meaningful” explanations, and exploring alternative perspectives on this question is an interesting topic for future work.
>
>
> > “Do the results bear any connections to the benchmarking of feature attributions which is frequently done through removals (e.g. Tomsett et al. 2020; Rong et al., 2022)?”
>
> Thank you for raising this question, we are happy to cite several papers from that related line of work. After carefully considering the subject, we do not think there’s an interesting connection between these research directions; instead, we think they are best viewed as distinct, yet important notions of good behavior from an attribution method. And in fact, they may represent competing notions of good behavior, because as we allude to in Section 3.3, perfect robustness can only be achieved by designing an uninformative attribution method (which would necessarily perform badly on any removal/ablation-based metric). Thus, future work on new attribution methods should be sure to account for their performance from the perspective of both robustness and ablation-based metrics.
>
> > “Table 2 seems not to be referenced in the text and thus appears a bit context-less.”
>
> Thank you for reading our paper carefully. We will include a short comment after Lemmas 5 & 6 to reference Table 2.

---

> > ### Comment · Reviewer_dzDD · 2023-08-15
> > **Response to Rebuttal**
> >
> > Thank you for the clarifications.
> >
> > In my remark *"The Lipschitz continuity does not imply that the method needs to pass the parameter randomization test"*, I was referring to line 323-325 in the paper, which seems to suggest that the result in the previous Theorem 2 (robustness of removal-based explanations to model perturbations) already implies that we pass the parameter randomization test. However, as it is an upper bound, we cannot rule out constant attributions even in the case of model perturbation. While this is an interesting finding, I don't think it is a strict consequence of the theoretic analysis.
> >
> > I think if the write-up will be updated to incorporate the above major and minor points, in particular by
> >
> >    * including the discussion on applications of the results mentioned here
> >    * including at least some experiments on the additional page
> >    * clarifying that the formulation of the problem was transferred from Covert et al. (2021)
> >
> > this work will be accessible to the broader audience and a nice contribution, which warrants acceptance at the conference in my opinion.

---

> > > ### Author Response · Authors · 2023-08-16
> > >
> > > Thank you for getting back to us and for considering our work a nice contribution!
> > >
> > > Regarding the sanity check results, we recognize that the connection between our theory and experiments is not as clear as in the input perturbation case. The connection is that if you read the plots from right to left, we see that decreasing the model perturbation decreases the difference in attribution scores, which is implied by our theory via the bound in Theorem 2. However, your point is well taken, because our analysis does not imply that the difference must be large on the far right side; for this, we would require a lower bound rather than an upper bound. For example, for constant-valued attributions we would have $h(\text{summary}) = 0$, so the upper bound in Theorem 2 is always zero regardless of $||f - f’||$. We will substitute the word “naturally” with “empirically” in line 325 to avoid implying such a connection. In any case, given the significance of these sanity checks and the fact that they are the main benchmark for model perturbations, we thought that our findings were interesting and worth including.
> > >
> > > We will be sure to incorporate all the highlighted points in our revised Discussion and Experiments sections, as outlined in our responses.

---

### Official Review · Reviewer_2wCy · 2023-07-06

**Soundness:** 3 good
**Presentation:** 3 good
**Contribution:** 3 good
**Rating:** 6
**Confidence:** 3

**Summary:**

This paper theoretically analyzes the robustness of removal-based feature attributions against input perturbation, and model perturbation with different summary techniques. Empirical experiments on synthetic datasets support their theoretical analyses such as conditional sampling is more robust to model perturbations compared to baseline or marginal samplings. Experiments on real-world datasets (UCI wine quality and MNIST) give more insights regarding the robustness under different model training settings and comparison to gradient-based explanation methods.

**Strengths:**

(1) The proposed theoretical analysis of removal-based feature attributions is technically sound. The input perturbation and model perturbation together with the summary technique cover the different aspects of removal-based attributions.

(2) As there are more explanation techniques proposed, this work gives a good starting point to analyze the robustness limitations from a theoretical perspective, which can enable solid explanation evaluation on robustness among different explanation methods.

(3) Besides providing theory proofs, empirical experiments also validate and support the findings from theoretical analyses.

(4) Messages from the proposed theories are clear. The paper is well written and presented.

**Weaknesses:**

(1) The analyses are only limited to several removal-based feature attribution methods. Extending the analyses to other explanation techniques such as gradient-based explanations would make the contribution stronger. In fact, gradient-based explanations are more popularly used.

(2) The project is inspiring and gives insights into different removal-based attributions. However, the impact of the analyses can be broader if the authors can propose a technically sound robustness evaluation benchmark based on the theory.

(3) To analyze the robustness is computationally costly if the input data is high-dimensional or the model is huge. The author should consider extending the proposed analyses on complex datasets and some experimental results would make the paper stronger.

**Questions:**

The overall robustness proposed at the end of Section 4, which combines theorems 1 and 2, seems to be synthesized and is hard to adapt into a realistic scenario. For instance, the input perturbation robustness measures the consistency of explanations generated by one model, while model perturbation robustness analyzes the consistency given different (similar) target models. If use the overall robustness, it does not provide a clear message about different removal-based attributions.

**Limitations:**

Broader impact of the proposed framework should be discussed. More examples to use proposed robustness would be necessary to make the contribution stronger.

---

> ### Author Rebuttal · Authors · 2023-08-09
>
> > “Extending the analyses to other explanation techniques such as gradient-based explanations would make the contribution stronger. In fact, gradient-based explanations are more popularly used.”
>
> Other works have focused on the robustness of gradient-based methods [R1, R6], but the feature attribution literature lacks comprehensive theoretical analyses on the robustness of removal-based methods, which are also highly popular (e.g., the SHAP GitHub repository has ~19.9k stars, and the LIME GitHub repository has ~10.8k stars). Our paper makes a distinct contribution by addressing this gap. In the Discussion, we draw connections between the robustness of removal- and gradient-based methods. Primarily, the robustness of removal-based methods under input perturbations is related to a model’s Lipschitz continuity, whereas robustness of gradient-based methods is often related to the Lipschitzness of model gradient (i.e., Lipschitz smoothness). It would be interesting to pursue a unified, theoretical analysis of the robustness of both classes of methods, but we leave this topic to future work.
>
> > “The impact of the analyses can be broader if the authors can propose a technically sound robustness evaluation benchmark based on the theory.”
>
> The goal of a robustness benchmark is to provide an objective comparison between attribution methods, perhaps in the context of a specific model and dataset, but ideally with more generalizable conclusions. Several existing works address this topic empirically, and our work intends to provide a complementary theoretical perspective. We represent the essential properties of the model and dataset through our assumptions (namely $L$, $B$ and $M$), and this allows us to compare removal-based explanations in a more generalized sense by focusing on their implementation choices.
>
> That said, there is also a clear connection between our analysis and possible empirical benchmarks, and this can be seen in our experiments. For robustness to input perturbations, we verify that our theoretical findings are demonstrated in practice (Theorem 1) by testing the stability of attributions when adding noise to the input (e.g., Figure 4). These experiments resemble the empirical analysis conducted by Ghorbani et al. for gradient-based methods [R7], and they are also similar to metrics used in an existing benchmark library [R8]. In other words, our theory provides a way to characterize properties that other works study empirically.
>
> Regarding model perturbation, two types of benchmarks could include (i) making small perturbations to verify that the attributions don’t change much, and (ii) making large perturbations to verify that the attributions change significantly. In terms of existing works, Anders et al. explore the first direction [R6], and Adebayo et al. explore the latter [R5]. We incorporated the sanity checks from Adebayo et al. into our experiments (Figures 5-6), and Figure 2 includes an experiment with logistic regression similar to (i). There is certainly room to create new benchmarks along these lines, but a difficult design choice is how to define the small perturbations. Giving this choice proper consideration and generating results across a wide class of methods is best left as a topic for future work. Again, it is perhaps best to view our work as studying theoretically what other works have studied empirically, including not only [R5] and [R6] but also Slack et al. [R9].
>
> > “To analyze the robustness is computationally costly if the input data is high-dimensional or the model is huge. The author should consider extending the proposed analyses on complex datasets and some experimental results would make the paper stronger.”
>
> We thank the reviewer for this suggestion to improve the impact of our paper. We ran additional experiments with (i) ResNet-18 trained on CIFAR-10, and (ii) ResNet-50 trained on 10 classes from ImageNet (i.e., Imagenette). The figures are included in the attached file in the “global” response. We again observe empirical results consistent with our theoretical insights. In particular, ResNet-18 and ResNet-50 trained with stronger weight decay have more robust attributions under input perturbations. Also, under cascading model randomization (a form of increasing model perturbation) [R5], the unperturbed and perturbed ResNets indeed have increasingly dissimilar attributions (and as in our previous experiments, more so for removal-based methods than gradient-based ones).
>
> > “The overall robustness proposed at the end of Section 4, which combines theorems 1 and 2, seems to be synthesized and is hard to adapt into a realistic scenario.”
>
> The overall robustness in Corollary 1 is included for the sake of completeness and to show that our theory can account for this case. The result can be useful in situations where we want to interpret a model $f$ on a sample $x$, but the system is subject to simultaneous perturbations, such that we only have access to a perturbed model $f’$ and perturbed sample $x’$ (e.g., hiding a black-box model’s undesirable dependency on certain features with simultaneous, small model and input perturbations). However, we agree that such simultaneous perturbations may be uncommon in practice, so Corollary 1 can be deferred to the Appendix (thus making more room for our experiments).

---

> > ### Comment · Reviewer_2wCy · 2023-08-20
> >
> > Thank you for your response and clarification of my concerns. I think one future direction proposed by the authors, "It would be interesting to pursue a unified, theoretical analysis of the robustness of both classes of methods," would make a valuable contribution to the community. I raised my score to reflect my appreciation for the authors' efforts.

---

### Official Review · Reviewer_isMK · 2023-07-07

**Soundness:** 3 good
**Presentation:** 3 good
**Contribution:** 3 good
**Rating:** 7
**Confidence:** 3

**Summary:**

This paper studies the robustness properties of an explanation to small perturbations in the input space (i.e. like an adversarial example) and also to the model parameters. The authors use a number of Lipschitz-style bounds to then derive overall limits on how much explanations can change.

Update: As the reviewers have addressed the main concern on the Lipschitz constant, I have no major reason to reject this paper.

**Strengths:**

The bounds are principled and rigorous for explanations, as opposed to heuristic.

The paper considers multiple removal techniques and summary statistics to show how the bounds change based on the methods.

These bounds can sometimes lead to significant asymptotic differences.

**Weaknesses:**

The elephant in the room is that the results all need some kind of Lipschitz or Lipschitz-like bound, and that the final bound depends on the constant. This constant could be significant and not necessarily ignored, but there does not seem to be any evidence that the actual Lipschitz constant is at all close to being small enough to be useful.

There are barely any experimental results in the main paper. Almost the entirety is deferred to the appendix.

The results seem to focus mainly on synthetic settings and linear settings where Lipschitz constants and other constants can be directly computed. They also focus on fairly simple datasets such as UCI wine and MNIST.

**Questions:**

Can you measure or estimate the Lipschitz constants and show that these are meaningfully non-vacuous assumptions?

It would be better to highlight some kind of experimental result in the main paper.

Is there a reason or limiting factor preventing the work from applying to higher dimensional work such as text or image settings?

**Limitations:**

The limitations discuss some conservativeness of the bounds, which could be addressed with work on certified robustness.

---

> ### Author Rebuttal · Authors · 2023-08-09
>
> > “The elephant in the room is that the results all need some kind of Lipschitz or Lipschitz-like bound […] there does not seem to be any evidence that the actual Lipschitz constant is at all close to being small enough to be useful.”
>
> First, it’s worth emphasizing that our work aims to understand these attribution methods, not necessarily advocate for them, so it’s not a weakness that our analysis highlights the dependence on a quantity that is difficult to measure. In settings where $L$ can be large, our theory explains their lack of robustness; however, more positively, it also highlights the opportunity to ensure robustness via methods for controlling $L$, such as Lipschitz-constrained networks. Ultimately, this doesn't seem like a reason to reject our work, as we have correctly characterized this family of attribution methods. Note also that earlier works on gradient-based methods like Dombrowski et al. [R1] are based on properties of the network’s Hessian, which are more difficult to measure than the Lipschitz constant.
>
> Nonetheless, we can also provide a brief description of the current literature on estimating $L$. Exact computation of the Lipschitz constant, even for a two-layer neural network, is NP-hard [R2]. Even for simple networks on MNIST, current methods for estimating their Lipschitz constants give large estimates [R2-R4], which can be conservative and obscure meaningful conclusions with our theoretical results. That said, our theoretical analysis suggests empirical techniques for improving the robustness (e.g., increasing weight decay to shrink an upper bound on the Lipschitz constant), and these show the intended effect in our experiments. Therefore, our theoretical results regarding Lipschitz continuity are indeed useful in practice.
>
> > “It would be better to highlight some kind of experimental result in the main paper.”
>
> We thank the reviewer for this suggestion to improve our paper. To address this suggestion, we propose to move Corollary 1 to the Appendix to make room for a longer discussion of the experiments. Also, with the extra page allowed in the camera-ready version, we can move Figures 4 & 5 to the main text to highlight the empirical implications of our theory. We believe that a focus on theoretical insights and a concise description of experiments in the main text will allow the average reader to best understand the contributions and findings from our paper.
>
> > “Is there a reason or limiting factor preventing the work from applying to higher dimensional work such as text or image settings?”
>
> We thank the reviewer for this suggestion to improve the impact of our paper. To address this point, we ran additional experiments with (i) ResNet-18 trained on CIFAR-10, and (ii) ResNet-50 trained on 10 classes from ImageNet (i.e., Imagenette). The figures are included in the attached file in the “global” response. We again observe empirical results consistent with our theoretical insights. In particular, ResNet-18 and ResNet-50 trained with stronger weight decay have more robust attributions under input perturbations. Also, under cascading model randomization (a form of increasing model perturbation) [R5], the unperturbed and perturbed ResNets indeed have increasingly dissimilar attributions (and as in our previous experiments, more so for removal-based methods than gradient-based ones).

---

> > ### Comment · Reviewer_isMK · 2023-08-14
> >
> > Thanks for your reply!
> >
> > 1. I think you may have missed the point about the Lipschitz constant---I understand that it plays a role in the theoretical results, and that it is hard to estimate. I am just looking for evidence that this assumption does not make the results vacuous, especially since networks can have notoriously large Lipschitz constants. It is not necessary to describe the current literature on estimating Lipschitz constants. For example, what you suggested is a great example of what would fit the bill here---if a Lipschitz-constrained network does in fact improve the robustness, then this implies that the constant can in fact be small enough to have non-vacuous implications in practice.
> >
> > 2. You also mention using weight decay to shrink an upper bound on the Lipschitz constant---can you expand more on this? I did not see much about this in the main paper. As weight decay has been used before to improve robustness of the model, it makes sense that this could also help robustness of the attribution, but is there a deeper theoretical link here?

---

> > > ### Author Response · Authors · 2023-08-15
> > >
> > > Thank you for getting back to us! We answer your second question first, because it can help clarify the first question.
> > >
> > > - There is indeed a theoretical link between weight decay and a network’s Lipschitz constant. Weight decay shrinks an affine layer’s Frobenius norm, which upper bounds the layer’s spectral norm; and the product of all layers’ spectral norms upper bounds the network’s Lipschitz constant [R11]. We discussed this in the Experiments section in our Appendix, and in our revision we will be sure to highlight it in the main text.
> > >
> > > - Like you said, if our theoretical results were vacuous in practice, then reducing the Lipschitz constant of a network (or in practice some proxy to the Lipschitz constant) would not improve attribution robustness. However, we see that regularizing the Frobenius norm of a network, which upper bounds the network Lipschitz constant as discussed above, can indeed empirically improve attribution robustness (see Figure 4 in the Appendix and Figures R1-R2 in our rebuttal pdf). Hence, our theoretical results do bear practical implications.
> > >
> > > - On the other hand, we do recognize that some networks can potentially have very large Lipschitz constants (assuming we can somehow accurately approximate those Lipschitz constants). In such a situation, our theoretical results suggest that those networks should be avoided because the guarantees for their attribution robustness are not useful. In fact, those networks should also be avoided because the worst-case guarantees for their robustness against general adversarial attacks are not useful neither. This is why there are methods designed for improving the Lipschitz regularity of neural networks [R11-R13].

---

> > > > ### Author Response · Authors · 2023-08-21
> > > >
> > > > We would like to thank the reviewer again for the thoughtful review! We hope it is clear in our response that the Lipschitz bound in our analysis is useful. For example, reducing an upper bound of the Lipschitz constant empirically improves attribution robustness to input perturbations, which "fits the bill here" to show non-vacuous implications in practice.

---

### Official Review · Reviewer_aeSn · 2023-07-20

**Soundness:** 3 good
**Presentation:** 3 good
**Contribution:** 3 good
**Rating:** 6
**Confidence:** 4

**Summary:**

Previous research has primarily focused on the robustness of gradient-based feature attributions, but the robustness properties of removal-based attribution methods are not well understood. To fill this gap, the authors of the paragraph aim to theoretically analyze and characterize the robustness of removal-based feature attributions. They provide a unified analysis of such methods and prove upper bounds for the difference between intact and perturbed attributions under various settings of input and model perturbation. The authors validate their theoretical findings through empirical experiments on synthetic and real-world data, showcasing their practical implications.

**Strengths:**

1. This paper addresses a crucial XAI problem: Interpretation robustness concerning both input and model perturbations. The study may benefit the derivation of the theoretical impact of some important tricks in XAI, such as baseline value selection and marginal distribution approximation.

2. The derivation of robustness considers input perturbation and model perturbation comprehensively.

3. Section 4 provides valuable insights, revealing that interpretation robustness relies on both the Lipschitz continuity constant and feature number for input perturbation, while it depends on the ∞-norm and feature number in the case of model perturbation.

4. The experiment observations in Section 5 are also enlightening, highlighting the advantages of removal-based attribution over gradient-based attribution.

**Weaknesses:**

The paper should be re-arranged by including experiment results in the main content and moving some theoretical results to the appendix.

**Questions:**

Please refer to the Weakness.

**Limitations:**

The authors have adequately addressed the limitations of this work in the last section.

---

> ### Author Rebuttal · Authors · 2023-08-09
>
> > “The paper should be re-arranged by including experiment results in the main content and moving some theoretical results to the appendix.”
>
> We thank the reviewer for this suggestion to improve our paper. To address this suggestion, we will move Corollary 1 to the Appendix to make room for a longer discussion of the experiments. Also, with the extra page allowed in the camera-ready version, we can move Figures 4 & 5 to the main text to highlight the empirical implications of our theory. While we agree that it would be ideal to include our full experiments section in the main text, we believe that a focus on theoretical insights along a concise description of experiments will allow the average reader to best understand the findings from our work.

---

### Author Rebuttal · Authors · 2023-08-09

We thank all the reviewers for their insightful and constructive comments, which have helped us further improve our paper. The specific issues raised by each reviewer are addressed in the individual responses below. We hope you will consider raising the scores if we have adequately addressed your comments. References relevant to all responses are included below.

References

[R1] Towards robust explanations for deep neural networks - Dombrowski et al. 2022

[R2] Lipschitz regularity of deep neural networks: analysis and efficient estimation - Scaman and Virmaux 2018

[R3] Efficient and Accurate Estimation of Lipschitz Constants for Deep Neural Networks - Fazlyab et al. 2019

[R4] Efficiently Computing Local Lipschitz Constants of Neural Networks via Bound Propagation - Shi et al. 2022

[R5] Sanity Checks for Saliency Maps - Adebayo et al. 2018

[R6] Fairwashing Explanations with Off-Manifold Detergent - Anders et al. 2020

[R7] Interpretation of Neural Networks is Fragile - Ghorbani and Abid et al. 2019

[R8] OpenXAI: Towards a Transparent Evaluation of Post hoc Model Explanations - Agarwal et al. 2022

[R9] Fooling LIME and SHAP: Adversarial Attacks on Post hoc Explanation Methods - Slack and Hilgard et al. 2020

[R10] Do Input Gradients Highlight Discriminative Features? - Shah et al. 2021

[R11] Regularisation of Neural Networks by Enforcing Lipschitz Continuity - Gouk et al. 2018

[R12] The Singular Values of Convolutional Layers - Sedghi et al. 2018

[R13] Lipschitz-Margin Training: Scalable Certification of Perturbation Invariance for Deep Neural Networks - Tsuzuku et al. 2018

[R14] True to the Model or True to the Data? - Chen and Janizek et al. 2020

---

### Decision · Program_Chairs · 2023-09-21

**Decision:**

Accept (poster)

**Comment:**

This paper presents a theoretical analysis of the robustness of removal-based feature attributions, deriving bounds for how much attributions may change under input or model perturbations. The bounds are based on Lipschitz continuity of the model, with factors that depend on the type of removal distribution and summary method. This is primarily a theory paper, with the main contribution being the bounds given in Section 4, as well as some discussion of the implications of these bounds and how to improve them.

Reviews were positive, agreeing that robustness of attributions are an important problem in XAI and that the paper makes a solid theoretical contribution. Weaknesses include the limited experimental results which are almost entirely in the appendix, although reviewers appreciated these and the authors have said they will move some to the main paper for camera-ready. One reviewer also expressed concern that it may be difficult to obtain a Lipschitz constant low enough to provide a useful bound; however, this concern was allayed by the discussion and the score was raised significantly.